# Planar Symmetric Pattern Generation

**Ning Lin** [* 1]  **Luxi Chen** [* 1]  **Huaguan Chen** [1]  **Jiacheng Cen** [1]  **Chongxuan Li** [1]  **Wenbing Huang** [1]  **Hao Sun** [1]

## Abstract

Generating objects with specific symmetries is essential in various real-world scenarios. However, adapting existing 2D continuous representations to enforce planar group symmetry remains a challenge, as the transformation of non-reflective group elements may disrupt continuity. To overcome this limitation, we propose a symmetrization framework for arbitrary planar groups. Our method transforms any 2D continuous representation into a symmetric one while preserving continuity. We provide the mathematical formulation of this representation, demonstrate its approximation capability for symmetric functions, and detail the construction methodology. We validate our approach through three visual design tasks (pattern design, paper-cutting design and stylized topology design) and one material design task. Experiments confirm that our representation enables effective symmetry control and demonstrate its broader applicability. Code is available at https://github.com/GLAD-RUC/Sym2D.

## 1. Introduction

In visual arts and manufacturing engineering, symmetry plays an important role. It shapes geometric aesthetics and serves as a geometric prior that facilitates manufacturing processes, for example by enabling mold reuse, while also helping satisfy physical requirements. Classic examples range from wallpaper art to reflection-symmetric patterns in paper-cutting crafts formed by folding and unfolding, as well as periodic designs in architectural ornaments and lattice structures. All planar symmetry groups are illustrated in Fig. 1. In these tasks, designers often seek to impose constraints associated with planar symmetry groups while preserving spatial continuity and avoiding abrupt transitions.

---

[*]Equal contribution  [1]Gaoling School of Artificial Intelligence, Renmin University of China. Correspondence to: Hao Sun <hao-sun@ruc.edu.cn>.

*Proceedings of the 43$^{rd}$ International Conference on Machine Learning*, Seoul, South Korea. PMLR 306, 2026. Copyright 2026 by the author(s).

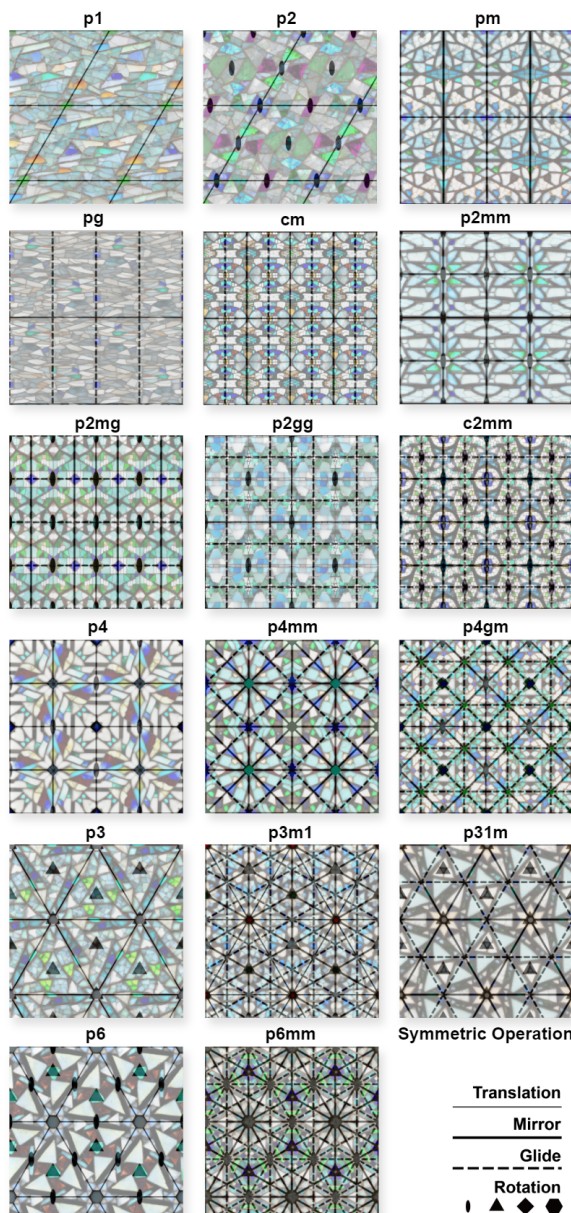

*Figure 1.* Generated images for the 17 planar groups using the prompt *stained-glass mosaic fragments...*. Annotated symmetry transformations demonstrate that our patterns exhibit perceptually perfect preservation of the target symmetries.

However, existing multimodal large language models (MLLMs) fail to produce strictly symmetric images even with rich textual and visual guidance. Previous work such as Bergmann et al. (2017) studies periodic texture generation, but does not provide a approach with exact symmetry across planar groups. We therefore seek symmetry-embedded representations and begin with the symmetrization of general 2D representations. A naive approach is to define the representation only on the asymmetric unit and extend it to the full plane via group transformations.

Unfortunately, for continuous representation, extending a function from the asymmetric unit by symmetry can introduce boundary discontinuities for non-reflective transformations: points just inside the boundary remain fixed, while points just outside are abruptly mapped to the opposite side. To resolve this, we propose a symmetric continuous representation framework that embeds any planar group into an affine reflection group to preserve boundary continuity, and constructs a continuous $G$-invariant field via a combination of high-symmetry coefficients and low-symmetry bases.

Based on this representation, we develop a unified pipeline for symmetry-constrained controllable generation with diffusion priors. Given a target planar group, we optimize the parameters of the proposed symmetric representation with respect to the loss functions. This separates symmetry constraints from other task-specific objectives: symmetry is handled by the representation, while other task-specific objectives can be imposed through loss functions. As a result, the same framework can be applied across diverse design tasks without collecting symmetric data or training symmetry-specific generative models.

We validate the versatility of our framework through three visual design tasks: (i) Pattern Design: Generating symmetric RGB images that align with given text descriptions (*Visual Semantic Constraints*). (ii) Paper-Cutting Design: Generating globally connected, symmetric binary masks subject to volume constraints and text semantics (*Visual + Connectivity Constraints*). (iii) Topology Design: Generating connected binary masks optimized for mechanical properties under volume constraints, alongside text-aligned stylized images (*Visual + Connectivity + Mechanical Constraints*). In addition to visual design, we further extend the same idea of symmetry control to metamaterial design based on diffusion prior. Experimental results demonstrate that our framework achieves stable symmetry control under physical constraints.

**Organization.** § 3 details the theoretical construction of the symmetric representation based on affine reflection group embeddings. § 4 presents the computation of the required bases. § 5 introduces the unified generative framework and loss functions. § 6 elaborates on the implementation and results of the three applications.

## 2. Preliminaries

**Planar groups.** In $n$-dimensional space, a Euclidean transformation is defined as the composition of an orthogonal transformation and a translation. The set of all Euclidean transformations constitutes the **Euclidean group**, denoted as $E(n)$. A **crystallographic group** $G$ is a discrete subgroup of $E(n)$ that contains $n$ linearly independent translations. Specifically, crystallographic groups are referred to as **planar groups** when $n = 2$, and **space groups** when $n = 3$. We primarily focus on the case where $n = 2$.

A classical result states that planar groups are classified into 17 types up to affine coordinate transformations. The simplest planar group is $p1$, generated by translations along two linearly independent directions. For any other planar group $G$, its elements contain nontrivial orthogonal components in addition to translations, and the quotient $G/T(G)$ is finite, where $T(G)$ denotes the translation subgroup of $G$. An important subclass is formed by **affine reflection groups**, which are generated by affine reflections. There are four such groups: $p2mm$, $p4mm$, $p3m1$, and $p6mm$.

**Symmetric functions.** For a planar group $G$, a function $f$ on $\mathbb{R}^2$ is called $G$-invariant if $f(g(\mathbf{x})) = f(\mathbf{x})$ for all $g \in G$. We denote by $C_G(\mathbb{R}^2)$ the space of continuous $G$-invariant functions. Consider a $p1$-invariant function $f$ periodic with respect to two fundamental translations $\mathbf{a}_1$ and $\mathbf{a}_2$. Let $S = \{m\mathbf{b}_1 + n\mathbf{b}_2\}$ be the reciprocal lattice, where $\langle \mathbf{a}_i, \mathbf{b}_j \rangle = 2\pi\delta_{ij}$. We choose one representative from each non-zero pair $(\mathbf{k}, -\mathbf{k})$. Denoting such a half reciprocal lattice by $S^+$, then $f$ can be written as a Fourier series

$$f(\mathbf{x}) = \frac{a_0}{2} + \sum_{\mathbf{k} \in S^+} [a_{\mathbf{k}} \cos(\langle \mathbf{k}, \mathbf{x} \rangle) + b_{\mathbf{k}} \sin(\langle \mathbf{k}, \mathbf{x} \rangle)]. \quad (1)$$

**Asymmetric unit.** A symmetric function is determined by its values on the asymmetric unit. The asymmetric unit represents the minimal, non-redundant portion of space from which the full periodic structure is obtained through the symmetry operations of the group. For $p1$, any fundamental period constitutes an asymmetric unit. However, for other planar groups, the existence of additional transformations implies that the asymmetric unit is only a portion of the period. Specifically for affine reflection groups, any point in space can be mapped into the asymmetric unit via a finite sequence of reflection transformations.

## 3. Symmetric Continuous Representations

Continuous representations play a critical role in modeling 2D and 3D objects. Continuity prior prevents abrupt variations during optimization, as illustrated in Fig. 2. In this section, we discuss how to symmetrize 2D continuous representations with respect to planar groups.

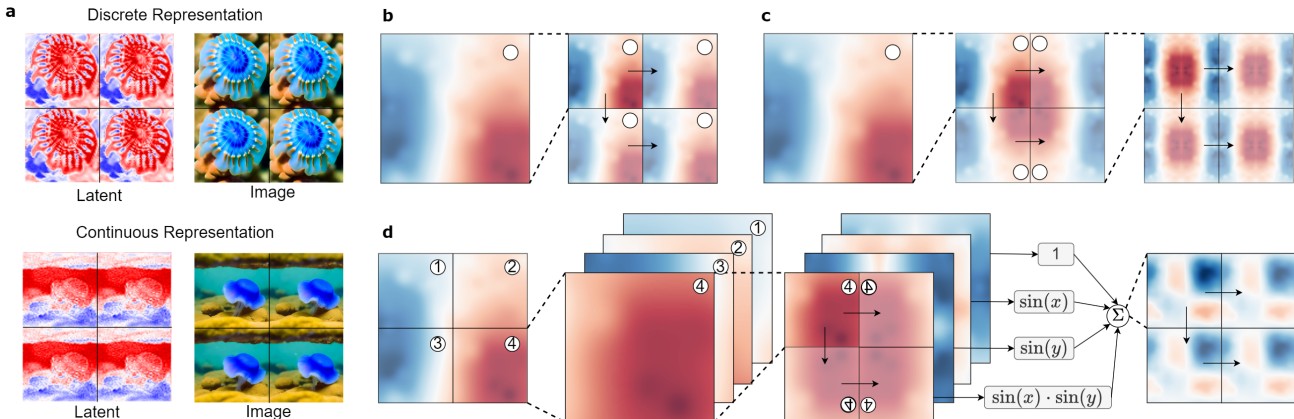

*Figure 2.* **Naive symmetrization vs. our symmetrization.** (a) A discrete representation extended by group transformations can induce abrupt latent transitions and local clustering, whereas continuous fields yield smoother result. (b) For continuous representations, naive extension may introduce seams near boundaries of asymmetric units for non-reflective groups. (c) For affine reflection groups, the same strategy is valid because reflections preserve boundary continuity. (d) Our construction combines high-symmetry coefficients and low-symmetry bases to produce a continuous symmetric function which is seamless across boundaries of asymmetric units.

### 3.1. Representations for Affine Reflection Groups

For planar groups, a naive strategy to adapt general representations to symmetric ones involves directly parameterizing the minimal asymmetric unit and extending it to the rest of the plane via group transformations.

In the specific case of affine reflection groups, since any point in the plane can be mapped into the asymmetric unit through a finite sequence of reflections and mirror reflections preserve boundary continuity, this approach can directly transform general continuous representations into symmetric ones. However, for other planar groups, this approach disrupts boundary continuity, as illustrated in Fig. 2. To address this, we require a method that bridges continuous modeling across different symmetries.

**Example 3.1** (Decomposition for $p1$ cases). *Consider the $p1$ symmetry and its Fourier bases. Let $c_1, s_1$ correspond to $\mathbf{k} = (1,0)$ and $c_2, s_2$ to $\mathbf{k} = (0,1)$. Using the multiple-angle formulas $\cos(nx) = T_n(\cos x)$ and $\sin(nx) = U_{n-1}(\cos x)\sin x$ ($T_n$ and $U_n$ denote the Chebyshev polynomials of the first and second kind) and trigonometric identities, truncated Fourier series can be written as a polynomial in $\mathbb{R}[c_1, c_2, s_1, s_2]$, denoted by $P_{\text{torus}}$. With $\eta_1 = 1$, $\eta_2 = s_1$, $\eta_3 = s_2$, and $\eta_4 = s_1 s_2$, every $f \in P_{\text{torus}}$ admits*

$$f(\mathbf{x}) = \sum_{i=1}^{4} p_i(c_1, c_2)\eta_i(\mathbf{x}), \qquad (2)$$

*where $p_i$ are bivariate polynomials.*

*Considering the unit basis vectors $\mathbf{e}_1, \mathbf{e}_2$ as fundamental translations for the $p1$ symmetric function, the coefficient functions $p_i(c_1, c_2)$ actually exhibit higher $p2mm$ symmetry, while the basis terms $g_i$ correspond to the lower $p1$ symmetry. Consequently, the low-symmetry function is suc-*

*cessfully decomposed into a combination of high-symmetry coefficients and low-symmetry bases.*

This example provides the intuition for constructing symmetric continuous representations. To generalize this method to arbitrary planar groups via affine reflection groups, we address three key questions: (i) **Existence of Higher Symmetry:** For any planar group $G$, does there always exist an affine reflection group $W_a$ serving as a supergroup? (ii) **Existence of Decomposition:** If such a group exists, does the decomposition into high-symmetry coefficients and a low-symmetry bases always hold for any $G$-invariant continuous function? (iii) **Computation of Bases:** How do we explicitly construct the low-symmetry bases?

We answer the first two existence questions using rigorous mathematical theory in § 3.2, and present the computational approach for the basis in § 4.

### 3.2. Hironaka Decomposition

We first address the problem of symmetry group inclusion. While the general answer is negative, we can transform any planar group into a subgroup of an affine reflection group up to conjugation by an invertible linear transformation.

**Theorem 3.2.** *Any planar group $G$ is conjugate to a subgroup of some affine reflection group $W_a$, i.e., there exists $A \in \text{GL}(2, \mathbb{R})$ such that $AGA^{-1} \subseteq W_a$.*

In the subsequent discussion, we will focus solely on the symmetries obtained after such an transformation, as the corresponding parameterizations can be mutually converted via invertible transformations.

We can construct $G$-invariant continuous representations by representation of affine reflection group $W_a$. Concretely,

---

**Algorithm 1:** Parameterized planar $G$-invariant continuous function $f$ evaluation at query point

---

**Input:** Continuous parameterization $\theta \mapsto \varphi_\theta$; parameters $\{\theta_i\}_{i=1}^r$; query point $\mathbf{p} = (x, y)$

**Output:** G-inv func $f(\mathbf{p})$

```
// Map p to asym unit of W_a
```
1 $\hat{\mathbf{p}} \leftarrow$ `reflect_to_asym_cell(p)`;
2 $f \leftarrow 0$ ;
3 **for** $i = 1$ **to** $r$ **do**
```
     // Get coefs on asym unit
```
4     $h_i \leftarrow \varphi_{\theta_i}(\hat{\mathbf{p}})$;
```
     // Comb with fixed bases
```
5     $f \leftarrow f + h_i \cdot \eta_i(\mathbf{p})$;
6 **end**
7 **return** $f$

---

we start from an underlying continuous representation and extend it via group transformations to obtain a $W_a$-invariant coefficient function which is continuous by construction. Given $r$ fixed $G$-invariant functions $\eta_1, \ldots, \eta_r$ and $W_a$-invariant functions $\{h_i\}_{i=1}^r$, we obtain a $G$-symmetric continuous representation in the form

$$f(\mathbf{x}) \;=\; \sum_{i=1}^r h_i(\mathbf{x})\,\eta_i(\mathbf{x}). \tag{3}$$

For $G$-invariant continuous functions, the existence of a high-symmetry coefficients and low-symmetry bases decomposition implies the approximation power of the form Eq. (3). Due to the algebraic nature of continuous functions, an exact decomposition may not exist in general. Instead, we establish an approximation guarantee.

**Theorem 3.3.** *Let $r = |W_a/G|$, and let $\Omega$ denote the unit cell. Then there exist $r$ fixed $G$-invariant basis functions $\eta_1, \ldots, \eta_r$ such that for any $f \in C_G(\mathbb{R}^2)$ and any $\epsilon > 0$, there exist $h_1, \ldots, h_r \in C_{W_a}(\mathbb{R}^2)$ satisfying*

$$\int_\Omega \left| f(\mathbf{x}) - \sum_{i=1}^r h_i(\mathbf{x})\,\eta_i(\mathbf{x}) \right| \mathrm{d}\mathbf{x} \;<\; \epsilon. \tag{4}$$

Consequently, given $r$ parameterized continuous functions $\varphi_{\theta_i} \in C(\mathbb{R}^2)$, we can obtain an expressive parameterization of $G$-invariant continuous functions. See Algorithm 1 for the evaluation procedure.

Regarding the construction of bases, we analyze trigonometric polynomials and then extend the results to continuous functions. Recall that the polynomial ring $P_{\text{torus}}$ is constructed specifically to encode the periodicity of the translation subgroup $T(G)$. For $T(G)$ is a normal subgroup of $G$, the $G$-action is closed on $P_{\text{torus}}$, and since $T(G)$ acts trivially, the invariant subring $P_{\text{torus}}^G$ (invariant truncated

Fourier series) is determined entirely by the induced action of the finite quotient group $G/T(G)$. Although $P_{\text{torus}}$ is a quotient ring and the action is non-linear, it remains a commutative domain. The fact allows us to apply results analogous to classical invariant theory (see, e.g., Sturmfels (2008)) if we relax the homogeneity requirement.

The Hironaka decomposition guarantees that there exist primary invariants $\theta_1, \theta_2$ and a finite set of secondary invariants $\eta_1, \ldots, \eta_r$ such that any $f \in P_{\text{torus}}^G$ admits the unique decomposition $f(\mathbf{x}) = \sum_{i=1}^r p_i(\theta_1, \theta_2)\,\eta_i(\mathbf{x})$. One can choose $\theta_1, \theta_2$ to be the primary invariants of $P_{\text{torus}}^{W_a}$, which in turn yields the desired low-symmetry bases $\{\eta_i\}$. Ex. 3.1 illustrates this construction in the $p1$ case.

### 3.3. General Theory in $\mathbb{R}^n$

Due to the similarity in algebraic structures, the Hironaka decomposition always exists for the general case of $\mathbb{R}^n$ (see Thm. C.9). However, not every crystallographic group is conjugate to a subgroup of an affine reflection group. This condition is determined by the structure of conjugacy classes of finite subgroups in $\mathrm{GL}(n, \mathbb{Q})$ (see Thm. B.6). Theoretical results indicate that the assertion of Thm. 3.2 still holds for $n = 3$ (see Thm. B.9), but fails for $n > 3$ (see Thm. B.11 and Thm. B.13).

Following a derivation analogous to the $n = 2$ case, we conclude that the required approximation condition Thm. 3.3 holds only for $n \leq 3$ (see § C.4). This is sufficient for practical applications, as we are primarily concerned with periodic structures where $n \leq 3$.

## 4. Computation of Basis

For a given planar group $G$, according to Thm. 3.3, to obtain a continuous representation, we first need to identify the high-symmetry group $W_a$. For the case of $n = 2$, this can be determined via table lookup. Next, we need to calculate the secondary invariants of $P_{\text{torus}}^G$ with respect to the low-symmetry bases of $W_a$. A construction via Fourier series can be found in Ex. 3.1 and § D.3. We next discuss how to perform the computation using existing results.

Assume that $G$ is a subgroup of an affine reflection group $W_a$, and $G$ contains a subgroup $H$ such that $G$ is generated by $H$ and an element $\tau$ of order 2. Suppose we have obtained the bases for $H$ with respect to $W_a$. If the action of $\tau$ maps each secondary invariant $\eta_i$ of $H$ to either itself or its negation, then the secondary invariants for $G$ consist of precisely those $\eta_i$ that remain invariant under $\tau$.

Calculations for the primary and secondary invariants of symmorphic planar groups (planar groups without glide reflections) can be found in Tab. 7 in Kim (2001). We focus on the treatment of non-symmorphic planar groups.

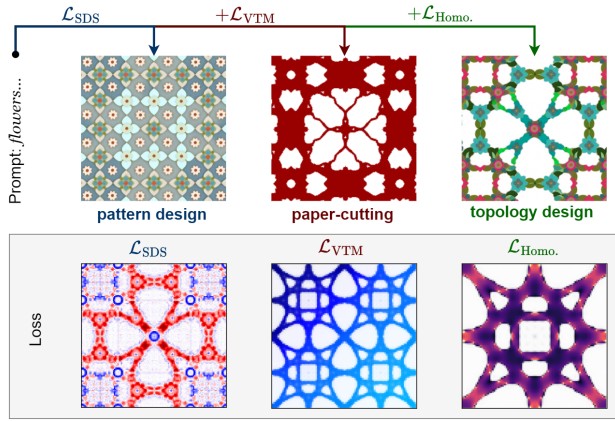

Figure 3. **Unified framework across three tasks.** Given the same prompt, pattern design optimizes $\mathcal{L}_{\text{SDS}}$. Paper-cutting adds the VTM penalty $\mathcal{L}_{\text{VTM}}$ to encourage connectivity. Topology design further incorporates $\mathcal{L}_{\text{Homo}}$ to maximize the effective bulk modulus evaluated through homogenization.

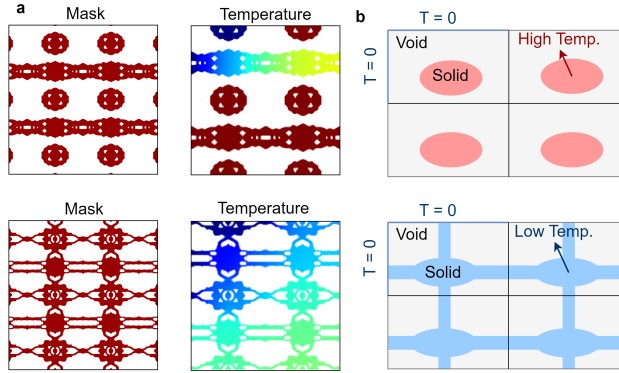

Figure 4. **VTM for connectivity.** Disconnected solid islands cannot dissipate heat to the sink on $\Gamma$ ($T = 0$) and thus become high-temperature regions: (a) generative examples and (b) an abstract schematic. Minimizing the VTM loss penalizes these hot components, promoting global connectivity.

Here, the additional generator $\tau$ is a glide reflection, and the subgroup $H$ consists of all orientation-preserving operations in $G$. The computation procedure is detailed in § D.1 and § D.2, and the resulting invariants are summarized in § G.

# 5. Generative Objectives and Loss Constraints

In this section, we formulate downstream generation over our symmetric continuous representation, combining diffusion priors and other task-specific constraints. Fig. 3 summarizes our progressively complex downstream tasks and the corresponding losses used to impose different constraints.

## 5.1. Pattern Design

**Objective.** Given a planar group $G$ and a text condition $y$, we consider the general task of symmetric pattern generation: generating 2D patterns $I$ that satisfy specific group symmetry constraints. We construct symmetric continuous representations within the latent space. Leveraging the symmetrization technique proposed in the previous section, we transform a general continuous representation into a continuous $G$-invariant vector field $f_\theta : \mathbb{R}^2 \to \mathbb{R}^C$ where $C$ denotes the number of latent channels, and $\theta$ represents the learnable parameters. We obtain the finite-dimensional latent tensor $z_\theta \in \mathbb{R}^{H \times W \times C}$ by evaluating the symmetric vector field on a regular grid.

**Distribution Constraint.** In the latent space, we employ Score Distillation Sampling (SDS) (Poole et al., 2023) as the generative objective to align the generated image with the conditional distribution of the diffusion model given $y$. The gradient of the SDS loss is formulated as

$$\nabla_\theta \mathcal{L}_{\text{SDS}} \propto \mathbb{E}_{t,\epsilon} \left[ w(t) \left( \widehat{\epsilon}_\phi(z_t, t, y) - \epsilon \right) \nabla_\theta z_\theta \right], \quad (5)$$

where $w(t)$ is a weighting term, $z_t = \alpha_t z_\theta + \sigma_t \epsilon$ represents the result of forward noise injection on $z_\theta$ at step $t$, $\alpha_t$ and $\sigma_t$ are the noise schedule parameters, and $\widehat{\epsilon}_\phi(z_t, t, y)$ is computed via classifier-free guidance as a weighted combination of the unconditional and conditional scores.

**Latent Vector Decoding.** In the output phase, we utilize a convolutional decoder to decode $z_\theta$ into an image. Prior to decoding, to further enhance generation quality, we perturb $z_\theta$ with moderate noise and denoise it using the diffusion model. Since convolutional operations are translation equivariant and the learned diffusion prior encourages coherent symmetric completions, the planar group symmetry of the image constrained by the symmetric initialization is approximately preserved after denoising and decoding. Compared with direct modeling in pixel space, latent-space parameterization is more computationally and memory efficient, supports higher-resolution generation, and often leads to more stable optimization in practice.

## 5.2. Paper-Cutting Design

**Objective.** Given a planar group $G$ and a text condition $y$, we extend the framework to generate periodic binary masks $m$ for paper-cutting. Here, we specifically restrict $G$ to the four affine reflection groups, as they correspond to valid fold-and-cut operations in physical fabrication. The goal is to optimize a mask $m$ where $m = 1$ (solid) and $m = 0$ (void) satisfy both semantic condition $y$ and maintain global connectivity to prevent structural detachment.

**Mask Parameterization.** We obtain mask from segment process. We define fixed foreground and background colors, $c_{\text{solid}}$ and $c_{\text{void}}$, and compute a soft density field $\rho_\theta = 1 - \sigma((d(f_\theta, c_{\text{void}}) - d_0)/\tau)$, where $\sigma$ denotes the sigmoid function, $\tau$ controls the binarization sharpness and $d_0$ controls the threshold. Unlike the pattern design task,

we bypass the pixel-space decoder and directly optimize the mask structure within the latent space, leveraging its geometric consistency stated in § 5.1.

To guide visual style, we sample from the field $c_{\text{solid}}\rho_\theta + c_{\text{void}}(1 - \rho_\theta)$ and get a pseudo-binary latent vector $z_\theta^{\text{bin}}$. Applying SDS directly to binary renderings often destabilizes optimization. Thus, we calculate the SDS loss on a progressive weighted mixture of the sampled latent vector $z_\theta$ and the rendered $z_\theta^{\text{bin}}$. Furthermore, we apply Low-Rank Adaptation (LoRA) (Hu et al., 2022) to the pre-trained diffusion model using a paper-cutting dataset, ensuring the diffusion prior aligns with the domain distribution.

**Connectivity Constraint.** To prevent unmanufacturable isolated islands, we employ the virtual temperature method (VTM) (Li et al., 2016) to enforce global connectivity. Conceptually, we model the material distribution as a heat conduction system within a domain $\Omega$: solid regions act as conductors that generate and transport heat, while void regions function as insulators. By placing a heat sink at a boundary $\Gamma$, any disconnected component unable to dissipate heat will accumulate high temperature. Therefore, minimizing the maximum temperature eliminates solid islands.

Formally, the virtual temperature field $T$ is governed by the steady-state Poisson equation $-\nabla \cdot (k(\rho_\theta)\nabla T) = s(\rho_\theta)$ subject to Dirichlet boundary conditions $T = 0$ on $\Gamma$ and adiabatic conditions elsewhere. The thermal conductivity $k(\rho_\theta)$ and heat source $s(\rho_\theta)$ are coupled to the density via SIMP interpolation, e.g., $k(\rho) = k_{\min} + (k_0 - k_{\min})\rho^p$, where $k_0$ denotes the conductivity of the solid material and $k_{\min}$ is a small conductivity used to avoid degeneracy in void regions. The penalization $p > 1$ encourages $\rho$ to converge toward binary values. We penalize the maximum temperature to enforce connectivity

$$\mathcal{L}_{\text{VTM}} = \left( \frac{1}{|\Omega|} \int_\Omega T(\rho_\theta)(\mathbf{x})^p \, d\mathbf{x} \right)^{1/p}, \qquad (6)$$

where the $p$-norm approximates the max operator, and gradients are efficiently computed via the adjoint method.

To ensure global connectivity of a periodic mask, the choice of the domain $\Omega$ and the boundary $\Gamma$ is crucial. Enforcing connectivity only within a single unit cell is insufficient, as it may still permit disconnected strip-like patterns. We therefore set $\Omega$ to be a $2 \times 2$ supercell and impose an intersection constraint with $\Gamma$ inside one constituent cell. In Thm. E.3, we prove that this strategy guarantees the global connectivity of the mask. Choosing $\Gamma$ as an arbitrary interior point of $B$ would unnecessarily force the mask to pass through a prescribed location. To avoid this over-constraint, we take $\Gamma$ to be two boundary segments of the unit cell. As illustrated in Fig. 4, isolated solid islands in a periodic mask exhibit high temperatures, and the VTM loss penalizes these hot regions, thereby promoting global connectivity.

**Volume Constraint.** To regulate material usage and control the pattern's sparsity, we introduce a volume constraint. Denoting the target volume fraction as $\rho_0 \in (0, 1)$, we define the loss over the unit cell $\Omega$ as

$$\mathcal{L}_{\text{vol}} = \left( \frac{1}{|\Omega|} \int_\Omega \rho_\theta(\mathbf{x}) \, d\mathbf{x} - \rho_0 \right)^2. \qquad (7)$$

This term enables systematic control over the solid-to-void ratio, allowing for stylistic adjustments.

### 5.3. Topology Design

**Objective.** Distinct from the previous tasks, we must simultaneously synthesize (i) a binary structural mask $m$ defining the geometric topology, and (ii) the texture content $I$ within the mask, effectively treating $m$ as an alpha transparency channel. The objective is to maximize the effective mechanical properties under symmetry and volume constraints while maintaining aesthetic appeal and stylistic consistency.

Provided the background color, the image parameterization follows the pattern design, while the mask parameterization adopts the segmentation approach from § 5.2. Next, we discuss the mechanical loss on the mask.

**Mechanical Constraints.** For periodic structures, stiffness is typically evaluated via the effective elastic properties. We employ homogenization-based topology optimization, calculating the equivalent elastic tensor

$$E_{ijkl}^H(\rho) = \frac{1}{|\Omega|} \int_\Omega E_{pqrs}\epsilon_{pq}^{A(ij)}\epsilon_{rs}^{A(kl)} \, d\Omega, \qquad (8)$$

where $\epsilon_{rs}^{A(kl)}$ represents the actual strain field induced within the unit cell $\Omega$ under the $(kl)$-th unit test strain. With the unit test strains and periodic boundary conditions, the structural equilibrium equations in $\Omega$ are given by

$$\nabla_S^\top \boldsymbol{\sigma} = \mathbf{0}, \quad \boldsymbol{\epsilon} = \nabla_S \mathbf{u}, \quad \boldsymbol{\sigma} = \mathbf{D}(\rho_\theta)\boldsymbol{\epsilon}. \qquad (9)$$

Here, $\nabla_S$ is the symmetric gradient operator, and $\mathbf{D}(\rho) \propto \mathbf{D}_0$ is the SIMP-interpolated constitutive matrix, where $\mathbf{D}_0$ is the baseline elasticity matrix determined by the material's Young's modulus and Poisson's ratio. In the 2D case, using Voigt notation $(11 \to 1, 22 \to 2, 12 \to 3)$, the bulk modulus is expressed as $c = E_{11}^H + E_{12}^H + E_{21}^H + E_{22}^H$. We aim to maximize this modulus via the loss

$$\mathcal{L}_{\text{Homo}} = -c(\rho_\theta), \qquad (10)$$

and gradients can be computed using the adjoint method.

**Masked Latent Decoding.** To ensure texture content appears exclusively in solid regions, we use the mask to exclude background latents prior to decoding. Following denoising and decoding, the mask is interpolated to the image resolution and serves as the alpha transparency channel.

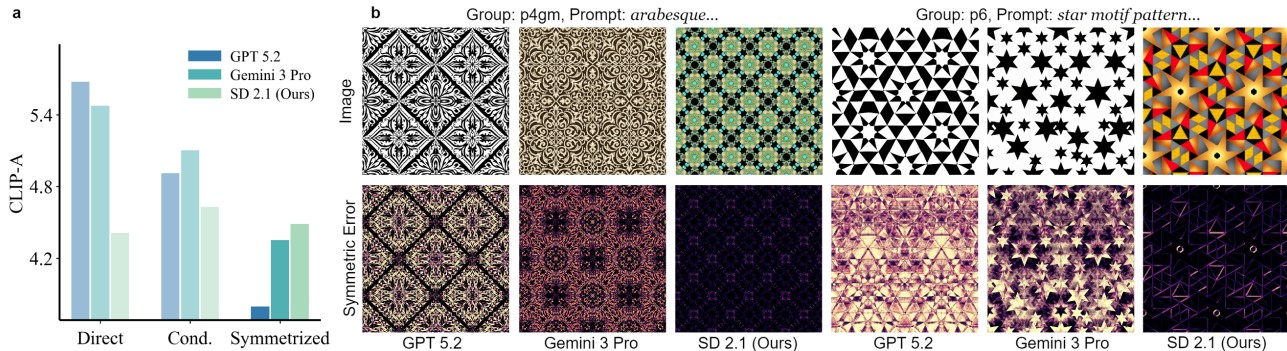

*Figure 5.* **Results of symmetric pattern design.** (a) CLIP-A scores (higher is better) across 17 prompts under direct generation, conditional generation, and post-symmetrization settings. Our method achieved the best results in generating strictly symmetric images in the post-Symmetrization setting. (b) Representative results for groups $p4gm$ and $p6$. For each group, the top row shows the images from the conditional generation setting, and the bottom row visualizes the pixel-wise MSE between images from conditional generation and post-symmetrization settings, and darker regions indicate smaller errors. This comparison highlights geometric inconsistencies introduced when enforcing strict symmetry in the baseline.

*Table 1.* Comparison with symmetrization based on basis projection (BP) under different values of fraction of bases $\alpha$ of Nyquist frequencies (see § A.2) and resolution $n$ of latent features.

| Method | $n = 64$ | $n = 128$ | $n = 256$ |
|---|---|---|---|
| BP ($\alpha = 0.25$) | 3.81 | 3.07 | 3.46 |
| BP ($\alpha = 0.50$) | 3.98 | 3.38 | 3.61 |
| BP ($\alpha = 1.00$) | 4.05 | 3.42 | 3.48 |
| Ours | **4.30** | **4.20** | **3.99** |

## 6. Experiment

In this section, we present the experimental settings and results for pattern design, paper-cutting design, and topology design. All symmetric parameterizations are implemented via symmetrization of a hash-coded bilinear interpolation scheme (Müller et al., 2022). More implementation details and results are provided in § F.

### 6.1. Results of Pattern Design

For the general symmetric pattern design task, we employ Stable Diffusion 2.1 (SD 2.1) (Rombach et al., 2022) as our base model. For our method, we utilize the generative process described in § 5.1.

**Visualization.** To verify the stability and diversity of our approach, we selected a specific text prompt to generate patterns for all 17 symmetry groups, as illustrated in Fig. 1. The results without markers are provided in Fig. 11.

**Comparison with Text-conditioned Generation.** To benchmark our approach against state-of-the-art generative capabilities, we use MLLMs, i.e., GPT-5.2 (OpenAI, 2025) and Gemini 3 Pro (Google DeepMind, 2025), as baselines. We conduct experiments across three distinct settings to assess geometric precision and aesthetic quality: (i) Direct

Generation: Models generate images directly from the text prompts without additional constraints. (ii) Conditional Generation: Generate images with symmetry control. For MLLMs, we provide the text prompt alongside an auxiliary visual instruction. (iii) Post-Symmetrization: To evaluate geometric consistency, we fit images generated in the second setting with our symmetric parameterization in § 3.2 using MSE loss. This process enforces strict symmetry on the generated images.

To ensure a comprehensive evaluation, we constructed a test set of 17 text prompts, one corresponding to each symmetry group. The prompts were adapted from the examples in Tab. 9 of Shubnikov & Koptsik (1974) and were labeled and simplified using Gemini. We evaluate the aesthetic quality of generated patterns using the CLIP aesthetic (CLIP-A) metric (Schuhmann et al., 2022).

The quantitative results in Fig. 5 show that state-of-the-art MLLMs achieve strong aesthetic scores in direct generation, but their performance degrades once symmetry constraints are imposed. In particular, post-symmetrization enforces exact symmetry but substantially reduces the aesthetic quality of MLLM-generated images, indicating that these models produce visually appealing images without faithfully satisfying symmetry constraints. In contrast, our method, despite using the weaker SD 2.1 backbone, maintains higher aesthetic quality under symmetry enforcement and outperforms the baselines in the post-symmetrization setting. This trend is further supported by the qualitative MSE results in Fig. 13. For full generated results of our method, see Fig. 12.

**Comparison with Other Symmetrization.** There are also several naive approaches to symmetrize representation. For example, one can project an asymmetric function onto the invariant function space spanned by invariant basis functions. This approach also preserves continuity, but may

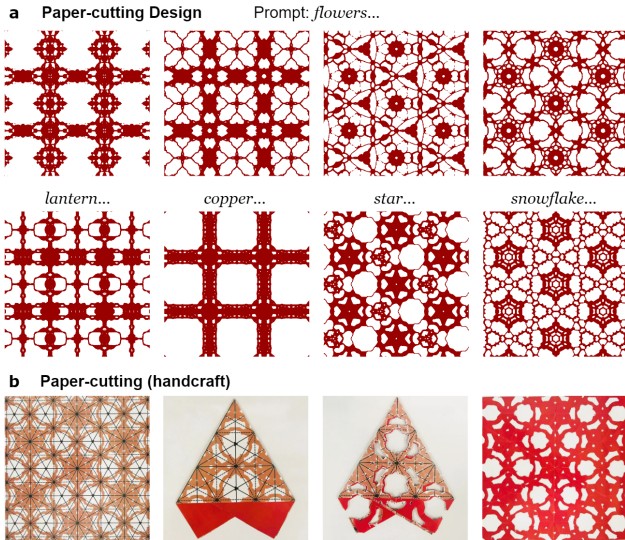

**a** **Paper-cutting Design**    Prompt: *flowers...*

*lantern...*    *copper...*    *star...*    *snowflake...*

**b** **Paper-cutting (handcraft)**

*Figure 6.* **Qualitative results of paper-cutting design.** (a) Diverse patterns generated by our method under varying text prompts while strictly adhering to symmetry and connectivity constraints. (b) Physical realization demonstrates the digital pattern, folding plan, crafting process, and final paper-cutting result to verify structural integrity and manufacturability.

lead to substantial computational and memory overhead. Moreover, projection aggregates values across group orbits, reducing variance and producing overly smooth initializations, which can degrade image quality during generation. We provide a more detailed analysis in § A.2. Experimentally, we compare our method with this projection-based baseline for pattern generation in Table 1 on prompts targeting the $p1$ symmetry group. The results show that our symmetrization achieves better visual performance.

## 6.2. Results of Paper-Cutting Design

For the paper-cutting task, we adopt Stable Diffusion XL base 1.0 (SDXL 1.0) (Podell et al., 2023) as the backbone and fine-tune a LoRA module on the dataset of Wang et al. (2025) using `diffusers` library (von Platen et al., 2022), following the procedure described in § 5.2.

We report qualitative results in Fig. 6. Our method generates semantically aligned patterns while preserving connectivity. We further validate manufacturability by fabricating a generated mask. In practice, we dilate thin structures for cuttability, fold the sheet along the reflection axis, cut through the folded layers, and then unfold to obtain the final paper-cutting. This end-to-end fabrication verifies the practical feasibility of our designs.

## 6.3. Results of Topology Design

For the topology design task, we employ the SDXL 1.0 as our backbone. We benchmark our approach against

the CLIP-based topology optimization method proposed by Zhong et al. (2023). Following their protocol, we utilize their set of 12 test prompts of the stylization gallery. Aesthetic quality is evaluated using the CLIP-A. When testing the baseline, we retain its original mask modeling method and apply our symmetrization procedure to construct the corresponding symmetric continuous representation.

To rigorously compare the efficacy of the generative objectives, we maintain identical hyperparameters and optimize the representation using either the baseline's CLIP loss or our SDS loss. The target volume fraction is set to 0.45. We conduct experiments across two distinct settings: (1) varying the lattice angle $\gamma \in \{90°, 85°, 80°\}$ under the $p1$ symmetry group, and (2) varying the symmetry constraints across $p1$, $p2$, and $pm$ groups. We sample results at five volume fraction intervals $(0.43, 0.44, 0.45, 0.46, 0.47)$ by adjusting the threshold applied to the mask for each setting.

The results are illustrated in Fig. 7. In the variable angle setting, our method achieves mechanical properties comparable to the baseline while maintaining a substantial lead in aesthetic quality. Notably, in the variable symmetry setting, particularly for $p2$ and $pm$ groups, our method outperforms the baseline in both mechanical performance and aesthetic scores, highlighting its improved stability when optimizing under higher-symmetry constraints. The results demonstrate that our SDS-based optimization successfully synthesizes structures that are not only mechanically robust but also visually consistent with the text prompts. Additional results of our method across different symmetry groups are provided in Figs. 14 to 16.

## 6.4. Extension to Metamaterial Design

We further extend our framework to mechanical metamaterial design. Structures are represented as binary unit-cell patterns, and symmetry is closely related to their mechanical properties. We consider a zero-shot symmetry-control setting: the base diffusion model is trained only on topology-optimized unit cells with the basic $p1$ symmetry. At sampling time, we impose additional planar-group symmetries that share the same lattice as the training samples. The goal is therefore to generate symmetric unit-cell structures with mechanical constraint without symmetric training data.

The training dataset contains 36,000 binary samples generated by homogenization method using the optimality criteria method (Xia & Breitkopf, 2015). Each sample is initialized from random perturbations and optimized to maximize the bulk modulus subject to a prescribed volume constraint. The diffusion model is parameterized by a convolutional U-Net and trained for 100 epochs.

We apply the same parametric symmetric representation as in the previous visual design tasks. The SDS loss guides

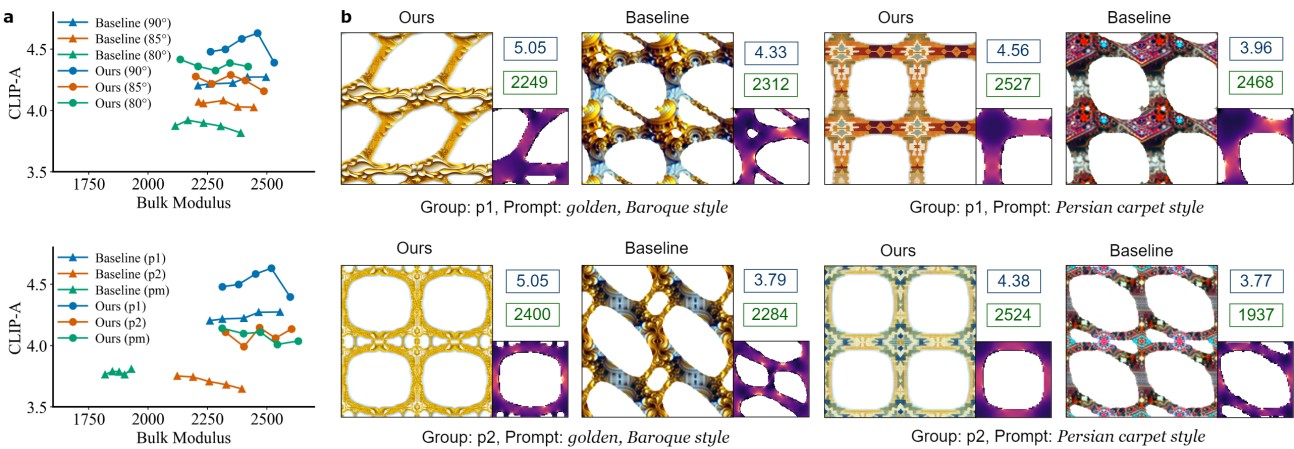

*Figure 7.* **Quantitative and qualitative comparison on topology design.** (a) The top row compares performance under different angles $\gamma$ for the $p1$ group, while the bottom row compares different symmetry groups. Our method consistently achieves higher CLIP-A while matching or exceeding the baseline's mechanical performance. (b) The blue and green boxes indicate the value of CLIP-A and Bulk modulus, respectively. Our method generates structures with clearer semantic patterns and better mechanical performance.

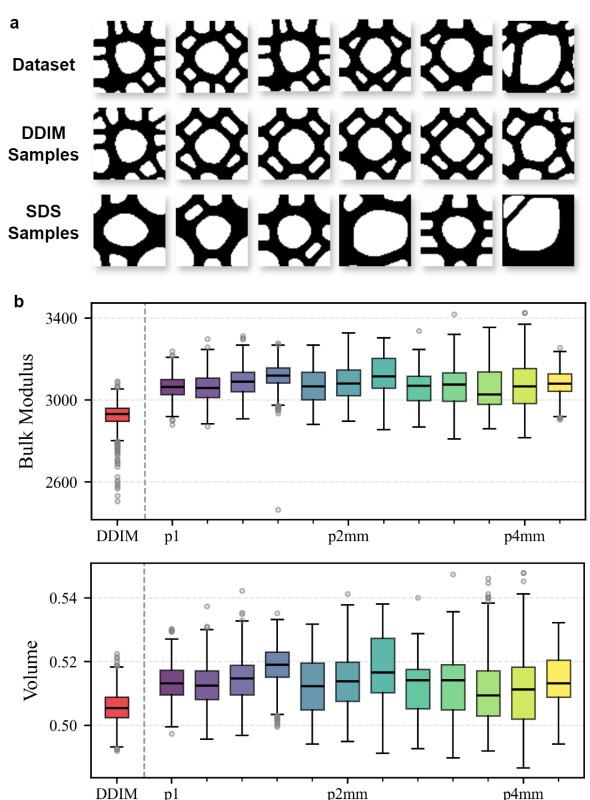

*Figure 8.* **Generated Sample Distribution under Symmetry Constraints.** (a) Samples of unit cell from the training dataset, DDIM generation, and our SDS with $p1$ symmetry constraint. (b) Distribution of bulk modulus and volume for 1000 samples generated by DDIM and by SDS with symmetry constraints of the first 12 planar groups. The proposed method produces structures with higher mechanical performance while satisfying the volume constraint without symmetry-specific training data.

the optimization in the symmetric representation space toward the high-performance structure distribution captured by the pretrained diffusion model, while the representation itself ensures that the generated binary masks satisfy the prescribed planar-group symmetry.

We evaluate the method on the first 12 planar groups with 1,000 generated samples for each setting. As shown in Fig. 8, the proposed method produces diverse symmetric structures with strong mechanical performance. Quantitatively, the generated samples achieve high bulk modulus with a volume MAE below $1.5\%$. These results demonstrate that our framework is not limited to visual pattern generation, but can also be applied to material design problems where symmetry constraints should be enforced while maintaining structural performance.

# 7. Conclusion

We introduce a general symmetrization method that transforms any continuous 2D representation into an exactly symmetric and spatially continuous one under arbitrary planar groups, avoiding the boundary discontinuities of naive asymmetric-unit extensions. By embedding planar groups into affine reflection groups, our method constructs a differentiable representation from high-symmetry coefficients and low-symmetry bases. This further yields a zero-shot controllable generation framework for symmetry-constrained design: symmetry is enforced directly in the representation space, while other task-specific objectives guide optimization. Experiments on three visual design tasks and one material design task demonstrate the broad applicability of our framework, as well as its ability to maintain symmetry control under diverse geometric and physical constraints.

## Acknowledgment

We would like to acknowledge that the work is financially supported by the National Natural Science Foundation of China (No. 62276269, 62376276), the Beijing Natural Science Foundation (No. F261002), and the Beijing Nova Program (No. 20230484278).

## Author Contributions

Ning Lin organized the project and led the theoretical development in § 2–§ 4, including the corresponding theoretical proofs. Ning Lin and Luxi Chen jointly led the framework design in § 5 and the experimental studies in § 6. Specifically, Ning Lin mainly contributed to the paper-cutting and topology design tasks, together with their controllable generation experiments. Luxi Chen mainly contributed to pattern design task, LoRA fine-tuning in § 6.2, diffusion model training in § 6.4. Huaguan Chen and Jiacheng Cen contributed to parameter tuning, figure preparation, and visualization. Chongxuan Li, Wenbing Huang, and Hao Sun jointly supervised and guided the project. All authors contributed to writing and revising the manuscript.

## Impact Statement

This paper presents work whose goal is to advance the field of machine learning. There are many potential societal consequences of our work, none of which we feel must be specifically highlighted here.

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

# Contents of Appendix

# A. Background

## A.1. Controllable Generation

**Zero-shot Controllable Generation.** Diffusion model (Ho et al., 2020) and related flow matching models (Lipman et al., 2023) have been widely applied across diverse domains, including image synthesis (Rombach et al., 2022; Podell et al., 2023; Lu et al., 2025), video generation (OpenAI, 2024; Guan et al., 2025), 3D object generation (Poole et al., 2023; Chen et al., 2025), and material design (Jiao et al., 2023; Wu et al., 2026b; Li et al., 2026). Building on these advances, controllable generation aims to guide these generative models toward samples that satisfy user-specified conditions or constraints, such as text prompts (Chung et al., 2023; Domingo-Enrich et al., 2025; Ye et al., 2024), spatial constraints (Zhang et al., 2023), or physical properties (Vatani et al., 2025; Wu et al., 2026a; Chen et al., 2026).

Given a fixed constraint cost model, existing controllable generation methods can be broadly categorized into training-based and training-free approaches. Training-based methods, such as Domingo-Enrich et al. (2025), rely on additional constraint-related data to fine-tune the base generative model. In contrast, training-free, or zero-shot, methods, such as Chung et al. (2023), require no extra training data and instead incorporate the cost model as a plug-in guidance module during generation.

Our work belongs to the training-free paradigm of controllable generation, focusing on the incorporation of exact symmetry constraints into diffusion-based optimization. Symmetric pattern generation is naturally a zero-shot problem, since real-world datasets rarely contain samples that strictly satisfy prescribed symmetry constraints. We decouple exact symmetry from task-specific objectives by enforcing symmetry through the representation, while imposing other requirements via loss functions. Starting from randomly initialized design parameters, we generate designs by optimizing the symmetric representation with the corresponding task-specific losses. The overall generation pipeline and its correspondence to the main technical components are illustrated in Fig. 9.

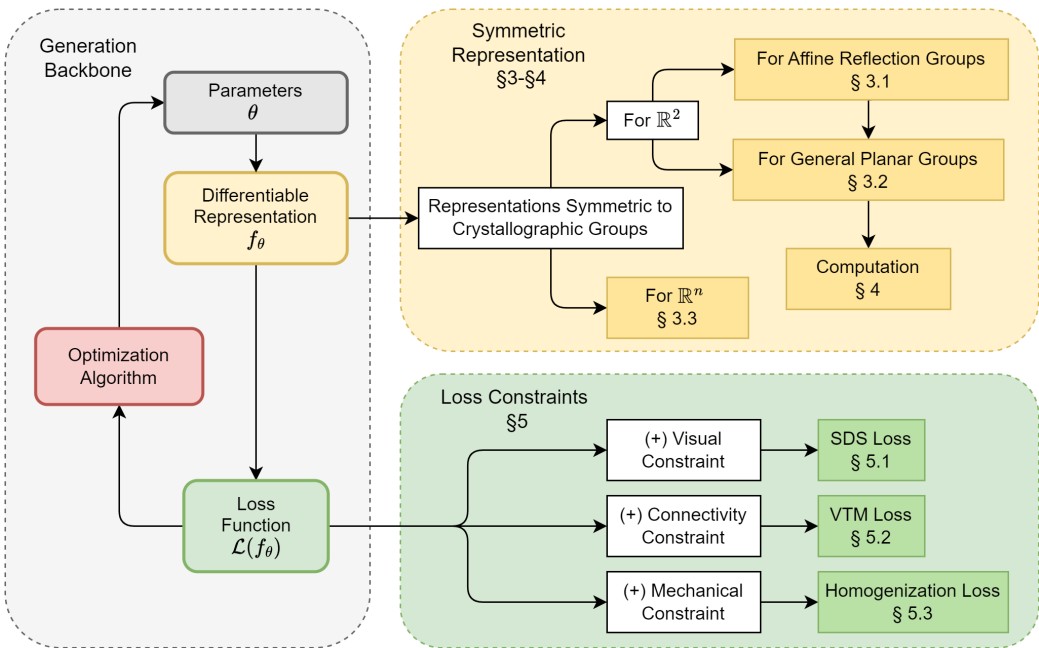

*Figure 9.* Overview of our training-free controllable generation pipeline.

**Stylized Topology Generation.** In mechanical engineering, topology optimization improves structural performance by optimizing the spatial distribution of materials within a prescribed design domain (Bendsoe & Sigmund, 2013), with representative algorithms including the solid isotropic material with penalization (SIMP) method (Andreassen et al., 2011). In periodic material design, topology optimization is often combined with homogenization methods, which evaluate the effective macroscopic properties of the optimized microstructures (Bendsoe & Sigmund, 2013).

Several recent works have explored the trade-off between mechanical performance and visual appearance in material design (Martínez et al., 2015; Zhong et al., 2023). The most related work is Zhong et al. (2023), which jointly optimizes the mechanical performance of a binary material mask, represented by the alpha channel, and its semantic similarity to a text

description, measured through the RGB image channel. This approach relies on pre-trained vision-language models, which provide a shared embedding space for visual and textual representations (Radford et al., 2021; Chen et al., 2023). Unlike our approach in § 5.3, it models the material mask as an additional channel rather than extracting it through segmentation. In addition, connectivity is encouraged by penalizing the area of small connected components.

However, such channel-based mask modeling can be vulnerable to the generative prior. Since the diffusion prior favors visually realistic RGB content, the optimized image may fill the masked material region with background-like textures or semantic content, weakening the correspondence between the visual appearance and the actual material layout. Moreover, the connectivity penalty only suppresses disconnected components rather than actively reconnecting them to the main structure, making it difficult to repair or refine the optimized topology.

Our method instead focuses on periodic structure design and derives the material layout from the generated appearance through segmentation, which better aligns the visual content with the optimized mask structure. Motivated by diffusion-prior-based optimization methods such as DreamFusion (Poole et al., 2023), we introduce a diffusion prior to improve visual quality while optimizing the periodic material representation. Furthermore, rather than merely penalizing small isolated components, we incorporate a differentiable VTM loss into the objective, which provides a more direct mechanism for controlling and improving structural connectivity.

**Material Design and Symmetry Constraints.** Planar- and space-group symmetries provide a natural language for describing periodic structures. In materials science, crystals are a representative class of discrete periodic structures: their unit cells are microscopic atomic arrangements, often modeled as graphs whose nodes correspond to atoms. Metamaterials, in contrast, provide a continuous counterpart: their unit cells are mesoscopic structures composed of continuum materials and are represented by binary masks, where the values indicate the absence or presence of material. For both crystals and metamaterials, symmetry plays a crucial role in determining material properties (Levy et al., 2025; Mao et al., 2020).

Recent works in crystal and metamaterial generation have therefore studied how to ensure that generated structures exactly satisfy prescribed symmetries. In crystal generation, Jiao et al. (2024) considers generation under a given space-group type together with the site symmetry of atoms, which can be regarded as a symmetry template. From the perspective of controllable generation, Jiao et al. (2024) puts symmetry constraints into the generation process through this prescribed template. Levy et al. (2025) follows a conditional-generation formulation and further relaxes the dependence on predefined templates by treating the atomic site symmetries as generative variables. Given a space-group type and the number of orbit representatives, it learns the site-symmetry assignments from data and reconstructs the full crystal by replicating the generated asymmetric unit.

In metamaterial design, Mao et al. (2020) study the generation of two-dimensional metamaterials under prescribed planar-group symmetries. Their method requires training a symmetry-specific generative model for each plane group using symmetric unit-cell masks. In contrast, when applied to symmetric metamaterial generation, our method does not require symmetry training data or a separately trained model for each symmetry group. Instead, our method only needs a generator trained on $p1$ periodic patterns, i.e., patterns without additional symmetry. To generate a pattern with a desired symmetry, we optimize our symmetric representation using the SDS loss. Therefore, changing the target symmetry does not require collecting symmetry-specific data or retraining the generator.

### A.2. Other Symmetrization.

We compare our method with several natural alternatives for enforcing planar-group symmetry. These alternatives can be divided into two categories: enforcing symmetry directly in the parameter space of a representation, and transforming the output function after evaluation.

**Constraining the Parameter Space.** A seemingly direct approach is to impose symmetry constraints on the parameters of a continuous representation. However, this strategy is generally impractical for two reasons. First, plane groups are infinite because they contain translations, which would in principle induce infinitely many equality constraints. Solving such constraints numerically is therefore intractable without additional truncation or approximation. Second, for modern continuous representations such as NeRF (Mildenhall et al., 2021) and InstantNGP (Müller et al., 2022), the mapping from parameters to function values is highly nonlinear. As a result, even if the symmetry constraints are linear, there may not be an explicit or numerical tractable solution of parametrization of the corresponding symmetric parameter subspace.

**Transforming the Output Representation.** Another class of methods enforces symmetry at the value of function. Besides our method, we consider three alternative approach. The first is group averaging, which averages the representation output

over all symmetry-transformed inputs. Although this produces an invariant function in principle, it is infeasible for infinite plane groups, which only yields approximate symmetry. The second is asymmetric-unit extension, which parametrizes the function only on an asymmetric unit and extends it to the whole domain via group actions. This approach is efficient and direct, but it can introduce boundary discontinuities when the asymmetric unit is not a reflection chamber or when adjacent regions are related by non-reflective transformations. The third is basis projection, which projects a general representation onto an invariant function space spanned by symmetry-adapted basis functions. This approach is theoretically valid, but in practice it suffers from severe computational and optimization limitations, as discussed below.

**Complexity of Basis Projection.** Let the target image resolution be $n \times n$. A natural choice of basis for plane-group-symmetric functions is the trigonometric basis. By the Nyquist sampling theorem, the number of recoverable frequency components is on the same order as the number of pixels in the asymmetric unit. Let $\alpha$ denote the fraction of frequency components retained, and let $\beta$ denote the number of asymmetric-unit copies in the full unit cell. Then the number of basis functions is approximately $m \approx \alpha n^2 / \beta$. Projection requires orthogonalizing $m$ basis functions, where each basis function is sampled as an $n^2$-dimensional vector. By applying QR decomposition, this leads to a time complexity of $O(m^2 n^2) = O(n^6)$, and a memory complexity of $O(mn^2) = O(n^4)$. Therefore, both runtime and memory become prohibitive at high resolutions. In contrast, our method introduces only a constant per-pixel time cost from reflection operations and the evaluation of the fixed basis functions $\eta_i$. The overall evaluation cost remains linear in the number of pixels, namely $O(n^2)$, making the method substantially more efficient in both time and memory.

**Effect on Generation Quality.** Basis projection also has an unfavorable effect on generation. For locally supported representations, pixel values in different unit cells are initialized independently. After projection, values related by symmetry are aggregated, which reduces variance and produces an overly smooth initialization. This is problematic for SDS-based optimization, whose performance is sensitive to initialization. In particular, approximately Gaussian-like random initializations tend to provide richer high-frequency content and more diverse optimization trajectories, whereas projected initializations are biased toward low-frequency smooth patterns. As a result, projection-based symmetrization often degrades visual quality. In our pattern-generation experiments in § 6.1, our method consistently achieves better generation quality than the projection-based baseline, as reflected by higher CLIP-A scores across different resolutions.

# B. Higher Reflectional Symmetry

## B.1. Crystallographic Groups and the Integral General Linear Group

We begin with some necessary preliminaries by defining crystallographic groups in $\mathbb{R}^n$.

**Definition B.1.** *A **crystallographic group** $G$ in $\mathbb{R}^n$ is a discrete subgroup of the Euclidean group $E(n)$ that contains $n$ linearly independent translations.*

We also consider an alternative definition attributed to Bieberbach: a crystallographic group $G$ is a discrete subgroup of $E(n)$ such that the quotient space $\mathbb{R}^n/G$ is compact. Bieberbach's First Theorem establishes the equivalence between these two definitions; see (3.2) in Hiller (1986). This definition leads to an important property regarding the finiteness of the point group.

**Proposition B.2** (Hiller (1986), Thm. 3.2.1). *Let $G$ be a crystallographic group. The translation subgroup $T(G) = G \cap \mathbb{R}^n$ is isomorphic to $\mathbb{Z}^n$ and is a normal subgroup of $G$. Furthermore, the point group $P(G) = G/T(G)$ is finite.*

For a crystallographic group, the conjugation action of any group element on a translation element always yields a translation element:

$$(A, \mathbf{t}_2)(O, \mathbf{t}_1)(A, \mathbf{t}_2)^{-1}(\mathbf{x}) = \mathbf{x} + A(\mathbf{t}_1). \tag{11}$$

That is, $(A, \mathbf{t}_2)(O, \mathbf{t}_1)(A, \mathbf{t}_2)^{-1} = (O, A(\mathbf{t}_1))$. Thus, the translation subgroup is a normal subgroup, and the action of the point group $P(G)$ on a translation vector results in another translation vector.

Noting that all translation vectors are generated by a set of fundamental translations, the point group $P(G)$ can be represented by integer matrices under the basis determined by these fundamental translations. Specifically, the point group can be represented by elements of the general linear group over integers, $\mathrm{GL}(n, \mathbb{Z})$. Under this construction, the crystallographic group is represented as a finite subgroup of $\mathrm{GL}(n, \mathbb{Z})$, to be specific, as a group of automorphisms of the lattice defined by the translation subgroup $T(G) \cong \mathbb{Z}^n$.

The choice of different fundamental translations results in matrix representations of the subgroup that differ by conjugation via elements of $\mathrm{GL}(n, \mathbb{Z})$. This allows us to determine equivalence classes of crystallographic groups, known as $\mathbb{Z}$-classes or arithmetic classes. Consequently, the discussion of $\mathbb{Z}$-classes is equivalent to the discussion of conjugacy classes of finite subgroups of $\mathrm{GL}(n, \mathbb{Z})$.

If the representations differ by conjugation via elements of $\mathrm{GL}(n, \mathbb{Q})$, we refer to such equivalence classes as $\mathbb{Q}$-classes. It can be shown that any conjugacy class of finite subgroups of $\mathrm{GL}(n, \mathbb{Q})$ always contains a finite subgroup of $\mathrm{GL}(n, \mathbb{Z})$. Therefore, the discussion of $\mathbb{Q}$-classes is equivalent to the discussion of conjugacy classes of finite subgroups of $\mathrm{GL}(n, \mathbb{Q})$ (see the discussion in Sec. 1.10.1 of Lorenz (2005)).

There exists a crystallographic group that is isomorphic to the semidirect product $P \ltimes T$ corresponding to each arithmetic class. Such crystallographic groups are called symmorphic crystallographic groups. Symmorphic groups contain only pure orthogonal transformations. Otherwise, $G$ is called a non-symmorphic crystallographic group. Euclidean transformations associated with non-symmorphic crystallographic groups involve non-integer translations. That is, the translational component cannot be expressed as an integer linear combination of the fundamental translations. In the case of $n = 2$, these transformations are exclusively glide reflections, formed by the composition of a reflection and a non-integer translation. For $n = 3$, in addition to glide reflections, such transformations may also be screw rotations, which are the composition of a rotation and a non-integer translation.

We now introduce a very important class of symmorphic crystallographic groups: **affine reflection groups**. An affine reflection group is a discrete subgroup of $E(n)$ generated by reflections. These groups possess a classification theorem determined by root systems, thus transforming the classification problem of reflection groups into that of root systems. In subsequent discussions, references to root systems imply reduced or crystallographic root systems (see Sec. 2.8 in Humphreys (1992)).

**Proposition B.3** (Humphreys (1992), Sec. 4.10). *For an affine reflection group $G$ in $\mathbb{R}^n$, there exists a root system $\Phi$ in $\mathbb{R}^n$ such that $G = W \ltimes L(\Phi^\vee)$, where $W$ is the Weyl group of $\Phi$, and $L(\Phi^\vee)$ is the translation subgroup corresponding to the coroot lattice of $\Phi$. Conversely, if for a crystallographic group $G$ there exists a root system $\Phi$ in $\mathbb{R}^n$ such that $G = W(\Phi) \ltimes L(\Phi^\vee)$, then $G$ is an affine reflection group.*

For an affine reflection group, its action on the translation subgroup $T(G) = L(\Phi^\vee)$ is equivalent to the action of the Weyl

group on the coroot lattice. This property facilitates the identification of affine reflection groups within the conjugacy classes of finite subgroups of $\mathrm{GL}(n, \mathbb{Z})$ and $\mathrm{GL}(n, \mathbb{Q})$.

### B.2. Subgroups of Crystallographic Group

In Thm. 3.2, we need to investigate the conditions under which a crystallographic group is conjugate to a subgroup of an affine reflection group. We first address the problem of when a crystallographic group is a subgroup of an affine reflection group.

There are two types of simple subgroups for crystallographic groups: subgroups with a different translation subgroup but the same point group are called $k$-subgroups (klassengleiche subgroups), while subgroups with a different point group but the same translation subgroup are called $t$-subgroups (translationengleiche subgroups). The decomposition of a subgroup chain can be viewed as a combination of $t$-subgroup relations ($t$-steps) and $k$-subgroup relations ($k$-steps), because every maximal subgroup of a crystallographic group is either a $t$-subgroup or a $k$-subgroup (see Theorem 1.4.4.2.3 in Wondratschek & Müller (2011)).

The following lemma suggests that when considering symmetry inclusion problems, it suffices to consider symmorphic crystallographic groups.

**Lemma B.4.** *A non-symmorphic crystallographic group can always be extended to a symmorphic crystallographic group via a $k$-step.*

*Proof.* For a crystallographic group, the order of the point group is finite; thus, the order of every group element is finite. A non-symmorphic crystallographic group contains no pure orthogonal transformations; every element other than the identity involves a translation. Consider an element $(g, \mathbf{t})$, where $g$ is an orthogonal transformation and $\mathbf{t}$ is a translation. After applying the operation $|g|$ times (where $|g|$ denotes the order of $g$), we obtain $(e, |g|\mathbf{t})$, where the resulting translation $|g|\mathbf{t}$ must be contained in the translation subgroup.

Since the order $|g|$ divides the order of the point group $|P|$, for the translation subgroup $T$, all translation components associated with the affine transformations must be contained in the lattice $|P|^{-1}T$. Consequently, the non-symmorphic crystallographic group with translation subgroup $T$ is a subgroup of the symmorphic crystallographic group with translation subgroup $|P|^{-1}T$. This supergroup is isomorphic to the symmorphic crystallographic group belonging to the same arithmetic class as the non-symmorphic group. Therefore, we only need to discuss whether symmorphic crystallographic groups possess affine reflection supergroups. $\square$

**Lemma B.5.** *(Embedding of Symmorphic Crystallographic Group) A symmorphic crystallographic group $G$ is conjugate, via an invertible linear transformation, to a subgroup of another symmorphic crystallographic group $K$ if and only if the $\mathbb{Q}$-classes $(G)$ and $(K)$ associated with the actions on the translation subgroups satisfy $(G) \leq (K)$, i.e., there exists an element of $\mathrm{GL}(n, \mathbb{Q})$ such that the finite subgroup of $\mathrm{GL}(n, \mathbb{Q})$ induced by $G$ is conjugate to a finite subgroup induced by $K$.*

*Proof.* We first fix notation and the meaning of $(\cdot)$ and $\leq$. For a symmorphic crystallographic group $H \leq E(n)$, symmorphicity means that $H$ splits as a semidirect product

$$H \cong P(H) \ltimes T(H), \tag{12}$$

and the conjugation action of $H$ on $T(H)$ factors through $P(H)$, giving a faithful action of $P(H)$ on the lattice $T(H) \cong \mathbb{Z}^n$. After choosing a $\mathbb{Z}$-basis of $T(H)$, this action is represented by a finite subgroup

$$\rho_H\big(P(H)\big) \ \leq \ \mathrm{GL}(n, \mathbb{Z}) \ \subset \ \mathrm{GL}(n, \mathbb{Q}). \tag{13}$$

We define $(H)$ to be the $\mathrm{GL}(n, \mathbb{Q})$-conjugacy class of $\rho_H(P(H))$. Moreover, we write $(G) \leq (K)$ if there exists $M \in \mathrm{GL}(n, \mathbb{Q})$ such that

$$M \, \rho_G\big(P(G)\big) \, M^{-1} \ \leq \ \rho_K\big(P(K)\big) \tag{14}$$

as subgroups of $\mathrm{GL}(n, \mathbb{Q})$.

$(\Rightarrow)$ Assume there exists an invertible linear map $S(\mathbf{x}) = B(\mathbf{x})$ with $B \in \mathrm{GL}(n, \mathbb{R})$ such that

$$SGS^{-1} \ \leq \ K. \tag{15}$$

For any translation $t_{\mathbf{v}}(\mathbf{x}) = \mathbf{x} + \mathbf{v}$ in $T(G)$, $S\, t_{\mathbf{v}}\, S^{-1} = t_{B(\mathbf{v})}$, hence $B\big(T(G)\big) \subseteq T(K)$. Choosing $\mathbb{Z}$-bases of $T(G)$ and $T(K)$, the induced matrix of $B$ is in $\mathrm{GL}(n, \mathbb{Q})$. For any $A \in P(G)$, the linear part of $SAS^{-1}$ equals $BAB^{-1}$, and since $SGS^{-1} \le K$ we have $BAB^{-1} \in P(K)$. In lattice coordinates this implies

$$B\, \rho_G\big(P(G)\big) B^{-1} \;\le\; \rho_K\big(P(K)\big) \quad \text{with } B \in \mathrm{GL}(n, \mathbb{Q}), \tag{16}$$

hence $(G) \le (K)$.

($\Longleftarrow$) Assume $(G) \le (K)$. Choose linear isomorphisms $A_G, A_K \in \mathrm{GL}(n, \mathbb{R})$ such that

$$A_G\big(T(G)\big) = \mathbb{Z}^n, \quad A_K\big(T(K)\big) = \mathbb{Z}^n. \tag{17}$$

Conjugating $G$ and $K$ by $A_G$ and $A_K$, we may assume $T(G) = T(K) = \mathbb{Z}^n$ and still have

$$M\, \rho_G\big(P(G)\big) M^{-1} \;\le\; \rho_K\big(P(K)\big) \tag{18}$$

for some $M \in \mathrm{GL}(n, \mathbb{Q})$. Pick $m \in \mathbb{N}^*$ such that $N := mM \in M_n(\mathbb{Z})$. Since $mI$ commutes with every matrix,

$$N\, \rho_G\big(P(G)\big) N^{-1} = M\, \rho_G\big(P(G)\big) M^{-1} \;\le\; \rho_K\big(P(K)\big). \tag{19}$$

Now any element of $G$ can be written as $(A, \mathbf{v})$ with $A \in P(G)$ and $\mathbf{v} \in \mathbb{Z}^n$, and one checks

$$N\, (A, \mathbf{v})\, N^{-1} = (NAN^{-1}, N(\mathbf{v})). \tag{20}$$

Because $N$ has integer entries, $N(\mathbf{v}) \in \mathbb{Z}^n$, and by the previous inclusion $NAN^{-1} \in P(K)$. Therefore $NGN^{-1} \subset K$ in this normalized setting. Undoing the conjugations yields an invertible linear map

$$S := A_K^{-1} N A_G \in \mathrm{GL}(n, \mathbb{R}) \tag{21}$$

such that $SGS^{-1} \le K$.

This proves the equivalence. $\qquad\qquad\qquad\qquad\qquad\qquad\qquad\qquad\qquad\qquad\qquad\qquad\qquad\square$

In the previous section, we established that for an affine reflection group, the corresponding subgroup of $\mathrm{GL}(n, \mathbb{Z})$ is the representation of the Weyl group acting on the coroot lattice. Consequently, we provide the condition under which Thm. 3.2 holds in $\mathbb{R}^n$. Given the duality between the root lattice and the coroot lattice, we obtain the following result:

**Theorem B.6.** *Every crystallographic group in $\mathbb{R}^n$ is conjugate to a subgroup of some affine reflection group via an invertible linear transformation if and only if every maximal finite subgroup conjugacy class of $\mathrm{GL}(n, \mathbb{Q})$ contains the action of a Weyl group on the root lattice.*

## B.3. Existence of Higher Reflectional Symmetry for $n = 2, 3$

We interpret the action on the dual space as the usual contragredient action. Let $T = T(G) \cong \mathbb{Z}^n$ be the translation subgroup of a crystallographic group $G$. The point group $P(G)$ induces a rational representation $\rho$. On the dual (reciprocal) space we consider the induced action $\rho^*(A)$. In a chosen $\mathbb{Z}$-basis of $T$ and its dual basis of $V^*$, if $\rho(A)$ is represented by $M_A \in \mathrm{GL}(n, \mathbb{Z})$, then $\rho^*(A)$ is represented by $M_A^{-T}$.

**Lemma B.7.** *The action of a crystallographic group on its translation subgroup and the induced action on the dual space determine the same $\mathbb{Q}$-class.*

*Proof.* What we need to show is simply that the two matrix groups

$$\rho(P(G)) = \{M_A :\ A \in P(G)\} \tag{22}$$

and

$$\rho^*(P(G)) = \{M_A^{-T} :\ A \in P(G)\} \tag{23}$$

are conjugate inside $\mathrm{GL}(n, \mathbb{Q})$.

The standard way to relate a representation to its dual is to exhibit a nondegenerate bilinear form that is invariant under the group. Because $P(G)$ is finite, we can always *average* any bilinear form to make it invariant. Concretely, start with an arbitrary symmetric positive definite bilinear form on $V$, represented by a matrix $S_0 \in M_n(\mathbb{Q})$ (for instance $S_0 = I$). We then average it over the group:

$$S := \frac{1}{|P(G)|} \sum_{A \in P(G)} M_A^T S_0 M_A \in M_n(\mathbb{Q}). \tag{24}$$

This matrix $S$ is still symmetric and positive definite, hence invertible over $\mathbb{Q}$. More importantly, by construction it is $P(G)$-*invariant*: multiplying the average on the right by a fixed element $B \in P(G)$ only permutes the summands (since $A \mapsto AB$ is a bijection of the finite group $P(G)$). Therefore the average does not change, and we get

$$M_B^T S M_B = S \qquad \forall B \in P(G). \tag{25}$$

This invariance identity is exactly what we need. Indeed, rewriting it gives

$$S M_B = M_B^{-T} S \qquad \implies \qquad S^{-1} M_B S = M_B^{-T}, \tag{26}$$

for every $B \in P(G)$. Hence $S$ conjugates the original action matrices $\{M_B\}$ to the dual action matrices $\{M_B^{-T}\}$, showing that $\rho$ and $\rho^*$ are conjugate in $\mathrm{GL}(n, \mathbb{Q})$ and therefore define the same $\mathbb{Q}$-class.

Finally, note that the same "permutation of summands" argument implies the averaging operator is idempotent: once a form is $P(G)$-invariant, averaging it again does nothing. In particular, the matrix $S$ above satisfies

$$\frac{1}{|P(G)|} \sum_{A \in P(G)} M_A^T S M_A = S, \tag{27}$$

which is another way to state that the averaging construction produces a fixed point in the space of bilinear forms. $\square$

**Lemma B.8** ((Lorenz, 2005), Prop. 1.1). *With the exception of type $C_4$, considering the action on the root lattice, the automorphism group $\mathrm{Aut}(\Phi)$ of any irreducible root system constitutes a maximal finite subgroup of $\mathrm{GL}(n, \mathbb{Z})$.*

**Theorem B.9.** *For $n = 2, 3$, Thm. B.6 holds. Equivalently, every planar group ($n = 2$) and every space group ($n = 3$) is conjugate to a subgroup of some affine reflection group via an invertible linear transformation.*

*Proof.* We follow the same strategy in both dimensions: reduce maximal finite subgroups of $\mathrm{GL}(n, \mathbb{Q})$ to maximal finite subgroups of $\mathrm{GL}(n, \mathbb{Z})$, and then identify the latter with Weyl actions on root/weight lattices.

First, we recall the consistency between subgroup conjugacy in $\mathrm{GL}(n, \mathbb{Q})$ and $\mathbb{Q}$-classes arising from $\mathrm{GL}(n, \mathbb{Z})$. Let $H \leq \mathrm{GL}(n, \mathbb{Q})$ be finite. Then $H$ preserves a full lattice in $\mathbb{Q}^n$, hence $H$ is $\mathrm{GL}(n, \mathbb{Q})$-conjugate to a finite subgroup of $\mathrm{GL}(n, \mathbb{Z})$. Moreover, $\mathbb{Z}$-conjugacy implies $\mathbb{Q}$-conjugacy, so $\mathbb{Q}$-classes in $\mathrm{GL}(n, \mathbb{Z})$ are coarser than $\mathbb{Z}$-classes and can be obtained by merging $\mathbb{Z}$-classes.

Now let $H \leq \mathrm{GL}(n, \mathbb{Q})$ be a maximal finite subgroup. Conjugate $H$ into $\mathrm{GL}(n, \mathbb{Z})$ and denote the resulting finite subgroup by $H' \leq \mathrm{GL}(n, \mathbb{Z})$. Choose a maximal finite subgroup $M \leq \mathrm{GL}(n, \mathbb{Z})$ containing $H'$. Then $H \leq g^{-1} M g$ for some $g \in \mathrm{GL}(n, \mathbb{Q})$, and since $H$ is maximal finite in $\mathrm{GL}(n, \mathbb{Q})$, we must have $H = g^{-1} M g$. Therefore every maximal finite subgroup of $\mathrm{GL}(n, \mathbb{Q})$ is $\mathrm{GL}(n, \mathbb{Q})$-conjugate to a maximal finite subgroup of $\mathrm{GL}(n, \mathbb{Z})$. Equivalently, maximal $\mathbb{Q}$-conjugacy classes in $\mathrm{GL}(n, \mathbb{Q})$ are obtained from maximal $\mathbb{Z}$-conjugacy classes in $\mathrm{GL}(n, \mathbb{Z})$ by merging those that become $\mathbb{Q}$-conjugate.

It remains to verify, for $n = 2, 3$, that each maximal finite subgroup of $\mathrm{GL}(n, \mathbb{Z})$ corresponds to a Weyl group acting on a root lattice (or, at worst, on a dual lattice, which will be reduced to the root-lattice case using Lem. B.7).

For $n = 2$, we consider maximal finite subgroups of $\mathrm{GL}(2, \mathbb{Z})$. By Table 1.1 in Lorenz (2005), there are exactly two maximal finite subgroups up to $\mathbb{Z}$-conjugacy. By the lemma that $\mathrm{Aut}(A_2)$, $\mathrm{Aut}(B_2)$, and $\mathrm{Aut}(G_2)$ occur as maximal candidates, and by the well-known identifications

$$\mathrm{Aut}(B_2) \cong W(B_2), \quad \mathrm{Aut}(G_2) \cong W(G_2), \tag{28}$$

we may take representatives $\mathrm{Aut}(B_2)$ and $\mathrm{Aut}(G_2)$, whose orders are distinct and hence account for the two maximal $\mathbb{Z}$-classes. These are precisely the lattice symmetry groups corresponding to the planar groups $p4mm$ and $p6mm$ acting on

the translation lattice. Consequently, every maximal finite subgroup of $\mathrm{GL}(2,\mathbb{Z})$ contains a Weyl group acting on a root lattice, and hence the same holds for every maximal finite subgroup of $\mathrm{GL}(2,\mathbb{Q})$.

For $n = 3$, we consider maximal finite subgroups of $\mathrm{GL}(3,\mathbb{Z})$. By Tahara (1971), the order of a finite subgroup of $\mathrm{GL}(3,\mathbb{Z})$ is at most $48$. The maximal order $48$ is realized by the full octahedral point group $O_h$, which occurs for the three cubic Bravais lattices and corresponds to the symmorphic space groups

$$Pm\bar{3}m, \quad Im\bar{3}m, \quad Fm\bar{3}m. \tag{29}$$

By Kim (2001), with respect to the dual-space representation, $Pm\bar{3}m$ is self-dual, while $Im\bar{3}m$ and $Fm\bar{3}m$ form a dual pair. Among these three, $Pm\bar{3}m$ and $Im\bar{3}m$ are reflection groups.

On the other hand, by our lemma that $\mathrm{Aut}(A_3)$, $\mathrm{Aut}(B_3)$, and $\mathrm{Aut}(C_3)$ are maximal (in the relevant integral setting), and by the tabulated orders in Bourbaki (2002), all three have order $48$. Moreover,

$$\mathrm{Aut}(B_3) \cong W(B_3), \quad \mathrm{Aut}(C_3) \cong W(C_3), \tag{30}$$

and $\mathrm{Aut}(B_3)$ is self-dual. Interpreting the three cubic lattices as root/dual-root (equivalently weight/coweight) lattices, we obtain: the action for $Pm\bar{3}m$ corresponds to the $C_3$-coroot lattice action, i.e. the $B_3$ root-lattice action; the action for $Im\bar{3}m$ corresponds to the $B_3$-coroot lattice action, i.e. the $C_3$ root-lattice action. Since the latter is not self-dual, the remaining dual partner $Fm\bar{3}m$ corresponds to the dual lattice realization (weight lattice, equivalently the action on the dual of the coroot lattice).

By Table 1.1 in Lorenz (2005), $\mathrm{GL}(3,\mathbb{Z})$ has four maximal finite subgroups up to $\mathbb{Z}$-conjugacy, so one maximal class remains to be identified beyond the cubic $O_h$-classes. Since the point group $D_{6h}$ is also a maximal crystallographic point group, Lem. B.5 implies that it determines a maximal $\mathbb{Q}$-class (it cannot embed into a larger crystallographic point group), hence it must arise from a maximal $\mathbb{Z}$-class as well. Therefore every maximal $\mathbb{Z}$-class in dimension 3 corresponds either to a Weyl root-lattice action or to a Weyl weight-lattice (dual-lattice) action. Finally, by Lem. B.7, the action on the translation lattice and the action on the dual space lie in the same $\mathbb{Q}$-class, so at the $\mathbb{Q}$-class level these dual-lattice realizations are accounted for by the corresponding root-lattice Weyl actions. It follows that every maximal finite subgroup of $\mathrm{GL}(3,\mathbb{Q})$ contains a Weyl group acting on a root lattice.

Combining the cases $n = 2$ and $n = 3$ with the reduction at the beginning of the proof, Thm. B.6 follows for $n = 2, 3$. $\quad\square$

### B.4. Non-existence of Higher Reflectional Symmetry for $n > 3, n \neq 7, 8$

We now introduce a simple group-theoretic trick that will be used repeatedly in our counterexample constructions. Roughly speaking, if a finite group contains a large non-abelian simple subgroup (such as an alternating group), then it cannot embed into a semidirect product whose non-permutation part is abelian. Here $\mathscr{S}_k$ denotes the symmetric group on $k$ letters, i.e. the group of all permutations of $\{1, \ldots, k\}$.

**Lemma B.10.** *Let $n > 3$ and let $H$ be an abelian group. For any $k \leq n$, the alternating group $\mathscr{A}_{n+1}$ is not isomorphic to a subgroup of the semidirect product $\mathscr{S}_k \ltimes H$. Equivalently, there is no injective homomorphism $\mathscr{A}_{n+1} \hookrightarrow \mathscr{S}_k \ltimes H$.*

*Proof.* Assume for contradiction that there exists an injective homomorphism $i : \mathscr{A}_{n+1} \hookrightarrow \mathscr{S}_k \ltimes H$. Let

$$\pi : \mathscr{S}_k \ltimes H \to \mathscr{S}_k, \qquad \pi(\sigma, h) = \sigma \tag{31}$$

be the canonical projection. Then $\ker \pi = \{e\} \times H$ is abelian.

Consider the composition $\pi \circ i : \mathscr{A}_{n+1} \to \mathscr{S}_k$ and denote $K := \ker(\pi \circ i)$. Since $K$ is a normal subgroup of the simple group $\mathscr{A}_{n+1}$, we have $K = \{e\}$ or $K = \mathscr{A}_{n+1}$.

If $K = \mathscr{A}_{n+1}$, then $\pi(i(\mathscr{A}_{n+1})) = \{e\}$, hence $i(\mathscr{A}_{n+1}) \subset \ker \pi \cong H$, which is abelian. This contradicts the fact that $\mathscr{A}_{n+1}$ is non-abelian.

Therefore $K = \{e\}$ and $\pi \circ i$ is injective. It follows that

$$|\mathscr{A}_{n+1}| \leq |\mathscr{S}_k|. \tag{32}$$

But for $k \leq n$,

$$|\mathscr{A}_{n+1}| = \frac{(n+1)!}{2} > n! \geq k! = |\mathscr{S}_k|, \tag{33}$$

a contradiction. Hence no such embedding exists. $\qquad\square$

**Theorem B.11.** *For $n > 3$ and $n \neq 7, 8$, there exists a crystallographic group which is not isomorphic to any subgroup of an affine reflection group.*

*Proof.* Let $\Phi = A_n$ and consider the symmorphic crystallographic group

$$G := \mathrm{Aut}(\Phi) \ltimes L(\Phi^\vee), \tag{34}$$

where $L(\Phi^\vee)$ is the coroot lattice. Assume for contradiction that $G$ is isomorphic to a subgroup of some affine reflection group. Equivalently, there exist a root system $\Psi$ of rank $n$ and an injective homomorphism

$$\varphi : G \hookrightarrow W_a(\Psi) = W(\Psi) \ltimes L(\Psi^\vee). \tag{35}$$

Let $T(G)$ denote the translation subgroup of $G$ (so $T(G) \cong L(\Phi^\vee)$). It is the unique maximal abelian normal subgroup of $G$ and hence characteristic. Similarly, $L(\Psi^\vee)$ is the unique maximal abelian normal subgroup of $W_a(\Psi)$. Therefore

$$\varphi\big(T(G)\big) \subset L(\Psi^\vee), \tag{36}$$

and $\varphi$ induces an injective homomorphism on the finite quotients

$$\bar{\varphi} : G/T(G) \hookrightarrow W_a(\Psi)/L(\Psi^\vee) \cong W(\Psi). \tag{37}$$

Since $G/T(G) \cong \mathrm{Aut}(A_n)$, we obtain an embedding

$$\mathrm{Aut}(A_n) \hookrightarrow W(\Psi). \tag{38}$$

In particular, the alternating group $\mathscr{A}_{n+1}$ (a normal subgroup of $\mathrm{Aut}(A_n) \cong \mathscr{S}_{n+1} \rtimes \mathbb{Z}_2$) embeds into $W(\Psi)$:

$$\mathscr{A}_{n+1} \hookrightarrow W(\Psi). \tag{39}$$

For $n > 3$, $\mathscr{A}_{n+1}$ is a non-abelian simple group.

Write $\Psi = \Psi_1 \sqcup \cdots \sqcup \Psi_m$ as the decomposition into irreducible components, so that

$$W(\Psi) = W(\Psi_1) \times \cdots \times W(\Psi_m). \tag{40}$$

Let $\pi_i : W(\Psi) \to W(\Psi_i)$ be the canonical projections. Since $\mathscr{A}_{n+1}$ is simple, for each $i$ the kernel of $\pi_i|_{\mathscr{A}_{n+1}}$ is either trivial or all of $\mathscr{A}_{n+1}$. If all projections were trivial then the embedding would be trivial, impossible. Hence for some $i$ we have an injective map

$$\mathscr{A}_{n+1} \hookrightarrow W(\Psi_i). \tag{41}$$

Let $k := \mathrm{rank}(\Psi_i) \leq n$. We now rule out all possibilities for $\Psi_i$ when $n \neq 7, 8$.

If $\Psi_i$ is of type $B_k$ or $C_k$, then

$$W(\Psi_i) \cong \mathscr{S}_k \ltimes (\mathbb{Z}_2)^k, \tag{42}$$

and if $\Psi_i$ is of type $D_k$, then

$$W(\Psi_i) \cong \mathscr{S}_k \ltimes (\mathbb{Z}_2)^{k-1}, \tag{43}$$

where the second factor is abelian. By Lem. B.10 (with $k \leq n$), $\mathscr{A}_{n+1}$ cannot be a subgroup of $W(\Psi_i)$, a contradiction.

If $\Psi_i$ is of type $A_k$, then $W(\Psi_i) \cong \mathscr{S}_{k+1}$. If $k \leq n-1$, then $k+1 \leq n$ and

$$|\mathscr{A}_{n+1}| = \frac{(n+1)!}{2} > n! \geq (k+1)! = |\mathscr{S}_{k+1}|, \tag{44}$$

so $\mathscr{A}_{n+1} \not\hookrightarrow \mathscr{S}_{k+1}$, a contradiction. If $k = n$, then $\Psi = A_n$ and $W(\Psi) = W(A_n) \cong \mathscr{S}_{n+1}$; but

$$|\mathrm{Aut}(A_n)| = 2\,(n+1)! \;>\; (n+1)! = |W(A_n)|, \tag{45}$$

so $\mathrm{Aut}(A_n) \not\hookrightarrow W(A_n)$, contradicting $\mathrm{Aut}(A_n) \hookrightarrow W(\Psi)$.

It remains to exclude the exceptional types. If $\Psi_i = F_4$, then $|W(F_4)| = 2^7 \cdot 3^2$, so $5 \nmid |W(F_4)|$. Since $n > 3$ implies $n + 1 \geq 5$, we have $5 \mid |\mathscr{A}_{n+1}|$, hence $\mathscr{A}_{n+1} \not\hookrightarrow W(F_4)$. If $\Psi_i = E_6$, then $|W(E_6)| = 2^7 \cdot 3^4 \cdot 5$, so $7 \nmid |W(E_6)|$. This component can only occur when $n \geq 6$, in which case $n + 1 \geq 7$ and thus $7 \mid |\mathscr{A}_{n+1}|$, hence $\mathscr{A}_{n+1} \not\hookrightarrow W(E_6)$. Finally, if $\Psi_i$ is of type $E_7$ or $E_8$, then $W(\Psi_i)$ admits a faithful real representation of dimension 7 or 8, respectively. Thus an embedding $\mathscr{A}_{n+1} \hookrightarrow W(\Psi_i)$ would yield a faithful representation of $\mathscr{A}_{n+1}$ in dimension at most 8. For $n > 8$ this is impossible since the minimal dimension of a nontrivial (hence faithful) representation of $\mathscr{A}_{n+1}$ is $n$ (the deleted permutation representation). Therefore the only remaining possible cases are $n = 7, 8$, which are excluded by assumption.

This contradiction shows that no injective $\varphi$ can exist. Hence $G$ is not isomorphic to any subgroup of an affine reflection group. $\qquad\square$

### B.5. Non-existence of Higher Reflectional Symmetry for $n = 7, 8$

For $n \neq 7, 8$, the maximality of $\mathrm{Aut}(A_n)$ (as a finite subgroup of $\mathrm{GL}(n, \mathbb{Q})$) implies that, once the low-dimensional coincidences for $n = 2, 3$ no longer occur, the statement "every crystallographic group is conjugate to a subgroup of some affine reflection group via an invertible linear transformation" fails. The cases $n = 7, 8$ are exceptional: one has

$$\mathrm{Aut}(A_7) \subset W(E_7), \qquad \mathrm{Aut}(A_8) \subset W(E_8), \tag{46}$$

see Prop. II.8 in Plesken (1991), and in these dimensions $W(E_7)$ and $W(E_8)$ are maximal finite subgroups. Therefore a counterexample for $n = 7, 8$ requires a more delicate construction.

**Lemma B.12** (Plesken (1991), Prop. II.6). *Let $G_i \leq \mathrm{GL}(n_i, \mathbb{Q})$ be maximal finite irreducible subgroups for $i = 1, \ldots, k$ with $k > 1$, and let*

$$G := \mathrm{Diag}(G_1, \ldots, G_k) \leq \mathrm{GL}(n_1 + \cdots + n_k, \mathbb{Q}). \tag{47}$$

*Assume that, for $p = 2$, at most one of the $G_i$ has the trivial Brauer character 1 occurring in the restriction of its natural character to the 2-regular classes. If no two of the $G_i$ are primitively related, then $G$ is maximal finite in $\mathrm{GL}(n_1 + \cdots + n_k, \mathbb{Q})$.*

**Theorem B.13.** *For $n = 7, 8$, there exists a crystallographic group which is not isomorphic to any subgroup of an affine reflection group.*

*Proof.* We only need to show that there exists a maximal $\mathbb{Q}$-class in $\mathrm{GL}(n, \mathbb{Q})$ which is not represented by any Weyl group. Indeed, if a crystallographic group were isomorphic to a subgroup of an affine reflection group, then its point group (viewed as a finite subgroup of $\mathrm{GL}(n, \mathbb{Q})$ via the action on the translation lattice) would embed into the linear part of that affine reflection group, hence into a Weyl group. Therefore a maximal finite subgroup of $\mathrm{GL}(n, \mathbb{Q})$ which is not of Weyl type yields the desired counterexample.

For $n = 8$, there exists a maximal finite irreducible subgroup of $\mathrm{GL}(8, \mathbb{Q})$ which is not a Weyl group, namely $\mathrm{Aut}(4A_2)$ appearing in Thm. 11.14 of Lorenz (2005). It satisfies

$$\mathrm{Aut}(4A_2) \cong \mathscr{S}_4 \ltimes \mathrm{Aut}(A_2)^4. \tag{48}$$

It is straightforward to verify that $\mathrm{Aut}(4A_2)$ is not isomorphic to any Weyl group acting on $\mathbb{R}^8$. Hence there exists a maximal $\mathbb{Q}$-class in $\mathrm{GL}(8, \mathbb{Q})$ which is not of Weyl type.

According to Plesken (1991), for $n = 7$, all maximal finite irreducible subgroups of $\mathrm{GL}(7, \mathbb{Q})$ are $W(B_7)$ and $W(E_7)$, hence are Weyl groups. Thus we construct a maximal finite subgroup via a block-diagonal (reducible) embedding. We claim that

$$\mathrm{Aut}(A_3 \times A_4) = \mathrm{Diag}\big(\mathrm{Aut}(A_3), \mathrm{Aut}(A_4)\big) \leq \mathrm{GL}(7, \mathbb{Q}) \tag{49}$$

is maximal finite and is not Weyl group. By Lem. B.12, it suffices to verify:

(i) $\mathrm{Aut}(A_3)$ and $\mathrm{Aut}(A_4)$ are not primitively related, i.e. there does not exist an irreducible $H \subset \mathrm{GL}(m, \mathbb{Q})$ such that

$$\mathrm{Aut}(A_3) \subset \mathscr{S}_{k_1} \ltimes H, \qquad \mathrm{Aut}(A_4) \subset \mathscr{S}_{k_2} \ltimes H, \qquad k_1 m = 3, \; k_2 m = 4. \tag{50}$$

This is immediate since $\gcd(3,4) = 1$, hence $m = 1$, $k_1 = 3$, $k_2 = 4$, and then $H \leq \mathrm{GL}(1, \mathbb{Q})$ is finite, so $H \subset \{\pm I\}$. But $|\mathrm{Aut}(A_3)|$ and $|\mathrm{Aut}(A_4)|$ are strictly larger than 2, so the above containments cannot hold.

(ii) For $p = 2$, at most one of $\mathrm{Aut}(A_3)$ and $\mathrm{Aut}(A_4)$ has the trivial Brauer character 1 occurring in the restriction of its natural character to the 2-regular classes. It suffices to show that $\mathrm{Aut}(A_3)$ don't have the trivial Brauer character 1 occurring in the restriction of its natural character to the 2-regular classes

Consider the natural action of $\mathrm{Aut}(A_3)$ on the $A_3$ root lattice

$$L = \left\{ \mathbf{x} = (x_1, x_2, x_3, x_4) \in \mathbb{Z}^4 \ \Big| \ \sum_{i=1}^{4} x_i = 0 \right\}, \tag{51}$$

which is a rank-3 lattice embedded as a hyperplane in $\mathbb{R}^4$. Reducing modulo 2, this is equivalent to considering the $\mathbb{F}_2$-module

$$L/2L = \left\{ \mathbf{x} \in \mathbb{F}_2^4 \ \Big| \ \sum_{i=1}^{4} x_i = 0 \right\}. \tag{52}$$

The fixed-point space detects the occurrence of the trivial Brauer constituent in characteristic 2. In particular, it is enough to show that

$$(L/2L)^{\mathrm{Aut}(A_3)} \neq 0. \tag{53}$$

Since the characteristic is 2, the vector $(1,1,1,1)$ satisfies $1+1+1+1 = 0$ in $\mathbb{F}_2$, hence lies in $L/2L$. Moreover, $(1,1,1,1)$ is fixed by all coordinate permutations, and any global sign change is trivial over $\mathbb{F}_2$. Therefore $(1,1,1,1) \in (L/2L)^{\mathrm{Aut}(A_3)}$, and thus $(L/2L)^{\mathrm{Aut}(A_3)} \neq 0$.

Consequently, both hypotheses of Lem. B.12 are satisfied, and $\mathrm{Aut}(A_3 \times A_4)$ is maximal finite in $\mathrm{GL}\,7, \mathbb{Q})$. Finally, it is straightforward to check that $\mathrm{Aut}(A_3 \times A_4)$ is not isomorphic to any Weyl group acting on $\mathbb{R}^7$. Hence there exists a maximal $\mathbb{Q}$-class in $\mathrm{GL}(7, \mathbb{Q})$ which is not of Weyl type.

Combining the cases $n = 7$ and $n = 8$ completes the proof. $\qquad\square$

# C. Hironaka Decomposition

## C.1. Crystallographic Groups and Multiplicative Invariant Theory

We work in $\mathbb{R}^n$. Let $G$ be a crystallographic group with translation subgroup $T(G) \cong \mathbb{Z}^n$, and let $L \subset \mathbb{R}^n$ be the minimal translation lattice of $G$. Let $L^* := \{\boldsymbol{\alpha} \in \mathbb{R}^n : \langle \boldsymbol{\alpha}, \boldsymbol{\ell} \rangle \in \mathbb{Z} \text{ for all } \boldsymbol{\ell} \in L\}$ be the dual lattice. Choose generators $\boldsymbol{\alpha}_1, \ldots, \boldsymbol{\alpha}_n$ of $L^*$ and introduce the basic exponentials

$$e_j^{\pm}(\mathbf{x}) = \exp(\pm 2\pi i \langle \boldsymbol{\alpha}_j, \mathbf{x} \rangle), \quad j = 1, \ldots, n. \tag{54}$$

Define the trigonometric coordinates

$$c_j := \frac{e_j^+ + e_j^-}{2}, \qquad s_j := \frac{e_j^+ - e_j^-}{2i}, \quad j = 1, \ldots, n, \tag{55}$$

so that $(c_j, s_j)$ satisfy $c_j^2 + s_j^2 = 1$.

It suffices to study real polynomials in $c_1, s_1, \ldots, c_n, s_n$. Set

$$P_{\text{torus}} := \mathbb{R}[c_1, s_1, \ldots, c_n, s_n] \cong \mathbb{R}[x_1, y_1, \ldots, x_n, y_n] \big/ \langle x_j^2 + y_j^2 - 1 \rangle_{j=1}^n, \tag{56}$$

where $\mathbb{R}[x_1, y_1, \ldots, x_n, y_n]$ is the polynomial ring. After complexification, we obtain

$$\mathbb{C} \otimes_{\mathbb{R}} P_{\text{torus}} \cong \mathbb{C}[e_1^{\pm}, \ldots, e_n^{\pm}] \cong \mathbb{C}[x_1, y_1, \ldots, x_n, y_n] \big/ \langle x_j y_j - 1 \rangle_{j=1}^n, \tag{57}$$

which is the Laurent ring in $n$ variables.

**Lemma C.1.** *For a root system $\Phi$, the action of the affine reflection group $W_a$ on the reciprocal-space algebra $\mathbb{C} \otimes_{\mathbb{R}} P_{\text{torus}}$ is equivalent to the action of $W_a$ on the weight lattice $\Lambda(\Phi)$.*

*Proof.* Write the affine reflection group as the semidirect product

$$W_a = W \ltimes L(\Phi^{\vee}), \tag{58}$$

where $L(\Phi^{\vee})$ is the coroot lattice (the translation part). Let

$$\psi_\lambda(\mathbf{x}) := e^{2\pi i \langle \lambda, \mathbf{x} \rangle} \qquad (\lambda \in \Lambda(\Phi)). \tag{59}$$

Then $\mathbb{C}[\Lambda(\Phi)]$ is spanned by $\{\psi_\lambda\}$, and (after choosing a $\mathbb{Z}$-basis)

$$\mathbb{C} \otimes_{\mathbb{R}} P_{\text{torus}} \cong \mathbb{C}[e_1^{\pm}, e_2^{\pm}] \cong \mathbb{C}[\Lambda(\Phi)]. \tag{60}$$

For $(w, \mu) \in W \ltimes L(\Phi^{\vee})$, the induced action on characters is

$$\begin{aligned}
(w, \mu)\psi_\lambda(\mathbf{x}) &= \psi_{(w,\mu)^{-1}\lambda}(\mathbf{x}) \\
&= \psi_{w^{-1}(\lambda - \mu)}(\mathbf{x}) \\
&= e^{2\pi i \langle \lambda, w^{-1}\mathbf{x} \rangle} e^{-2\pi i \langle \lambda, w^{-1}\mu \rangle}.
\end{aligned} \tag{61}$$

Using $W$-invariance of the bilinear form, $\langle \lambda, w^{-1}\mathbf{x} \rangle = \langle w\lambda, \mathbf{x} \rangle$. Moreover $w^{-1}\mu \in L(\Phi^{\vee})$, hence the second factor is a phase. Therefore the $W_a$-action on $\mathbb{C} \otimes_{\mathbb{R}} P_{\text{torus}}$ matches (up to this canonical phase) the usual affine action of $W_a$ on $\Lambda(\Phi)$. $\square$

Consequently, the action of $W_a$ on $\mathbb{C} \otimes_{\mathbb{R}} P_{\text{torus}}$ is equivalent to its action on $\mathbb{C}[\Lambda(\Phi)]$. For a symmorphic crystallographic group $G = P(G) \ltimes T(G)$, the reciprocal-space action may be regarded as an action on a lattice algebra $\mathbb{C}[L]$, i.e. via a subgroup of $\text{Aut}(L)$. For a nonsymmorphic group, the presence of non-integral translations produces a phase on coefficients. Thus one is naturally led to consider twisted multiplicative actions (see Sec. 2.8 in Lorenz (2005)).

**Proposition C.2** ([Lorenz](2005), Cor. 7.1.2). *Let $L$ be a lattice and let $G$ be a finite subgroup of $\mathrm{GL}(L)$ acting on $\mathbb{C}[L]$. Then $\mathbb{C}[L]^G$ is a polynomial algebra if and only if there exists a root system $\Phi$ such that the action of $G$ on $L$ is equivalent to the action of the Weyl group $W$ on the weight lattice $\Lambda(\Phi)$.*

The sufficiency of the reflection group condition was established by [Bourbaki](2002), while the necessity was proven by [Farkas](1986). This theorem constitutes a generalization of the Chevalley–Shephard–Todd theorem from classical invariant theory to multiplicative invariant theory. [Lorenz](2005) discusses results for general $R[L]^G$, where $R$ is a regular commutative ring; since this covers the cases of $\mathbb{Z}$ and $\mathbb{C}$, the conclusion holds.

**Lemma C.3.** *Let $A$ be a finitely generated $\mathbb{R}$-algebra. Then*

$$\mathbb{C} \otimes_{\mathbb{R}} A \cong \mathbb{C}[x_1, \ldots, x_n] \iff A \cong \mathbb{R}[x_1, \ldots, x_n]. \tag{62}$$

*Proof.* Assume $\mathbb{C} \otimes_{\mathbb{R}} A \cong \mathbb{C}[x_1, \ldots, x_n]$. Fix a $\mathbb{C}$-algebra isomorphism

$$\sigma : \ \mathbb{C} \otimes_{\mathbb{R}} A \longrightarrow \mathbb{C}[x_1, \ldots, x_n]. \tag{63}$$

Let

$$\Gamma = \mathrm{Gal}(\mathbb{C}/\mathbb{R}), \tag{64}$$

the Galois group generated by complex conjugation. Since $\sigma$ is $\mathbb{C}$-linear, it is $\Gamma$-equivariant, and hence restricts to an $\mathbb{R}$-algebra isomorphism

$$\sigma : \ (\mathbb{C} \otimes_{\mathbb{R}} A)^{\Gamma} \longrightarrow \mathbb{C}[x_1, \ldots, x_n]^{\Gamma}. \tag{65}$$

Because $\Gamma$ acts trivially on $A$,

$$(\mathbb{C} \otimes_{\mathbb{R}} A)^{\Gamma} = A, \qquad \mathbb{C}[x_1, \ldots, x_n]^{\Gamma} = \mathbb{C}^{\Gamma}[x_1, \ldots, x_n]. \tag{66}$$

By Galois theory (see Chap. 16 in [Artin](2011)), $\mathbb{C}/\mathbb{R}$ is a Galois extension and $\mathbb{C}^{\Gamma} = \mathbb{R}$. Hence $A \cong \mathbb{R}[x_1, \ldots, x_n]$. $\square$

The corollary to Prop. C.2 follows from Lem. C.3.

**Corollary C.4.** *For a symmorphic crystallographic group $G$, the invariant algebra $P_{\mathrm{torus}}^G$ is isomorphic to a polynomial algebra if and only if $G$ is a reflection group.*

## C.2. Cohen-Macaulay Property and Hironaka Decomposition

**Proposition C.5** ([Lorenz](2005), Thm. 8.4.2). *Let $R$ be an algebra over $\mathbb{R}$. If $R$ is an integral domain and is a finitely generated module over a real polynomial subalgebra, then $R$ satisfies the Cohen-Macaulay property if and only if $R$ is a finitely generated free module over that polynomial subalgebra.*

**Lemma C.6.** $P_{\mathrm{torus}}^G$ *satisfies the Cohen-Macaulay property.*

*Proof.* For $\mathbb{C} \otimes_{\mathbb{R}} P_{\mathrm{torus}}$ symmetric with respect to a symmorphic crystallographic group, the proof is provided in Sec. 8.9 of [Lorenz](2005). We now proceed to prove the properties for real polynomials. The proof strategy employed below, utilizing the Reynolds averaging operator, is a standard technique in classical invariant theory, see [Stanley](1979).

Let $G$ and $K$ be arbitrary crystallographic groups with translation lattice $T$.

(i) $P_{\mathrm{torus}}^G$ is integral over $P_{\mathrm{torus}}^K$, and there exists a Reynolds operator from $P_{\mathrm{torus}}^G$ to $P_{\mathrm{torus}}^K$.

(ii) $\mathbb{C} \otimes_{\mathbb{R}} P_{\mathrm{torus}}^G$ is integral over $P_{\mathrm{torus}}^G$, and there exists a Reynolds operator from $\mathbb{C} \otimes_{\mathbb{R}} P_{\mathrm{torus}}^G$ to $P_{\mathrm{torus}}^G$.

Consider the proof of (i). For $f \in P_{\mathrm{torus}}^G$, consider the polynomial map:

$$P_f(t) = \prod_{g \in K/G} (t - g(f)) \tag{67}$$

This function is monic; the coefficients of $P_f(t)$ belong to $P_{\mathrm{torus}}^K$, and $P_f(f) = 0$. Thus, it is integral. We construct the map:

$$\rho_{G \to K} : P_{\mathrm{torus}}^G \to P_{\mathrm{torus}}^K, \quad \rho_{G \to K}(f) = [K:G]^{-1} \sum_{g \in K/G} g(f) \tag{68}$$

It is straightforward to verify that the above operator is $P^K_{\text{torus}}$-linear ($P^K_{\text{torus}}$-module homomorphism) and satisfies $\rho_{G \to K}|_{P^K_{\text{torus}}} = \text{Id}_{P^K_{\text{torus}}}$. Thus, it is a Reynolds operator.

The proof for (ii) is similar. For $f \in \mathbb{C} \otimes_{\mathbb{R}} P^G_{\text{torus}}$, we construct the polynomial map:

$$P_f(t) = (t - f)(t - f^\dagger) \tag{69}$$

The Reynolds operator is given by:

$$\rho : \mathbb{C} \otimes_{\mathbb{R}} P^G_{\text{torus}} \to P^G_{\text{torus}}, \quad \rho(f) = (f + f^\dagger)/2 \tag{70}$$

By Thm. 6.4.5 of Bruns & Herzog (1998) states that if (i) holds and $P^G_{\text{torus}}$ is Cohen-Macaulay, then $P^K_{\text{torus}}$ is Cohen-Macaulay. Similarly, if (ii) holds and $\mathbb{C} \otimes_{\mathbb{R}} P^G_{\text{torus}}$ is Cohen-Macaulay, then $P^G_{\text{torus}}$ is Cohen-Macaulay.

Consider the case $G = T$. From Section 8.9 of (Lorenz, 2005), we deduce that $\mathbb{C} \otimes_{\mathbb{R}} P_{\text{torus}}$ satisfies the Cohen-Macaulay property. Therefore, $P_{\text{torus}}$ satisfies the Cohen-Macaulay property. Consequently, for any crystallographic group $G$ with translation lattice $T$, $P^G_{\text{torus}}$ satisfies the Cohen-Macaulay property. $\square$

**Lemma C.7** (Noether Normalization, Mumford (2004), Sec. 1.1). *Let $R$ be a finitely generated commutative integral domain over a field $k$. Assume that the transcendence degree of $R$ over $k$ is $n$, i.e., the maximal number of algebraically independent elements in $R$ over $k$ is $n$. Then there exist $n$ algebraically independent elements $\theta_1, \ldots, \theta_n \in R$ such that $R$ is a finitely generated module over the subring $k[\theta_1, \ldots, \theta_n]$.*

**Lemma C.8.** *The invariant ring $P^G_{\text{torus}}$ has at most $n$ algebraically independent elements.*

*Proof.* Using the monic-polynomial construction in Lem. C.6, we see that the extension

$$P^G_{\text{torus}} \subset \mathbb{C} \otimes_{\mathbb{R}} P_{\text{torus}} \tag{71}$$

is integral. Hence it suffices to show that the Laurent polynomial ring

$$A := \mathbb{C} \otimes_{\mathbb{R}} P_{\text{torus}} \cong \mathbb{C}[x_1^{\pm 1}, \ldots, x_n^{\pm 1}] \tag{72}$$

contains at most $n$ algebraically independent elements (over $k := \mathbb{C}$).

Take arbitrary $m$ elements $f_1, \ldots, f_m \in A$ and assume that they are algebraically independent over $k$. Since $A$ is an integral domain, it embeds into its field of fractions

$$K := \text{Frac}(A). \tag{73}$$

If $f_1, \ldots, f_m$ were algebraically dependent in $K$, then there would exist a nonzero polynomial

$$P(T_1, \ldots, T_m) \in k[T_1, \ldots, T_m] \setminus \{0\} \tag{74}$$

such that $P(f_1, \ldots, f_m) = 0$ in $K$. But the same identity would then hold in the subring $A \subset K$, contradicting the assumed algebraic independence in $A$. Therefore $f_1, \ldots, f_m$ remain algebraically independent in $K$.

On the other hand, the Laurent ring is a localization of the polynomial ring:

$$A \cong k[x_1, \ldots, x_n]_{x_1 \cdots x_n}. \tag{75}$$

Hence the two rings have the same field of fractions:

$$K = \text{Frac}(A) = \text{Frac}(k[x_1, \ldots, x_n]) = k(x_1, \ldots, x_n), \tag{76}$$

so $K$ is a purely transcendental extension generated by $x_1, \ldots, x_n$.

Now we use the basic property of transcendence degree: for any field extension $F/k$, the transcendence degree $\text{trdeg}_k F$ equals the maximal cardinality of an algebraically independent subset of $F$ over $k$. In the purely transcendental field

$$k(x_1, \ldots, x_n), \tag{77}$$

the set $\{x_1, \ldots, x_n\}$ is algebraically independent and generates the whole field, hence

$$\operatorname{trdeg}_k K = n. \tag{78}$$

Consequently, every algebraically independent subset of $K$ has cardinality at most $n$, and in particular

$$m \leq \operatorname{trdeg}_k K = n. \tag{79}$$

Therefore $A$ contains at most $n$ algebraically independent elements, and since $P^G_{\text{torus}} \subset A$ is an integral extension, the same bound holds for $P^G_{\text{torus}}$. $\qquad \square$

**Theorem C.9** (Hironaka Decomposition). *For $P^G_{\text{torus}}$, there always exist $n$ primary invariants $\theta_1, \ldots, \theta_n$ and a set of secondary invariants $g_1, \ldots, g_r$, such that any invariant polynomial $f$ can be uniquely decomposed as*

$$f(\mathbf{x}) = \sum_{i=1}^{r} p_i(\theta_1, \ldots, \theta_n) \, g_i(\mathbf{x}), \tag{80}$$

*where each $p_i$ is a polynomial in $k[\theta_1, \ldots, \theta_n]$. In particular, for affine reflection groups, we may take $r = 1$ and $g_1(\mathbf{x}) = 1$.*

*Proof.* Since $P^G_{\text{torus}}$ admits at most $n$ algebraically independent elements, we apply Noether normalization to choose algebraically independent invariants $\theta_1, \ldots, \theta_n$ together with a corresponding set of module generators $g_1, \ldots, g_r$. This realizes $P^G_{\text{torus}}$ as a finitely generated module over the polynomial ring $k[\theta_1, \ldots, \theta_n]$.

By Lem. C.6, the ring $P^G_{\text{torus}}$ is Cohen–Macaulay. Therefore, by Prop. C.5, it is in fact a finite *free* module over $k[\theta_1, \ldots, \theta_n]$. The freeness implies that the above decomposition is unique.

For affine reflection groups, the polynomial structure of the invariant ring is given by Cor. C.4, which yields $r = 1$ and $g_1(\mathbf{x}) = 1$. $\qquad \square$

## C.3. Choice of Primary Invariants

**Proposition C.10.** *Let $K$ and $G$ be crystallographic groups with $G \leq K$. Let $T$ be the translation lattice of $G$, and let $R$ be the corresponding ring associated with $T$. Assume that $R^K$ is isomorphic to a real polynomial ring. Then $R^G$ is a finitely generated free $R^K$-module, and the rank of this module equals the index $[K : G]$.*

*Proof.* Since $R^G$ is integral over $R^K$, Prop. 5.1 in Atiyah & Macdonald (1994) implies that $R^G$ is a finitely generated $R^K$-module. Moreover, $R^K \cong \mathbb{R}[\theta_1, \ldots, \theta_n]$ is a polynomial ring and $R^G$ is Cohen–Macaulay; hence $R^G$ is in fact a finitely generated *free* $R^K$-module.

To determine the rank, we appeal to Galois theory, see Thm. 3.7.1. of Derksen & Kemper (2002). Consider the field extensions of fraction fields $\operatorname{frac}(R^G) \subset \operatorname{frac}(R)$ and $\operatorname{frac}(R^K) \subset \operatorname{frac}(R)$. By faithfulness of the action, both extensions are Galois, with Galois groups $G$ and $K$, respectively; hence their degrees equal the group orders. Using the tower formula,

$$[\operatorname{frac}(R) : \operatorname{frac}(R^K)] = [\operatorname{frac}(R) : \operatorname{frac}(R^G)] \, [\operatorname{frac}(R^G) : \operatorname{frac}(R^K)], \tag{81}$$

we obtain

$$[\operatorname{frac}(R^G) : \operatorname{frac}(R^K)] = \frac{[\operatorname{frac}(R) : \operatorname{frac}(R^K)]}{[\operatorname{frac}(R) : \operatorname{frac}(R^G)]} = [K : G]. \tag{82}$$

A basis of $R^G$ as an $R^K$-module induces a spanning set of $\operatorname{frac}(R^G)$ as a vector space over $\operatorname{frac}(R^K)$. Since the dimension of this vector space equals the extension degree $[\operatorname{frac}(R^G) : \operatorname{frac}(R^K)]$, any generating set has cardinality at least $[K : G]$. When the module is free, the basis elements are $\operatorname{frac}(R^K)$-linearly independent, hence form a vector-space basis; therefore the number of basis elements is exactly $[K : G]$. $\qquad \square$

If $G$ admits an affine reflection group $W_a$ as a supergroup, then by Cor. C.4 the invariant ring of $W_a$ has a polynomial structure. Consequently we obtain the following corollary.

**Corollary C.11.** *If $G$ admits an affine reflection group $W_a$ as a supergroup, then one can choose $\theta_1, \theta_2$ to be the primary invariants of $P^{W_a}_{\text{torus}}$. In this case, the number of basis elements $r$ is exactly the index $[W_a : G]$.*

## C.4. Approximation of Continuous Functions

**Theorem C.12.** *If $G$ admits an affine reflection group $W_a$ as a supergroup, then for every $f \in C_G(\mathbb{R}^2)$ and every $\epsilon > 0$, there exist functions $h_1, \ldots, h_r \in C_{W_a}(\mathbb{R}^2)$ such that*

$$\int_\Omega \left| f(x,y) - \sum_{i=1}^r h_i(x,y)\, g_i(x,y) \right|^2 \mathrm{d}\Omega < \epsilon, \tag{83}$$

*where $r = [W_a : G]$ and $\Omega$ is the minimal cell.*

*Proof.* Since $P_{\text{torus}}$ is dense in $C_{T(G)}(\mathbb{R}^n)$, it suffices to control the approximation error on a single period. Applying the Reynolds averaging operator, we deduce that $R^{W_a}$ is dense in $C_{W_a}(\mathbb{R}^n)$. Therefore, the $C_{W_a}(\mathbb{R}^n)$-module generated by $g_1, \ldots, g_r$ is also dense, which yields the desired approximation. $\qquad\square$

## D. Secondary Invariant Construction

### D.1. Secondary Invariant for Non-Symmorphic Crystallographic Groups

The results for symmorphic crystallographic groups are presented in Kim (2001). Our primary discussion focuses on the treatment of non-symmorphic crystallographic groups. We adopt the approach from Kim (2001). In the subsequent section, we will introduce a method based on Fourier expansions, for which a calculation example is provided.

For the case of $n = 2$, the non-symmorphic crystallographic groups are: $pg$ (a $k$-subgroup of $p2mm$ in the $pm$ arithmetic class); $p2mg$ and $p2gg$ (both $k$-subgroups of $p2mm$ in the $p2mm$ arithmetic class); and $p4gm$ (a $k$-subgroup of $p4mm$ in the $p4mm$ arithmetic class). In our discussion, it is necessary to halve the period of the translation subgroup along the direction corresponding to the glide plane.

The calculation of primary and secondary invariants for a non-symmorphic crystallographic group $G$ proceeds in two steps. The first step considers the reflection group acting as its $k$-supergroup $K$, and the second step considers its $t$-subgroup $H$ of index 2, which acts as a symmorphic crystallographic group. This decomposition is always feasible. Consider the mapping

$$\pi : G \to \{\pm 1\}, \qquad \pi((\mathbf{A}, \mathbf{t})) = \det(\mathbf{A}). \tag{84}$$

By the fundamental theorem of homomorphisms, $H = \ker(\pi)$ is the required subgroup of index 2 (since $G/H \cong \{\pm 1\}$). The subgroup $H$ constitutes the orientation-preserving part of $G$. For $n = 2$, transformations involving non-integer translations are exclusively glide reflections (whereas screw rotations also appear for $n = 3$). Since a glide reflection is the composition of a reflection and a non-integer translation, removing these leaves $H$ as a symmorphic crystallographic group. Since subgroups of index 2 are always normal, we have the group chain

$$H \lhd G \subset K. \tag{85}$$

**Step 1:** Express $P_{\text{torus}}^H$ as a finitely generated free module over $P_{\text{torus}}^K$:

$$P_{\text{torus}}^H = P_{\text{torus}}^K(1, \eta_2, \ldots, \eta_m), \qquad m = [K : H]. \tag{86}$$

**Step 2:** Consider the $G$-invariant elements $P_{\text{torus}}^G$ inside $P_{\text{torus}}^H$. Since $H$ is a normal subgroup of $G$, the action of $G$ stabilizes $P_{\text{torus}}^H$, and consequently stabilizes the vector space

$$V = \text{frac}\big(P_{\text{torus}}^H\big)/\text{frac}\big(P_{\text{torus}}^K\big). \tag{87}$$

At this stage, $V$ becomes a $\mathbb{Z}_2$-representation. The space $V$ decomposes into a direct sum of trivial representations and sign representations. The trivial component $V^G$ corresponds to a basis of $P_{\text{torus}}^G$. In the cases considered here, this decomposition is natural, since the non-orientation-preserving element $g$ (which together with $H$ generates $G$) acts on the linear span of the $\eta_i$ either as the identity or as a reflection.

**Step 3:** Choose elements $(1, \eta_2', \ldots, \eta_r')$ from $V^G$, thereby expressing $P_{\text{torus}}^G$ as a finitely generated free module over $P_{\text{torus}}^K$:

$$P_{\text{torus}}^G = P_{\text{torus}}^K(1, \eta_2', \ldots, \eta_r'). \tag{88}$$

Four explicit examples are given below. In each example, $K$ is chosen as a reflection group (a $k$-supergroup of $G$), and $H = \ker(\pi)$ is the index-2 symmorphic subgroup. We first express $P_{\text{torus}}^H$ as a free $P_{\text{torus}}^K$-module by listing a basis of secondary invariants $(1, \eta_2, \ldots)$, and then impose the action of a fixed $g \in G \setminus H$ on this basis. The $g$-fixed part $V^G$ determines a reduced basis $(1, \eta_2', \ldots)$ and hence $P_{\text{torus}}^G$.

$pg$. Choose the reflection $k$-supergroup $K = p2mm'$, where the glide direction forces the translation period in the $y$-direction to be halved. Thus

$$P_{\text{torus}}^K = p[c_1, c(2y)] = p[c_1, c_2^2]. \tag{89}$$

The index-2 symmorphic subgroup is $H = p1$, and

$$P_{\text{torus}}^H = p[c_1, c_2](1, s_1)(1, s_2) = p[c_1, c_2^2](1, c_2)(1, s_1)(1, s_2). \tag{90}$$

Hence we may take

$$(1, \eta_2, \eta_3, \eta_4) = (1, \ c_2, \ s_1, \ s_2). \tag{91}$$

Let $g \in G \setminus H$ be the glide $g(x, y) = (x, y + \frac{1}{2})$, acting by

$$(c_1, s_1, c_2, s_2) \mapsto (c_1, -s_1, -c_2, -s_2). \tag{92}$$

Thus

$$(1, \eta_2', \eta_3', \eta_4') = (1, \ s_1 c_2, \ s_1 s_2, \ c_2 s_2), \tag{93}$$

and

$$P_{\text{torus}}^G = P_{\text{torus}}^{pg} = p[c_1, c_2^2]\big(1, \ s_1 c_2, \ s_1 s_2, \ c_2 s_2\big). \tag{94}$$

$p2mg$. Choose the reflection $k$-supergroup $K = p2mm'$, where the glide plane forces the translation period in the $x$-direction to be halved. Thus

$$P_{\text{torus}}^K = p[c(2x), c_2] = p[c_1^2, c_2]. \tag{95}$$

The index-2 symmorphic subgroup is $H = p2$, and

$$P_{\text{torus}}^H = p[c_1, c_2](1, s_1 s_2) = p[c_1^2, c_2](1, c_1)(1, s_1 s_2). \tag{96}$$

Let $m \in G \setminus H$ be the reflection at $x = \frac{1}{4}$, $m(x, y) = (\frac{1}{2} - x, y)$, acting by

$$(c_1, s_1, c_2, s_2) \mapsto (-c_1, s_1, c_2, s_2). \tag{97}$$

Among the secondary invariants of $P_{\text{torus}}^H$, the sign-invariant ones are $1$ and $s_1 s_2$, yielding $[p2mg : p2mm'] = 2$ secondary invariants. Hence

$$P_{\text{torus}}^G = P_{\text{torus}}^{p2mg} = p[c_1^2, c_2](1, s_1 s_2). \tag{98}$$

$p2gg$. Choose the reflection $k$-supergroup $K = p2mm'$, where the glide planes force the translation periods in both the $x$- and $y$-directions to be halved. Thus

$$P_{\text{torus}}^K = p[c(2x), c(2y)] = p[c_1^2, c_2^2]. \tag{99}$$

The index-4 symmorphic subgroup is $H = p2$, and

$$P_{\text{torus}}^H = p[c_1, c_2](1, s_1 s_2) = p[c_1^2, c_2^2](1, c_1)(1, c_2)(1, s_1 s_2). \tag{100}$$

Let $g \in G \setminus H$ be the glide $g(x, y) = (\frac{1}{2} - x, y + \frac{1}{2})$, acting by

$$(c_1, s_1, c_2, s_2) \mapsto (-c_1, s_1, -c_2, -s_2). \tag{101}$$

The sign-invariant secondary invariants are products involving $0$ or $2$ sign changes, giving $[p2gg : p2mm'] = 4$ secondary invariants. Therefore

$$P_{\text{torus}}^G = P_{\text{torus}}^{p2gg} = p[c_1^2, c_2^2]\big(1, \ c_1 c_2, \ c_1 s_1 s_2, \ s_1 c_2 s_2\big). \tag{102}$$

$p4gm$. Choose the reflection $k$-supergroup $K = p4mm'$, where the glide planes force the translation periods in both the $x$- and $y$-directions to be halved. Thus

$$P_{\text{torus}}^K = p[c(2x) + c(2y), c(2x)c(2y)] = p[c_1^2 + c_2^2, c_1^2 c_2^2]. \tag{103}$$

The index-4 symmorphic subgroup is $H = p4$, and

$$P_{\text{torus}}^H = p[c_1 + c_2, c_1 c_2]\big(1, (c_1 - c_2)s_1 s_2\big) = p[c_1^2 + c_2^2, c_1^2 c_2^2](1, c_1 + c_2)(1, c_1 c_2)(1, (c_1 - c_2)s_1 s_2). \tag{104}$$

Let $m \in G \setminus H$ be the reflection across $y = -x + \frac{1}{2}$, $m(x, y) = (\frac{1}{2} - y, \frac{1}{2} - x)$, acting by

$$(c_1, s_1, c_2, s_2) \mapsto (-c_2, s_2, -c_1, s_1). \tag{105}$$

Expanding and retaining the reflection-invariant secondary invariants yields $[p4gm : p4mm'] = 4$ secondary invariants. Hence

$$P_{\text{torus}}^G = P_{\text{torus}}^{p4gm} = p[c_1^2 + c_2^2, c_1^2 c_2^2](1, c_1 c_2)(1, (c_1 - c_2)s_1 s_2) \tag{106}$$

$$= p[c_1^2 + c_2^2, c_1^2 c_2^2]\big(1, \ c_1 c_2, \ (c_1 - c_2)s_1 s_2, \ (c_1 - c_2)c_1 c_2 s_1 s_2\big). \tag{107}$$

## D.2. Adjusted Coordinate Conventions

The basis (choice of primitive cell) used in Kim (2001) differs from the ITA convention, most notably for the groups $cm$ and $c2mm$. In Kim (2001) the authors adopt a rhombic primitive cell, in which the cell admits the mirror line $y = x$. In contrast, the ITA typically uses the conventional rectangular cell. Therefore, to convert the formulas to the ITA setting, we apply the change of basis

$$\mathbf{e}_1 \mapsto \tfrac{1}{2}(\mathbf{e}_2 - \mathbf{e}_1), \qquad \mathbf{e}_2 \mapsto \tfrac{1}{2}(\mathbf{e}_2 + \mathbf{e}_1). \tag{108}$$

Equivalently, in coordinates this corresponds to

$$\begin{bmatrix} -1/2 & 1/2 \\ 1/2 & 1/2 \end{bmatrix}^{-1} \begin{bmatrix} x \\ y \end{bmatrix} = 2 \begin{bmatrix} 1 & -1 \\ -1 & -1 \end{bmatrix} \begin{bmatrix} x \\ y \end{bmatrix}. \tag{109}$$

Moreover, one may alternatively expand the invariants of $cm$ and $c2mm$ inside the $p2mm'$-framework; the underlying idea is the same as in the previous examples.

$cm$.   Consider the $k$-supergroup $p2mm'$ of $cm$. Since the glide periods in both the $x$- and $y$-directions are halved, its invariant ring is

$$P_{\text{torus}}^{p2mm'} = p[c(2x), c(2y)] = p[c_1^2, c_2^2]. \tag{110}$$

Let $pg$ be the index-2 symmorphic subgroup of $cm$. For bookkeeping under the ITA convention, we write $pm_+$ and $cm_+$ for the plane groups whose mirror line is parallel to the $y$-axis, and $pm_-$ and $cm_-$ for those whose mirror line is parallel to the $x$-axis.

Consider the symmorphic subgroup $pm_+$ of $cm_+$. Rewriting its invariant ring as a module over $P_{\text{torus}}^{p2mm'}$, we obtain

$$P_{\text{torus}}^{pm_+} = p[c_1, c_2](1, s_1) = p[c_1^2, c_2^2](1, c_1)(1, c_2)(1, s_2). \tag{111}$$

The group $cm_+$ is obtained from $pm_+$ by adjoining the glide at $x = \tfrac{1}{4}$,

$$g(x, y) = \left( \tfrac{1}{2} - x, \ y + \tfrac{1}{2} \right), \tag{112}$$

which acts by

$$(c_1, s_1, c_2, s_2) \mapsto (-c_1, \ s_1, \ -c_2, \ -s_2). \tag{113}$$

Hence the $cm_+$-invariant second invariants are exactly the products involving 0 or 2 sign changes, giving $[cm_+ : p2mm'] = 4$ second invariants and therefore

$$P_{\text{torus}}^{cm_+} = p[c_1^2, c_2^2]\left(1, \ c_1 c_2, \ c_1 s_2, \ c_2 s_2\right). \tag{114}$$

Swapping $x$ and $y$ in the standard coordinates yields

$$P_{\text{torus}}^{cm_-} = p[c_1^2, c_2^2]\left(1, \ c_1 c_2, \ c_2 s_1, \ c_1 s_1\right). \tag{115}$$

$c2mm$.   The group $c2mm$ simultaneously contains the symmetries of both $cm_+$ and $cm_-$, and their invariant rings coincide at the level of first invariants. Hence the second invariants for $c2mm$ are obtained by taking the intersection of the two second-invariant sets. This yields $[c2mm : p2mm'] = 4$ and

$$P_{\text{torus}}^{c2mm} = p[c_1^2, c_2^2]\left(1, \ c_1 c_2\right). \tag{116}$$

## D.3. Alternative Computational Approaches

Another way to compute invariant bases is to expand a two-dimensional periodic function into Fourier series. When the symmetry is enlarged, the associated function spaces spanned by Fourier modes are transformed and merged, producing representations of the plane group. In our construction, the nontrivial isotypic components are discarded, and the remaining part corresponds to the trivial representation of the plane group. Below we explain how to construct a basis for this trivial subspace by enforcing the relations among Fourier coefficients induced by symmetry operations.

Let $T = T(G)$ be the translation subgroup and consider the quotient $G/T$. For each coset representative $(\mathbf{P}_i \mid \mathbf{t}_i) \in G/T$, write the Fourier expansion of a density function as

$$\rho(\mathbf{r}) = \frac{1}{V} \sum_{\mathbf{k}} F(\mathbf{k}) \psi_{\mathbf{k}}(\mathbf{r}) = \frac{1}{V} \sum_{i=1}^{|G/T|} \sum_{\mathbf{k}'} \frac{1}{|G_{\mathbf{k}'}|} F(\mathbf{P}_i^T \mathbf{k}') \, \psi_{\mathbf{P}_i \mathbf{k}'}(\mathbf{r}), \tag{117}$$

where $V$ is a normalization factor depending on the lattice parameters, $\mathbf{k}'$ runs over orbit representatives, and $G_{\mathbf{k}'}$ denotes the stabilizer of $\mathbf{k}'$ in $G/T$. Following the notation in Shmueli (2010), for each orbit representative $\mathbf{k}'$ one introduces the orbit-symmetrized combination

$$C_{\mathbf{k}'}(\mathbf{r}) = \sum_{i=1}^{|G/T|} \psi_{\mathbf{k}'}(\mathbf{t}_i) \, \psi_{\mathbf{P}_i^T \mathbf{k}'}(\mathbf{r}) = A_{\mathbf{k}'}(\mathbf{r}) + i \, B_{\mathbf{k}'}(\mathbf{r}). \tag{118}$$

Tab. A1.4.3.1 in Shmueli (2010) gives explicit analytic expressions of this form for all 17 plane groups (and, more generally, for all 230 space groups).

In that table, the integers $h, k$ are the coordinates of the wave vector

$$\mathbf{k} = h \, \mathbf{a}^* + k \, \mathbf{b}^*, \tag{119}$$

whereas $x, y$ are the fractional coordinates of the position vector

$$\mathbf{r} = x \, \mathbf{a} + y \, \mathbf{b}. \tag{120}$$

For compactness, Tab. A1.4.3.1 omits the reflection conditions, hence for wave vectors forced to vanish by reflection constraints, the corresponding basis functions cannot be read off directly from that table. In practice, several plane groups share the same analytic form of basis functions and differ only by these reflection conditions. This information is organized in the ITA as the general reflection conditions, displayed in the right-lower corner of each plane-group entry (see Sec. 2.2 in Aroyo (2016)). Even when the reflection conditions are satisfied, some terms of $A_{\mathbf{k}}$ or $B_{\mathbf{k}}$ may still vanish. Conceptually, this happens when the orbits of $\mathbf{k}$ and $-\mathbf{k}$ merge and the trivial isotypic component becomes 1-dimensional, in which case $C_{\mathbf{k}}$ has only a real part. A particularly transparent case is when the point group already contains reflections: then $B_{\mathbf{k}}$ vanishes identically. Related phenomena are also summarized in Shmueli (2010) in the tables of structure factors.

We also note a minor convention issue: the hexagonal basis used in Shmueli (2010) differs from the ITA convention. For instance, for $p6$ the invariant polynomial basis $C(hki)$ is invariant under the change of variables $(x, y) \mapsto (-y, x - y)$; this corresponds to a 6-fold rotation only when the basis angle is $\pi/3$, rather than $2\pi/3$.

We illustrate the Fourier approach on a plane group $p4mm$ with a square lattice, point group $D_4$ (hence also Laue group $D_4$). We choose wave-vector representatives with $h \geq k \geq 0$, and impose the reflection conditions

$$h0 : \; h = 2n, \qquad 0k : \; k = 2n. \tag{121}$$

In this case, the real part $A_{\mathbf{k}}(\mathbf{r})$ can be written in the following piecewise form:

$$A_{\mathbf{k}}(\mathbf{r}) = \begin{cases} P(cc), & \begin{cases} h = 2n_1 \geq k = 2n_2 \geq 0, \\ h = 2n_1 + 1 \geq k = 2n_2 + 1 > 0, \end{cases} \\ M(ss), & \begin{cases} h = 2n_1 > k = 2n_2 + 1 > 0, \\ h = 2n_1 + 1 > k = 2n_2 > 0. \end{cases} \end{cases} \tag{122}$$

The basis functions have the explicit forms

$$P(cc) = c(hx)c(ky) + c(kx)c(hy) = p_{cc}(c_1, c_2), \quad h \equiv k \pmod{2}, \tag{123}$$

$$M(ss) = s(hx)s(ky) - s(kx)s(hy) = p_{ss}(c_1, c_2) s_1 s_2, \quad h \equiv k+1 \pmod{2}. \tag{124}$$

Using the multiple-angle identities, these expressions can be rewritten in terms of Chebyshev polynomials. Since the Chebyshev polynomials $T_n$ and $U_n$ are even/odd according to the parity of $n$, the polynomial

$$p_{cc} = T_h T_k + T_k T_h \tag{125}$$

is a symmetric polynomial in two variables and is invariant under reflections. Hence its first basic invariants are

$$c_1^2 + c_2^2, \quad c_1 c_2, \tag{126}$$

and, because we are in a reflection-group setting, there is no nontrivial second invariant. On the other hand,

$$p_{ss} = U_{h-1} U_{k-1} - U_{k-1} U_{h-1} \tag{127}$$

is an antisymmetric polynomial and changes sign under reflections. Therefore it has the same first basic invariants $c_1^2 + c_2^2$ and $c_1 c_2$, but admits a nontrivial second basic invariant $c_1 - c_2$. Consequently, the expansion takes the form

$$p_1(c_1^2 + c_2^2, \, c_1 c_2) \; + \; p_2(c_1^2 + c_2^2, \, c_1 c_2) \, (c_1 - c_2) \, s_1 s_2. \tag{128}$$

Finally, if we perform a $k$-lifting of $p4gm$ to $p4mm'$ (so that the periods in both $x$ and $y$ are halved), then the angles must be replaced by $\theta_1^{(2)}$ and $\theta_2^{(2)}$. In this setting, the set of second invariants enlarges: besides $(c_1 - c_2)s_1 s_2$ we also obtain $c_1 c_2$ and $(c_1 - c_2)c_1 s_1 c_2 s_2$. Thus the number of second invariants is $[p4gm : p4mm'] = 4$, and the nontrivial ones can be taken as

$$c_1 c_2, \quad (c_1 - c_2)s_1 s_2, \quad (c_1 - c_2)c_1 s_1 c_2 s_2. \tag{129}$$

## E. Symmetric Connectivity Criterion

In what follows, we fix the translation subgroup of the planar group $p1$ by the two primitive vectors

$$\mathbf{e}_1 = (1,0), \qquad \mathbf{e}_2 = (0,1), \tag{130}$$

and all occurrences of connected mean path-connected.

Let the (half-open) unit cell and the $2 \times 2$ supercell be

$$B = [0,1) \times [0,1), \qquad A = [0,2) \times [0,2). \tag{131}$$

Let

$$\pi : \mathbb{R}^2 \to \mathbb{T}^2 := \mathbb{R}^2/\mathbb{Z}^2 \tag{132}$$

be the quotient (covering) map.

For a path-connected subset $C \subset \mathbb{T}^2$, the full preimage $\pi^{-1}(C) \subset \mathbb{R}^2$ may have several path-connected components. For any such component $\widetilde{C}$, define its stabilizer

$$H(\widetilde{C}) = \{g \in \mathbb{Z}^2 \mid \widetilde{C} + g = \widetilde{C}\}. \tag{133}$$

**Lemma E.1.** *Let $C \subset \mathbb{T}^2$ be path-connected. If $\widetilde{C}_1, \widetilde{C}_2$ are any two path-connected components of $\pi^{-1}(C)$, then there exists $k \in \mathbb{Z}^2$ such that $\widetilde{C}_2 = \widetilde{C}_1 + k$, and moreover*

$$H(\widetilde{C}_1) = H(\widetilde{C}_2). \tag{134}$$

*Hence $H(\widetilde{C})$ depends only on $C$, and we may write it as $H(C)$.*

*Proof.* Pick $x_1 \in \widetilde{C}_1$ and set $\bar{x} := \pi(x_1) \in C$. Since $\pi(\widetilde{C}_2) = C$, there exists $x_2 \in \widetilde{C}_2$ with $\pi(x_2) = \bar{x}$. Then $x_2 - x_1 \in \mathbb{Z}^2$; write $k := x_2 - x_1$ so that $x_2 = x_1 + k$.

The translation $T_k(x) = x + k$ satisfies $\pi \circ T_k = \pi$, hence $T_k(\pi^{-1}(C)) = \pi^{-1}(C)$. Therefore $T_k(\widetilde{C}_1)$ is path-connected, contained in $\pi^{-1}(C)$, and contains $x_2$. By maximality of the component $\widetilde{C}_2$ containing $x_2$, we get $T_k(\widetilde{C}_1) \subset \widetilde{C}_2$. Applying the same argument to $T_{-k}$ yields the reverse inclusion, hence $\widetilde{C}_2 = T_k(\widetilde{C}_1) = \widetilde{C}_1 + k$.

Finally, for any $g \in \mathbb{Z}^2$,

$$(\widetilde{C}_1 + k) + g = \widetilde{C}_1 + k \iff \widetilde{C}_1 + g = \widetilde{C}_1, \tag{135}$$

so $g \in H(\widetilde{C}_1)$ iff $g \in H(\widetilde{C}_2)$. Thus $H(\widetilde{C}_1) = H(\widetilde{C}_2)$. $\square$

**Lemma E.2.** *Let $E, F \subset \mathbb{R}^2$ be nonempty, $\mathbb{Z}^2$-invariant subsets, i.e.*

$$E + (m,n) = E, \qquad F + (m,n) = F, \qquad \forall (m,n) \in \mathbb{Z}^2. \tag{136}$$

*If both $E$ and $F$ are path-connected, then $E \cap F \neq \varnothing$.*

*Proof.* Let $\pi : \mathbb{R}^2 \to \mathbb{T}^2$ be the quotient map. We first show $\pi(E) \cap \pi(F) \neq \varnothing$.

Pick $p \in E$. Since $p + \mathbf{e}_1 \in E$ and $E$ is path-connected, there exists a path

$$\gamma_x : [0,1] \to E, \qquad \gamma_x(0) = p, \ \gamma_x(1) = p + \mathbf{e}_1. \tag{137}$$

Set $\alpha := \pi \circ \gamma_x$, which is a loop in $\mathbb{T}^2$ based at $\pi(p)$. Likewise, pick $q \in F$. Since $q + \mathbf{e}_2 \in F$ and $F$ is path-connected, there exists a path

$$\gamma_y : [0,1] \to F, \qquad \gamma_y(0) = q, \ \gamma_y(1) = q + \mathbf{e}_2, \tag{138}$$

and set $\beta := \pi \circ \gamma_y$, a loop in $\mathbb{T}^2$ based at $\pi(q)$.

Lift $\alpha$ to $\widetilde{\alpha}$ with $\widetilde{\alpha}(0) = p$. By uniqueness of path lifting, $\widetilde{\alpha} = \gamma_x$, hence $\widetilde{\alpha}(1) - \widetilde{\alpha}(0) = \mathbf{e}_1$. Similarly, the lift of $\beta$ starting at $q$ satisfies $\widetilde{\beta}(1) - \widetilde{\beta}(0) = \mathbf{e}_2$. Thus, under the standard identification $\pi_1(\mathbb{T}^2) \cong \mathbb{Z}^2$, the loops $\alpha, \beta$ represent the classes $(1,0)$ and $(0,1)$.

By intersection theory on surfaces (e.g. the mod-2 intersection number; see Sec. 2.4 of Guillemin & Pollack (1974)), the mod-2 intersection number of two loops depends only on their homotopy (equivalently homology) classes, and the two coordinate generators $(1,0)$ and $(0,1)$ have mod-2 intersection equal to 1. Hence $\alpha$ and $\beta$ cannot be disjoint, so $\alpha([0,1]) \cap \beta([0,1]) \neq \varnothing$. Consequently,

$$\pi(E) \cap \pi(F) \neq \varnothing. \tag{139}$$

Now take $\bar{z} \in \pi(E) \cap \pi(F)$. Choose $e \in E$ and $f \in F$ with $\pi(e) = \pi(f) = \bar{z}$. Then $e - f \in \mathbb{Z}^2$. Let $t := e - f \in \mathbb{Z}^2$, so $f + t = e$. Since $F$ is $\mathbb{Z}^2$-invariant, $f + t \in F$, hence $e \in E \cap F$. Therefore $E \cap F \neq \varnothing$. □

**Theorem E.3.** *Let $S \subset \mathbb{R}^2$ be $\mathbb{Z}^2$-invariant. Fix $A = [0,2) \times [0,2)$, $B = [0,1) \times [0,1)$ and $\Gamma \subset B$ as above. Assume that every path-connected component of $S \cap A$ intersects $\Gamma$. Then $S$ is path-connected.*

*Proof.* Let $C := \pi(S) \subset \mathbb{T}^2$, and write the decomposition into path-connected components $C = \bigsqcup_{j \in J} C_j$. Since $S$ is $\mathbb{Z}^2$-invariant, one has the identity

$$S = \pi^{-1}(C), \tag{140}$$

because if $\pi(x) \in C$ then $\pi(x) = \pi(y)$ for some $y \in S$, hence $x - y \in \mathbb{Z}^2$ and thus $x \in S$.

Fix $j \in J$, and choose any lift component $\widetilde{C}_j \subset \pi^{-1}(C_j)$. By translating $\widetilde{C}_j$ by some integer vector (which yields another lift component of the same $C_j$ by Lemma E.1), we may assume

$$\widetilde{C}_j \cap B \neq \varnothing. \tag{141}$$

Consider $\mathbf{e}_1$. If $\mathbf{e}_1 \notin H(\widetilde{C}_j)$, then $\widetilde{C}_j$ and $\widetilde{C}_j + \mathbf{e}_1$ are two distinct (hence disjoint) path-connected components of $\pi^{-1}(C_j) \subset S$. Since $\widetilde{C}_j \cap B \neq \varnothing$, we have $(\widetilde{C}_j + \mathbf{e}_1) \cap (B + \mathbf{e}_1) \neq \varnothing$, so $(\widetilde{C}_j + \mathbf{e}_1) \cap A \neq \varnothing$. Let $D$ be any path-connected component of $(\widetilde{C}_j + \mathbf{e}_1) \cap A$. Then $D$ is a path-connected component of $S \cap A$ (it cannot connect inside $A$ to any other lift component because distinct lift components are disjoint). Moreover, $D \subset B + \mathbf{e}_1$, hence $D \cap \Gamma = \varnothing$ for all choices of $\Gamma \in \{\Gamma_1, \Gamma_2, \Gamma_1 \cup \Gamma_2\}$, since $\Gamma \subset \partial B$ and $B + \mathbf{e}_1$ is disjoint from $\partial B$. This contradicts the hypothesis that every path-connected component of $S \cap A$ intersects $\Gamma$. Therefore $\mathbf{e}_1 \in H(\widetilde{C}_j)$.

The same argument with $\mathbf{e}_2$ in place of $\mathbf{e}_1$ shows $\mathbf{e}_2 \in H(\widetilde{C}_j)$. Hence $H(\widetilde{C}_j)$ contains $\mathbf{e}_1$ and $\mathbf{e}_2$, and thus

$$H(\widetilde{C}_j) = \mathbb{Z}^2. \tag{142}$$

By Lemma E.1, this implies $H(C_j) = \mathbb{Z}^2$.

Assume for contradiction that $|J| \geq 2$, and pick two distinct components $C_{j_1}, C_{j_2}$. Choose lift components $\widetilde{C}_{j_1} \subset \pi^{-1}(C_{j_1})$ and $\widetilde{C}_{j_2} \subset \pi^{-1}(C_{j_2})$. By Step 1, both satisfy $H(\widetilde{C}_{j_\ell}) = \mathbb{Z}^2$, hence each $\widetilde{C}_{j_\ell}$ is a $\mathbb{Z}^2$-invariant path-connected subset of $\mathbb{R}^2$. Then Lemma E.2 yields $\widetilde{C}_{j_1} \cap \widetilde{C}_{j_2} \neq \varnothing$, which contradicts the fact that $\pi(\widetilde{C}_{j_1}) \subset C_{j_1}$ and $\pi(\widetilde{C}_{j_2}) \subset C_{j_2}$ with $C_{j_1} \cap C_{j_2} = \varnothing$. Therefore $|J| = 1$, i.e. $C$ is path-connected.

Since $C$ is path-connected and $H(C) = \mathbb{Z}^2$, Lemma E.1 implies that $\pi^{-1}(C)$ has only one lift component, hence $\pi^{-1}(C)$ is path-connected. Using $S = \pi^{-1}(C)$, we conclude that $S$ is path-connected. □

# F. Experimental Details

For continuous 2D representations, we adopt a hash-grid encoding with bilinear interpolation, inspired by InstantNGP (Müller et al., 2022). Unlike orignial InstantNGP, which feeds multi-level interpolated embeddings into an MLP, we remove the MLP for more stable latent-space SDS optimization and directly average embeddings across levels. The encoding is implemented with multi-resolution hash-table embeddings queried by bilinear interpolation, and we apply random rotations and translations to input coordinates to reduce grid artifacts. We set the number of levels and hash-table capacity to $L = 16$ and $T = 2^{19}$, respectively. Since the latent space typically contains denser information, we adopt a relatively narrow and high-resolution range with $N_{\min} = 128$ and $N_{\max} = 256$. While for the pixel-space baselines in § 6.3, we follow Zhong et al. (2023) and set $N_{\min} = 8$ and $N_{\max} = 128$.

## F.1. Pattern Design

The lattice parameters of all 17 planar groups are specified by the symmetry-operation markers shown in Fig. 10. We use these parameters in all pattern design experiments.

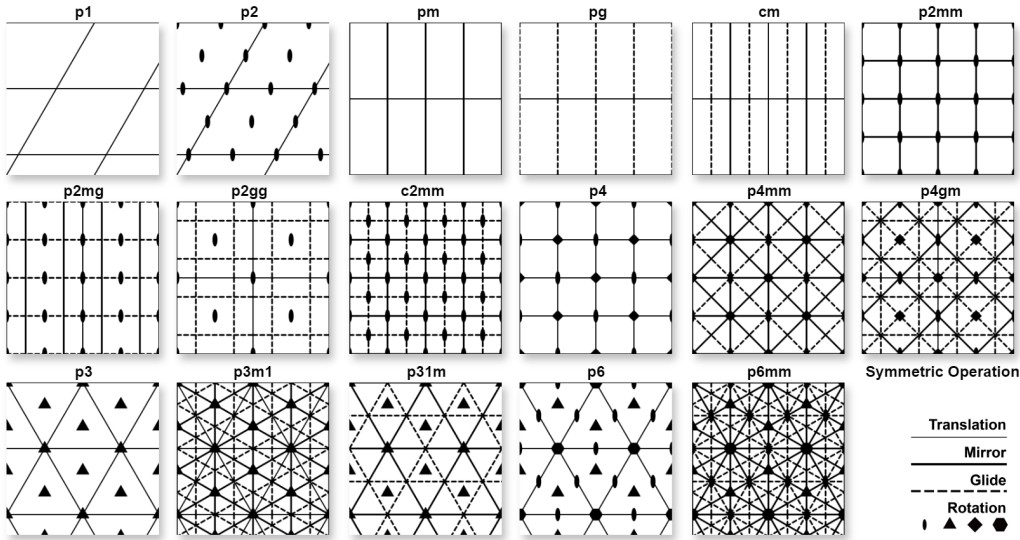

*Figure 10.* Reference images illustrating the symmetry operations of the 17 planar groups with markers.

### F.1.1. VISULIZATION

For the generative tasks, during SDS optimization, we use the positive prompt: *stained-glass mosaic fragments, simple polygon shards with thick lead outlines*, and the negative prompt: *lowres, bad anatomy, error, extra digit, fewer digits, worst quality, watermark*. We perform optimization in the latent space with a resolution of $128 \times 128$. The optimization process is conducted for a total of 200 steps using the AdamW optimizer with a learning rate of 0.01. We employ a dynamic guidance strategy, and the CFG scale is linearly annealed from an initial value of 100 to a final value of 7.5. We also employ a linear annealing strategy for the timestep, decreasing from $0.4T$ to 0 (where $T$ represents the total diffusion timesteps). Following the SDS convergence, we apply a refinement stage to enhance image quality and correct potential artifacts. We perturb the optimized latent code by injecting noise corresponding to $t = 0.4T$ and subsequently denoise it back to $t = 0$ using the standard diffusion sampling process with 100 inference steps. The resulting images are shown in Fig. 11.

### F.1.2. COMPARISON WITH TEXT-CONDITIONED GENERATION

For our method, we adopt the SDS-based optimization and refinement pipeline described in § F.1.1, with the following modifications. We run SDS optimization for 800 steps with a learning rate of 0.03, anneal the diffusion timestep from $0.5T$ to 0, and inject noise at $t = 0.5T$ before the final refinement denoising. The negative prompt remains unchanged, while the positive prompts are specified later. The final generation results are presented in Fig. 12.

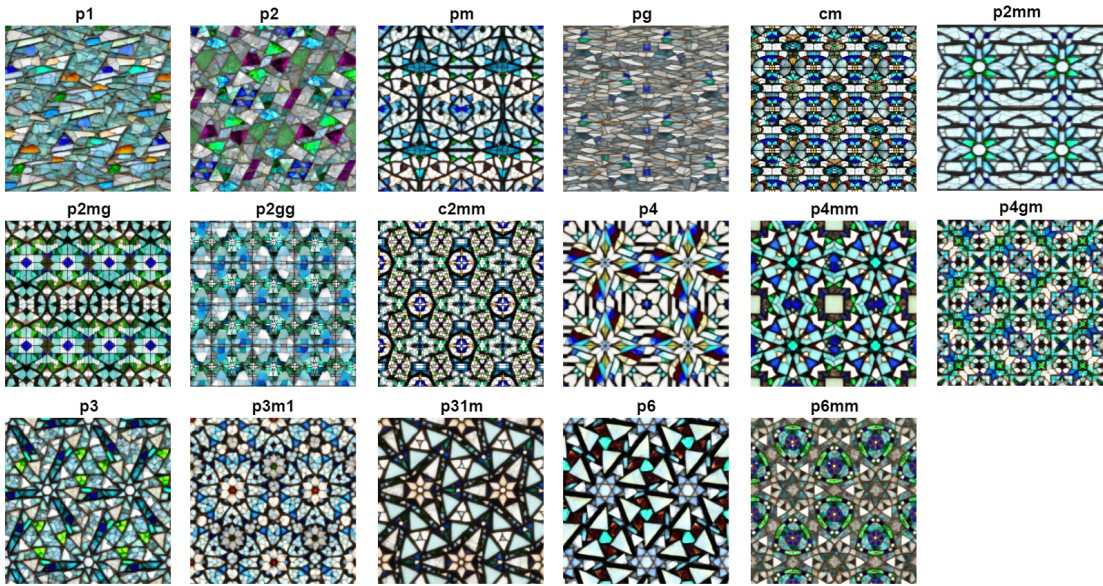

*Figure 11.* Generated patterns for visualization across the 17 plane symmetry groups.

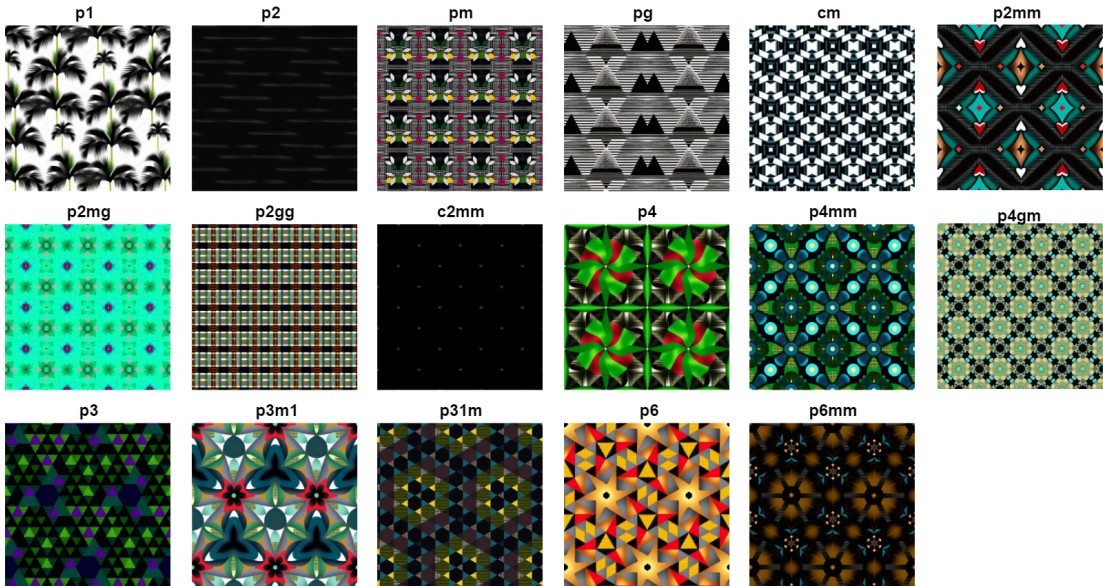

*Figure 12.* Generated patterns for comparison across the 17 plane symmetry groups.

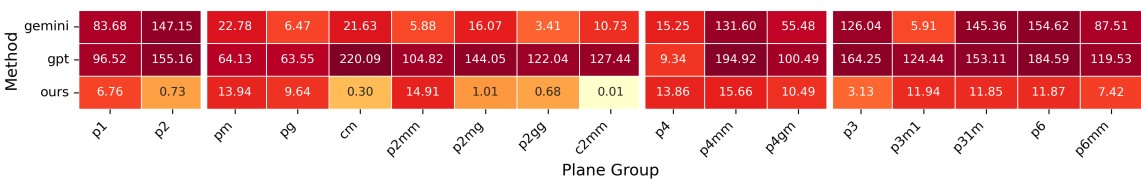

*Figure 13.* **MSE comparison across 17 plane groups for different methods.** The plane groups are grouped by lattice type. Entries in each cell report $\text{MSE} \times 10^3$, while cell colors are determined by $\log_{10}(\text{MSE})$. Our method consistently achieves lower errors across most groups, demonstrating superior symmetry preservation.

**Prompts Collection.** As mentioned in Sec. 6, we curated a set of 17 text prompts, each corresponding to one of the 17 symmetry groups. These prompts serve as the basis for our comparative evaluation. The original concepts were derived from the examples listed in Tab. 9 of Shubnikov & Koptsik (1974). We utilized Gemini to label these examples and simplify the descriptions into concise text prompts suitable for text-to-image generation. The complete mapping between symmetry groups and their corresponding prompts is detailed below

---

**Symmetry Groups with Corresponding Prompts**

**p1:** *palm fronds, seamless pattern, repeating, high contrast*

**p2:** *Horizontal striped pattern, vertical lines and running waves, seamless, repeating, high contrast*

**pm:** *floral line pattern, seamless, repeating, diagonal*

**pg:** *geometric triangle pattern, seamless, repeating, high contrast*

**cm:** *square forming a continuous meander maze, right-angled lines, seamless, repeating, geometric*

**p2mm:** *heart-shaped frames with arrowheads, seamless, repeating, high contrast*

**p2mg:** *lotus flowers, seamless, repeating, geometric*

**p2gg:** *rectangular blocks, checkerboard, seamless, repeating, geometric*

**c2mm:** *ornamental pattern, diamond grid, seamless, repeating*

**p4:** *pinwheel pattern, seamless, repeating, tiled*

**p4mm:** *radial circle pattern, seamless, repeating, geometric*

**p4gm:** *arabesque, seamless, repeating, symmetric*

**p3:** *triangular geometric pattern, seamless, repeating*

**p3m1:** *petal tiling pattern, seamless, repeating, curved*

**p31m:** *honeycomb geometric pattern, seamless, repeating*

**p6:** *star motif pattern, triangle tiling, seamless, repeating*

**p6mm:** *snowflake pattern, seamless, repeating, geometric*

---

**Prompt Templates of MLLMs.** The prompt templates used for baseline models, including GPT-5.2, Gemini 3 Pro, and SD 2.1, are defined as follows. For direct generation, the models are instructed to generate images directly from the text description without explicit symmetry constraints, using the template: *Generate a square image based on the prompt: [pos_prompt].* For conditional generation, we provide an auxiliary visual reference to guide MLLMs toward a specific plane symmetry group, using the template: *Based on the input prompt and the [Group Name] plane symmetry group in the reference image, draw a square picture. DO NOT draw markers or lines. Prompt: [pos_prompt].*

**Details of Post-Symmetrization.** In post-symmetrization, we strictly enforce symmetry on images. We project the generated non-perfect images into our symmetric parameterization space. Let $I_{\text{ref}}$ be the input image generated by a baseline model. We initialize our symmetric generator, denoted as a parameterized lattice representation $\mathcal{G}_\phi$, where $\phi$ represents the learnable parameters of the symmetric feature field. The lattice configuration is scaled (typically by a factor of 8) to accommodate high-resolution optimization ($1024 \times 1024$). We optimize the parameters $\phi$ such that the generated symmetric image $I_{\text{sym}} = \mathcal{G}_\phi$ approximates $I_{\text{ref}}$ by minimizing the MSE loss

$$\mathcal{L}_{\text{MSE}} = \|\mathcal{G}_\phi - I_{\text{ref}}\|_2^2.$$

The optimization is conducted using the AdamW optimizer with a 0.1 learning rate and 500 steps. The quantitative MSE results are reported in Fig. 13.

### F.1.3. COMPARISON WITH OTHER SYMMETRIZATION

For both method, we adopt the SDS-based optimization and refinement pipeline described in § F.1.1, with the following modifications. We conduct an ablation over the output resolution. In this experiment, we do not use a negative prompt and evaluate only the p1 group. The positive prompts are kept the same as the 17 prompts used in § F.1.2.

We implement the projection operator by constructing a finite Fourier basis associated with the target lattice. Given the lattice parameters, we first compute the corresponding reciprocal lattice and enumerate the integer reciprocal-lattice points $(h, k)$ within the Nyquist region. These frequencies define a band-limited periodic subspace. For each sampled pixel, we convert its Cartesian coordinate to the natural lattice coordinate $(u, v)$ and evaluate the Fourier basis functions $1$, $\cos(2\pi(hu + kv))$ and $\sin(2\pi(hu + kv))$. To avoid redundant basis functions, we keep only one representative from each pair of opposite reciprocal frequencies. Specifically, we retain the frequencies satisfying $h > 0$ or $h = 0, \; k > 0$.

Stacking the basis values over all sample points gives a basis matrix $\mathbf{\Phi}$. We then orthonormalize $\mathbf{\Phi}$ by QR decomposition, obtaining $\mathbf{\Phi} = \mathbf{QR}$. The projection of a flattened image $\mathbf{x}$ onto this periodic subspace is computed as $\mathbf{QQ^{T}x}$. The same projection is applied independently to each image channel. In practice, we compute the projection over the full sampling grid rather than only over the parallelogram cell, which reduces boundary discontinuities and alleviates ringing artifacts.

### F.2. Paper-Cutting Design

**Details of fine-tuning.** Our dataset consists of 140 Chinese paper-cutting images, each annotated with a regional style label. Based on these labels, we construct the text prompt for each training sample using the following template: *"traditional chinese papercut art, [regional style], high quality, detailed, artistic, traditional craftsmanship, paper cutting, chinese folk art, intricate patterns, cultural heritage"*. The prompts are used as the text conditioning input during training.

We fine-tuned SDXL using a LoRA-based adaptation implemented with Diffusers. All training images were resized to $1024 \times 1024$, randomly cropped, randomly horizontally flipped, and normalized to the range $[-1, 1]$. During fine-tuning, the VAE, both SDXL text encoders, and the original U-Net weights were frozen, and only the inserted LoRA parameters were optimized. LoRA adapters were added to the U-Net attention projection layers, with rank $r = 4$ by default. The model was trained with batch size 1 for 1000 epochs using AdamW, with a learning rate of $1 \times 10^{-4}$, a constant learning-rate scheduler, and gradient clipping with a maximum norm of 1.0.

**Details of paper-cutting design.** We select four representative symmetry groups: $p2mm$, $p4mm$, $p3m1$, and $p6mm$. To demonstrate generalization, we apply a shared prompt, *red Chinese paper cutting, flowers* across all four groups (shown in the top row of Fig. 6a). In contrast, the bottom row displays results generated using prompts tailored specifically to each group. The detailed prompts are listed below:

---

**Prompts Used in Paper-Cutting Design**

**p2mm:** *red Chinese paper cutting, lantern*

**p4mm:** *red Chinese paper cutting, copper*

**p3m1:** *red Chinese paper cutting, star*

**p6mm:** *red Chinese paper cutting, snowflakes*

---

For the connectivity constraint, we utilize the Virtual Temperature Method (VTM) with $\Gamma = \{0\} \times [0, 1) \cup [0, 1) \times \{0\}$ in the coordination determined by fundamental translation $\mathbf{a}$ and $\mathbf{b}$. We solve the heat-conduction equation on a $2 \times 2$ supercell. The mesh edge lengths are $1/2 \times 1/2$, with the included angle $\gamma$ between $\mathbf{a}$ and $\mathbf{b}$, and we use a $128 \times 128$ mesh. The heat source ranges from $10^{-8}$ to $10^{-4}$, and the thermal conductivity ranges from $10^{-4}$ to 1. The SIMP penalty is set to 5. To approximate the maximum temperature, we use a differentiable $p$-norm aggregation with exponent 20. We adopt four-node bilinear quadrilateral (Q1) shape functions with $2 \times 2$ Gauss quadrature. Before segmentation, we apply density filtering with radius 2 and step size 1.

In the cases, we keep the coefficient of the SDS loss fixed to 1. For generation, we fix the learning rate to $10^{-2}$, the target volume fraction $\rho_0$ to 0.35, and the connectivity penalty $\lambda_{\text{conn}}$ to $10^2$. To identify the optimal configuration for paper-cutting results, we perform a grid search over the volume penalty $\lambda_{\text{vol}} \in \{10^3, 3 \times 10^3, 5 \times 10^3, 10^4\}$, the end binary ratio in $\{0.0, 0.5\}$, and the number of optimization steps in $\{200, 400, 800\}$. The weight of the rendered $z_\theta^{\text{bin}}$ is increased linearly from 0 to the specified end binary ratio during optimization. We employ a linearly annealed CFG scale, which starts from 100 and decays to 7.5 over the course of optimization. We also apply a linear timestep annealing strategy, decreasing the timestep from $T$ to 0. For the color, $c_{\text{solid}}$ is chosen as the averaged latent vector across latent pixels obtained by feeding a pure red image into the SDXL VAE encoder, while $c_{\text{void}}$ is chosen analogously using a pure white image.

## F.3. Topology Design

As discussed in § 6, we utilize the set of 12 test prompts in Zhong et al. (2023). Details are as follows:

---

**Prompts Used in Topology Design**

*golden, Baroque style*

*rainbow-color, spider web style*

*red, koi, Chinese paper cutting style*

*Autumn branches*

*wood appliques, simple*

*kaleidoscope art*

*modern, dream, wavy texture*

*rosewood texture*

*floral ornament*

*Persian carpet style*

*Art Deco*

*Art Nouveau*

---

To mitigate potential artifacts during SDS optimization, we append the suffixes *, tessellation, pure white background* to the prompts. Additionally, a negative prompt is employed: *black, shadow, lowres, bad anatomy, error, extra digit, fewer digits, worst quality, watermark, 3d, shadow, blur, artifact, deformed, distorted, noisy*. As our SDS optimization operates within the latent space, it necessitates a latent representation of the background color. To achieve this, we encode a pure white image using the VAE encoder and utilize the resulting latent code as the white background in the latent space.

Both the baseline and our method share a unified physical simulation environment to ensure a fair comparison. VTM setting follows § F.2. For the mechanical constraint, we employ homogenized finite element analysis (FEA) on oblique lattice elements. The mesh edge lengths are $1 \times 1$, with the included angle $\gamma$ between $\mathbf{a}$ and $\mathbf{b}$; we use a $64 \times 64$ mesh during training and a $128 \times 128$ mesh during testing. We adopt the plane-stress constitutive matrix, with Young's modulus ranging from $10^{-6}$ to 1, Poisson's ratio $\nu = 0.3$, and the SIMP penalty factor set to $p = 10$. We again use Q1 quadrilateral elements with $2 \times 2$ Gauss quadrature. Before segmentation, we apply density filtering with radius 3 and step size 1. After segmentation, we apply a Heaviside projection before FEA:

$$\bar{\rho} = \frac{\tanh(\beta\eta) + \tanh\big(\beta(\rho - \eta)\big)}{\tanh(\beta\eta) + \tanh\big(\beta(1 - \eta)\big)},$$

where $\beta$ is linearly annealed from 1 to 8, and the threshold is $\eta = 0.3$. The target volume fraction for all topology design experiments is fixed at $\rho_0 = 0.45$. In all cases, we keep the coefficient of the mechanical loss fixed to 1. The loss weights and optimization schedules differ between the baseline and our method as follows.

For baseline, we optimize the topology for 401 steps. The loss weights are configured as: volume penalty $\lambda_{\text{vol}} = 3 \times 10^4$, connectivity penalty $\lambda_{\text{conn}} = 10^2$, and the semantic CLIP loss weight $\lambda_{\text{clip}} = 5 \times 10^3$. For our method, we have optimization process of 801 steps to ensure convergence of the generative objective. The physical constraint weights remain consistent with the baseline ($\lambda_{\text{vol}} = 3 \times 10^4$, $\lambda_{\text{conn}} = 10^2$) to enforce comparable structural validity. The SDS loss weight is set to $\lambda_{\text{sds}} = 0.3$. We employ a linearly annealing CFG scale, starting at 50 and decaying to 7.5 over the optimization. We also employ a linear annealing strategy for the timestep, decreasing from $T$ to 0. Following the SDS convergence, we apply a refinement stage identical to that described in § F.1.

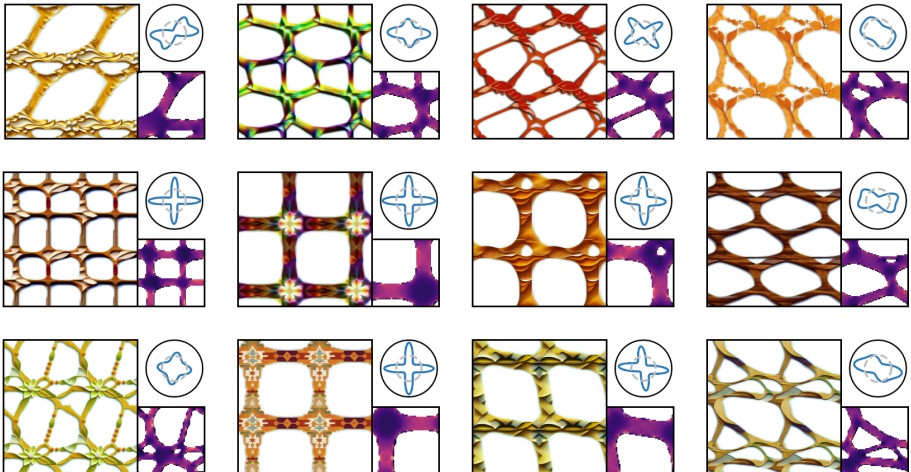

*Figure 14.* **Visualization of topology design under** $p1$ **symmetry.** Left: topology-optimized designs. Top right: directional Young's modulus, where blue curves indicate values across directions and the gray circle denotes the mean. Bottom right: energy distribution of the bulk modulus in unit cell.

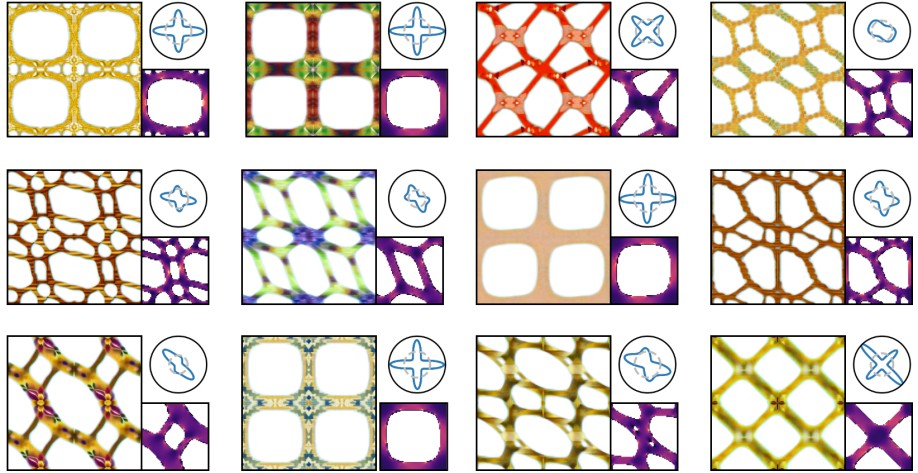

*Figure 15.* **Visualization of topology design under** $p2$ **symmetry.** Visualization settings are the same as in Fig. 14.

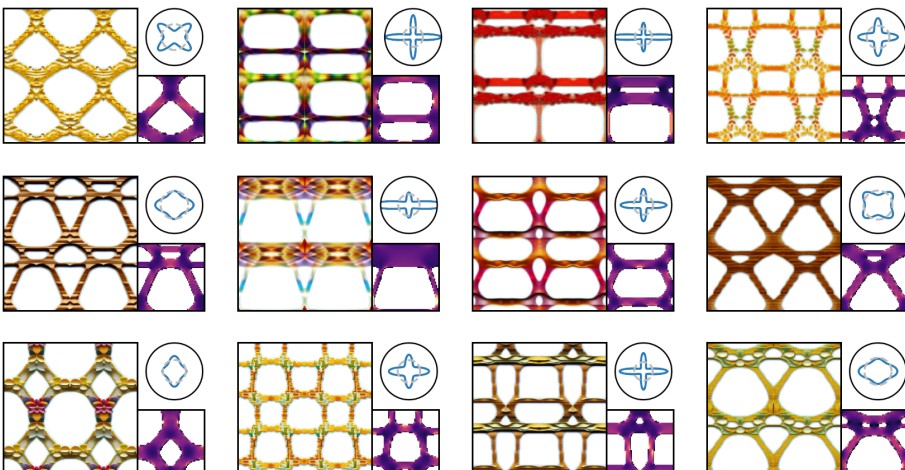

*Figure 16.* **Visualization of topology design under** $pm$ **symmetry.** Visualization settings are the same as in Fig. 14.

## F.4. Metematerial Design

**Data Generation.** We generate the $p1$ metamaterial training set using homogenization-based topology optimization. Each unit cell is discretized on a $64 \times 64$ square finite-element grid, where each element has a density variable. The material stiffness is interpolated by a SIMP-type model with penalization $p = 5$ and $E_{\min} = 10^{-6}$. We use periodic boundary conditions for numerical homogenization and optimize the density field to maximize the homogenized bulk modulus under a volume fraction constraint $0.5$.

Each sample is initialized from a uniform density field with a softened circular region at the center follows Xia & Breitkopf (2015), together with a small periodic Fourier perturbation to introduce diversity. We use Fourier modes up to $5$ and perturb the initial density with amplitude $0.03$. During optimization, sensitivities are smoothed by a periodic sensitivity filter with radius $3$, and the density variables are updated by the optimality criteria method with move limit $0.1$. Each design is optimized for at most $200$ iterations and terminated early when the maximum density change is below $0.01$. The final continuous density field is binarized by thresholding at $0.5$, resulting in a $64 \times 64$ binary unit-cell mask. No symmetry other than the basic translational periodicity is imposed during data generation.

**Diffusion model training.** We train an unconditional diffusion model on the generated $p1$ unit-cell masks. All samples are represented as single-channel $64 \times 64$ images and normalized to $[-1, 1]$. The denoising network is a convolutional U-Net with sinusoidal timestep embeddings, residual blocks, group normalization, SiLU activations, and encoder-decoder skip connections. The base channel width is set to $64$, with channel multipliers $(1, 2, 2, 2)$. We use a standard noise-prediction objective with $T = 1000$ diffusion steps and a linear noise schedule from $\beta_1 = 10^{-4}$ to $\beta_T = 0.02$. At each training iteration, we uniformly sample a timestep $t$, perturb the clean image $\mathbf{x}_0$ as

$$\mathbf{x}_t = \sqrt{\bar{\alpha}_t}\mathbf{x}_0 + \sqrt{1 - \bar{\alpha}_t}\epsilon, \epsilon \sim \mathcal{N}(0, I),$$

Here $\bar{\alpha}_t = \prod_{s=1}^{t}(1 - \beta_s)$ is the cumulative signal-preserving coefficient. We train the U-Net to predict the added noise $\epsilon$ using an MSE loss. The model is optimized with Adam using a batch size of $128$ and a learning rate of $10^{-4}$ for $100$ epochs. No symmetry labels or symmetry-specific data are used during training. For visualization, we periodically generate samples using DDIM sampling with $100$ denoising steps.

**SDS optimization.** For the generative tasks driven by SDS, we perform optimization in the parametric symmetric representation space at a resolution of $64 \times 64$. For each prescribed planar group, the representation is instantiated with one density channel and lattice parameters $a = b = 64$ and $\gamma = \pi/2$, consistent with the square unit-cell domain. Each sample is optimized for $300$ steps using the AdamW optimizer with a learning rate of $1 \times 10^{-1}$, $\beta = (0.9, 0.99)$, and $\epsilon = 10^{-15}$. The diffusion guidance uses $T = 1000$ timesteps, and the SDS timestep is sampled from the range $[0.02T, 0.98T]$. During optimization, we pass a normalized progress ratio $i/(N - 1)$ to the SDS objective, where $i$ denotes the current optimization step and $N = 300$ is the total number of steps. The output of the symmetric representation is transformed by a $\mathtt{tanh}$ activation before being fed into the SDS loss. After optimization, the resulting continuous density field is clipped to $[-1, 1]$, rescaled to $[0, 1]$, and binarized using a threshold of $0.5$ to obtain the final unit-cell mask.

## G. Table of Affine Reflection Supergroups and Secondary Invariant

The symmetric continuous representation in § 3 relies on the Hironaka-type decomposition: a $G$-invariant field can be written as a linear combination of a finite set of fixed $G$-invariant basis functions (secondary invariants) with coefficient fields that enjoy a higher affine reflection symmetry $W_a$ (cf. Eq. (3) and Thm. 3.3). For practical use, the only group-dependent ingredient is the explicit choice of these basis functions $\{\eta_i\}_{i=1}^r$, where $r = [W_a : G]$. This section tabulates non-trivial $\eta_i$ for all planar groups (we omit trivial $\eta_1 = 1$y), together with a compatible embedding $G \subset W_a$ and the associated lattice generators $(\mathbf{a}, \mathbf{b})$ of $W_a$. The table serves as a plug-in recipe: once the target symmetry group $G$ and lattice are fixed, we directly obtain $(W_a, \mathbf{a}, \mathbf{b})$ and the corresponding $\eta_i$, and then parameterize $G$-symmetric continuous fields via Eq. (3).

All notations in § G follow the same conventions as in § 3. In particular, $c_i$ and $s_i$ denote the cosine and sine generators associated with the fundamental lattice directions. For the hexagonal lattice, the secondary invariants appearing in the table are defined as

$$\phi_1^{-+} = s_1 + s_2 - (c_1 s_2 + c_2 s_1),$$
$$\phi_2^{--} = s_1 - s_2 + c_1 s_2 - c_2 s_1 + 2(c_1 - c_2)(c_1 s_2 + c_2 s_1).$$

*Table 2.* Affine Reflection Supergroups and Secondary Invariants

| Lattice | $G$ | $W_a$ | $\mathbf{a}$ | $\mathbf{b}$ | $\eta_2$ | $\eta_3$ | $\eta_4$ | $r$ |
|---|---|---|---|---|---|---|---|---|
| Oblique | $p1$ | | $\mathbf{a}$ | $\mathbf{b}$ | $s_1$ | $s_2$ | $s_1 s_2$ | 4 |
| Oblique | $p2$ | | $\mathbf{a}$ | $\mathbf{b}$ | $s_1 s_2$ | – | – | 2 |
| Rectangular | $pm$ | | $\mathbf{a}$ | $\mathbf{b}$ | $s_1$ | – | – | 2 |
| Rectangular | $pg$ | | $\mathbf{a}$ | $\mathbf{b}/2$ | $s_1 c_2$ | $s_1 s_2$ | $c_2 s_2$ | 4 |
| Rectangular | $cm$ | $p2mm$ | $\mathbf{a}/2$ | $\mathbf{b}/2$ | $c_1 c_2$ | $c_1 s_2$ | $c_2 s_2$ | 4 |
| Rectangular | $p2mm$ | | $\mathbf{a}$ | $\mathbf{b}$ | – | – | – | 1 |
| Rectangular | $p2mg$ | | $\mathbf{a}/2$ | $\mathbf{b}$ | $s_1 s_2$ | – | – | 2 |
| Rectangular | $p2gg$ | | $\mathbf{a}/2$ | $\mathbf{b}/2$ | $c_1 c_2$ | $c_1 s_1 s_2$ | $s_1 c_2 s_2$ | 4 |
| Rectangular | $c2mm$ | | $\mathbf{a}/2$ | $\mathbf{b}/2$ | $c_1 c_2$ | – | – | 2 |
| Square | $p4$ | | $\mathbf{a}$ | $\mathbf{b}$ | $(c_1 - c_2)s_1 s_2$ | – | – | 2 |
| Square | $p4gm$ | $p4mm$ | $\mathbf{a}/2$ | $\mathbf{b}/2$ | $c_1 c_2$ | $(c_1 - c_2)s_1 s_2$ | $(c_1 - c_2)c_1 c_2 s_1 s_2$ | 4 |
| Square | $p4mm$ | | $\mathbf{a}$ | $\mathbf{b}$ | – | – | – | 1 |
| Hexagonal | $p3$ | | $\mathbf{a}$ | $\mathbf{b}$ | $\phi_1^{-+}$ | $\phi_2^{--}$ | $\phi_3^{+-} = \phi_1^{-+}\phi_2^{--}$ | 4 |
| Hexagonal | $p3m1$ | | $\mathbf{a}$ | $\mathbf{b}$ | $\phi_1^{-+}$ | – | – | 2 |
| Hexagonal | $p31m$ | $p6mm$ | $\mathbf{a}$ | $\mathbf{b}$ | $\phi_2^{--}$ | – | – | 2 |
| Hexagonal | $p6$ | | $\mathbf{a}$ | $\mathbf{b}$ | $\phi_3^{+-}$ | – | – | 2 |
| Hexagonal | $p6mm$ | | $\mathbf{a}$ | $\mathbf{b}$ | – | – | – | 1 |

