# OpenReview forum: "Planar Symmetric Pattern Generation"
_ICML.cc/2026/Conference — ICML 2026 regular_

### Official Review · Reviewer_pQEj · 2026-03-01

**Soundness:** 3
**Presentation:** 3
**Significance:** 3
**Originality:** 3
**Overall Recommendation:** 4
**Confidence:** 4

**Summary:**

This paper introduces a framework for generating continuous, symmetric patterns that follow geometric group constraints. By embedding planar groups into an affine reflection structure, the method ensures that patterns remain continuous across the boundaries, overcoming the common issue of artifacts or breaks. The framework's effectiveness is demonstrated across tasks like general pattern design, connectivity-preserving paper-cutting, and mechanical topology optimization. By combining this symmetric representation with diffusion models and physical constraints, the authors can generate high-quality, manufacturable structures that are both aesthetically pleasing and geometrically precise.

**Compliance With Llm Reviewing Policy:**

Affirmed.

**Final Justification:**

I think it is a legit paper to publish. I do keep the ranking the same since the symmetry / asymmetry or multiple symmetries scenarios is not mathematically modeled in this paper and I believe it should. While the suggested approach is useful I think that the real challenge remains in those two aspects.

**Key Questions For Authors:**

1. Please provide numerical comparison of this approach versus other methods - quantify the symmetry level.
2. Did you research the symmetry level versus asymmetry modeling - specifically softening the symmetric rules while building a symmetric shapes with imperfect parts.

**Limitations:**

The system is designed to handle one distinct planar symmetry. This approach is limited to enforcing one specific symmetry group per generation. It would be great if the authors explain how to mis different types of symmetry types - for example - having one part of the image follow square symmetry and another hexagonal symmetry or localizing asymmetric details compromising the overall geometry coded perfection.

**Strengths And Weaknesses:**

The primary strength of this approach is its ability to ensure global continuity across all known planar symmetry groups, even those involving non-reflective elements like rotations and glide reflections. Unlike simple mirroring, which often creates seams or visual breaks where tile edges meet—this method uses a mathematical foundation (affine reflection group embedding and Hironaka decomposition) to construct the pattern as a single, continuous field. This ensures that the design is not just mathematically perfect in its symmetry, but also visually flawless, with no abrupt transitions or artifacts at the unit boundaries.

A major weakness of this approach is the lack of asymmetric modeling.  By strictly hard-coding symmetry into the generation process, the model loses the ability to include the subtle, intentional imperfections or asymmetric flourishes that make designs feel natural and visually dynamic. In the paper's experiments, when artistic images from unconstrained models (like GPT-5 or Gemini) were forced into this symmetric framework, their aesthetic scores dropped. This suggests that the price of achieving perfect geometric alignment is a reduction in the fluid, creative diversity that unconstrained generative models naturally provide. The authors did not quantified the level of symmetry and asymmetry wither quantitively or qualitatively to have a fair apples to apples comparison.

---

> ### Author Rebuttal · Authors · 2026-03-31
>
> > **Link of supplementary materials**
>
> For visualization results of all additional experiments, kindly refer to: https://anonymous.4open.science/r/Sym2D-EB27/supp.pdf.
>
> > **C1. Asymmetric modeling**
>
> We thank the reviewer for the suggestion. In our response to Reviewer xBcx, we include additional experiments on symmetric metamaterial generation. In material design, symmetry is often linked to desirable physical properties and thus needs to be strictly enforced. In contrast, for visual design tasks, modeling asymmetry can provide greater flexibility.
>
> **Symmetric–asymmetric decomposition.** To model asymmetry, we decompose an image into symmetric and asymmetric components. In a special case, both components are plane-group symmetric, with the symmetric part corresponding to a higher-symmetry affine reflection group. Based on the Hironaka decomposition, a plane-group symmetric function can be written as
> $$
> f(x)=\sum_i h_i(x)\eta_i(x).
> $$
> Here, $\eta_1(x)\equiv1$ corresponds to the higher-symmetry component, while $i>1$ captures deviations, yielding a natural symmetric–asymmetric decomposition. More generally, such a decomposition can be constructed explicitly by combining a symmetric parameterization $f_1$ and an asymmetric parameterization $f_2$, i.e., $f(x)=w_1f_1(x)+w_2f_2(x)$.
>
> **Regularization.** This decomposition enables control of asymmetry via a penalty on the asymmetric component. Increasing the penalty progressively enhances symmetry. In our experiments (p2mm as symmetric, p1 as asymmetric), stronger penalties lead to more symmetric results (supplementary Fig. 9).
>
> We also test mixed symmetry settings without asymmetric penalties (supplementary Fig. 10, “Coupled”). In some cases, we observe a spontaneous tendency toward symmetry, indicating that symmetric structures can emerge as stable solutions under certain semantics. This also explains why, in Fig. 5 of the main text, imposing symmetry constraints can even improve the aesthetic quality of the generated results.
>
> > **C2. Handle mix symmetry**
>
> We thank the reviewer for the suggestion. The reviewer pointed out a practical scenario where different spatial regions may exhibit different symmetries, which indeed arises in real-world pattern design. We conduct a preliminary study on modeling such mixed symmetries.
>
> To handle this setting, we introduce spatially varying weights in the parameterization:
> $$
> f(x)=w_1(x)f_1(x)+w_2(x)f_2(x),
> $$
> where different regions can be assigned different symmetry preferences. We consider three representative types of mixed symmetry:
> (1) **Partitioned:** $w_1,w_2$ are designed such that one half of the image is asymmetric while the other half is symmetric;
> (2) **Gradient:** $w_1,w_2$ vary smoothly from left to right, transitioning from asymmetric to symmetric;
> (3) **Coupled:** $w_1,w_2$ are constant, resulting in a globally mixed symmetry.
>
> We conduct generation experiments under three symmetry–asymmetry combinations and these three mixing strategies (supplementary Fig. 10). We observe that:
> (1) The partitioned case is inherently non-smooth, and the optimization may lead to noticeable discontinuities;
> (2) In the gradient and coupled cases, asymmetric details can appear on top of the symmetric structure; however, for certain prompts and symmetry groups, the model tends to produce nearly symmetric results even without explicit penalties on the asymmetric component.
>
> > **C3. Quantify the symmetry level**
>
> We thank the reviewer for the suggestion. Error maps after symmetrization are already presented in Fig. 5 of the main text. Here, we further provide quantitative results by computing the MSE and comparing with baselines. The results show that our method achieves significantly lower symmetry error. Detailed MSE comparisons for each group are provided in supplementary Fig. 8.
>
> ||MSE
> -|-:|
> gemini|6.1e-02
> gpt|1.3e-01
> ours|**7.9e-03**

---

> > ### Author Rebuttal · Reviewer_pQEj · 2026-04-04
> >
> > Thank you for clarifying my concern.  I think it is a legit paper to publish.
> > I do keep the ranking the same since the symmetry / asymmetry or multiple symmetries scenarios is not mathematically modeled in this paper and I believe it should. While the suggested approach is useful I think that the real challenge remains in those two aspects.

---

> > > ### Author Response · Authors · 2026-04-07
> > >
> > > Thank you for your thoughtful feedback and for recognizing the value of our paper. We are glad that our rebuttal has addressed your main concerns.
> > >
> > > We would like to respectfully clarify that the two aspects you mentioned have in fact been discussed in our rebuttal. Specifically, in **C1** we provided a formulation for the **symmetry/asymmetry scenario** based on a symmetric-asymmetric decomposition, together with a regularization strategy to control the asymmetric component, while in **C2** we discussed the **multiple-symmetries scenario** through a mixed-symmetry parameterization. In addition to these formulations, we also provided corresponding visualization results to illustrate the behavior of our solution in these settings. These were intended to show that both scenarios can already be incorporated into our framework in a concrete way.
> > >
> > > We agree that these aspects deserve a systematic mathematical treatment in the paper itself. Overall, these scenarios can be naturally handled as extensions of our framework, and we will present these formulations explicitly in a revised version of the paper.

---

### Official Review · Reviewer_bDY6 · 2026-03-08

**Soundness:** 3
**Presentation:** 3
**Significance:** 4
**Originality:** 4
**Overall Recommendation:** 5
**Confidence:** 3

**Summary:**

In the paper, the authors propose a symmetric continuous representation for planar symmetric image generation. The core idea is to avoid naively defining a field on the asymmetric unit and extending it by group actions, because the paper argues that this creates boundary discontinuities for non-reflective groups. Instead, it embeds any planar group G into an affine reflection supergroup Wa, then represents a G-invariant field as a sum of fixed basis function with coefficients. The paper claims an approximation guarantee for continuous G-invariant functions and provides an explicit table of basis functions for all planar groups. On top of that representation, it builds a unified SDS-based pipeline for three tasks: text-conditioned symmetric pattern design, paper-cutting with connectivity enforced by the VTM, and topology design with homogenization.

**Compliance With Llm Reviewing Policy:**

Affirmed.

**Key Questions For Authors:**

- How symmetric the output image is? I hope the authors could show some error maps.
- Would it be possible to compare with some existing symmetric image generator in addition to the generic ones for pattern design? As for the topology optimization, I think it would better to measure the isotropy of the material since the paper is about symmetry.

**Limitations:**

The paper does not talk about limitation. One of the limitation I can think about is: if the generation pipeline can guarantee the output is symmetric?

**Strengths And Weaknesses:**

Strength:
- In my opinion, the strongest part is the conceptual bridge between invariant theory and generative design.
- The qualitative experiments over the proposed method's output look convincing.

Weakness:
- The biggest weakness is that the paper does not distinguish exact symmetry in the latent/continuous representation with exact symmetry in the final decoded image. I hope authors could clarify if the final image is exact symmetry or not and add some disscussion.
- The comparisons look thin, as there is not symmetric image generator for comparsion, but general image generator like GPT-5.2, Gemini 3 Pro. The CLIP-A is not a good metric in showing the symmetry.
- The presentation is a bit unclear. I hope the author could add some overview figure that show the complete pipeline including the proposed representation

---

> ### Author Rebuttal · Authors · 2026-03-31
>
> > **Link of supplementary materials**
>
> For visualization results of all additional experiments, kindly refer to: https://anonymous.4open.science/r/Sym2D-EB27/supp.pdf.
>
> > **C1. Symmetry error of the decoded image**
>
> We thank the reviewer for the comment and apologize for any confusion. We clarify the symmetry property of our method as follows. The latent representation we construct is strictly symmetric. However, the decoded images are not perfectly symmetric, since the decoder and the diffusion denoising process do not strictly preserve symmetry.
>
> **Analysis of Asymmetry.** We note that qualitative results are provided in Fig. 1 and quantitative visualization of error maps in Fig. 5 in the main text. Due to the translation equivariance of the convolutional decoder and the strong prior of diffusion models, the resulting symmetry error is visually negligible. To further quantify this effect, we compute the MSE and compare it with baseline methods. The results show that our method achieves significantly lower symmetry error. Detailed MSE comparisons for each group are provided in supplementary Fig. 8.
>
> ||MSE
> -|-:|
> gemini|6.1e-02
> gpt|1.3e-01
> ours|**7.9e-03**
>
> We further clarify that, in the metamaterial generation experiments (see our response to Reviewer xBcx), symmetry is defined directly in the physical space, where strictly symmetric results can be obtained (see supplementary Fig. 2). In addition, in the topology design experiments (Sec. 6.3), the baseline adopts a pixel-space parameterization, under which symmetry can also be strictly enforced.
>
> > **C2. Compare with symmetric image generators**
>
> Good suggestion! To the best of our knowledge, there is no existing method that can directly generate images with a specified plane-group symmetry. The closest related work [A] focuses on symmetric metamaterial generation, but requires large symmetry training data and trains separate models for each symmetry, which limits its applicability to general pattern generation.
>
> **Projection-based baseline.** As discussed in our response to Reviewer xBcx, a natural baseline is to project a 2D representation onto the group-invariant function space. However, this approach suffers from two key limitations: (1) prohibitive complexity ($O(n^6)$ time and $O(n^4)$ memory for an $n×n$ image), and (2) overly smooth initialization, leading to degraded image quality.
>
> We implement this baseline using trigonometric bases within the Nyquist frequency range with proportion $\alpha$. Under identical hyperparameters, our method shows clear advantages over this projection baseline.
>
> Method|n=64|n=128|n=256
> -|-:|-:|-:
> $α=0.25$|3.81|3.07|3.46
> $α=0.5$|3.98|3.38|3.61
> $α=1.0$|4.05|3.42|3.48
> Ours|**4.30**|**4.20**|**3.99**
>
> > **C3. Add a pipeline figure**
>
> We thank the reviewer for the suggestion and have added a pipeline figure to clarify the overall framework. Our method can be viewed as a special case of zero-shot controllable generation, consisting of three components: a differentiable parametric representation, task-specific losses, and a generative optimization process. We focus on imposing plane-group symmetry together with other task-related constraints. An architecture diagram is provided in the supplementary material (Fig. 4).
>
> **Symmetric Representation.** We enforce symmetry via a continuous 2D representation. Sec. 3 presents its formulation and approximation properties, and Sec. 4 describes its computation.
>
> For affine reflection groups (Sec. 3.1), symmetry is achieved by extending functions from the asymmetric unit while preserving continuity, but this does not generalize to plane groups due to discontinuities from non-reflective transformations.
>
> For general plane groups (Sec. 3.2), groups can be embedded into higher symmetric affine reflection groups, allowing symmetric functions to be decomposed into a reflection-symmetric function with basis weighting. This enables symmetrization for general plane groups using the construction in Sec. 3.1. Full computational details are provided in Sec. 4 and the appendix.
>
> **Tasks and Loss Constraints.** Task-specific constraints are imposed via regularization losses. Sec. 5 considers three tasks of increasing complexity: symmetric pattern design, papercutting, and metamaterial design, with corresponding SDS (visual), VTM (connectivity), and homogenization-based mechanical losses.
>
> > **C4. Measure the isotropy of the material**
>
> We thank the reviewer for the helpful suggestion. Compared to a single mechanical metric, isotropy provides a more comprehensive characterization of material properties. Following [A], we evaluate isotropy by computing the Young’s modulus along different directions as well as its mean value. We annotate the results for three symmetry groups on square unit cells, and the results are provided in the supplementary material (Figs. 5–7).
>
> [A] Mao et al. Designing complex architectured materials with generative adversarial networks. Science advances 6.17 (2020).

---

> > ### Author Rebuttal · Reviewer_bDY6 · 2026-04-05
> >
> > Overall my questions have been resolved.

---

> > > ### Author Response · Authors · 2026-04-07
> > >
> > > Thank you for your positive feedback. We are glad that our rebuttal has adequately addressed your concerns and resolved your questions. We sincerely appreciate your time and consideration.

---

### Official Review · Reviewer_xBcx · 2026-03-10

**Soundness:** 3
**Presentation:** 2
**Significance:** 3
**Originality:** 3
**Overall Recommendation:** 4
**Confidence:** 3

**Summary:**

The paper proposes a method for parametrizing symmetric functions that are invariant under a crystallographic group. In my understanding, the method proceeds in two steps: First, the lattice of parallelepipeds underlying all crystallographic groups is parametrized by a band-limited Fourier basis that is smooth across cell boundaries and can be hand-crafted / derived to fit the grid. Afterwards, additional linear constraints are computed to model additional "lower" symmetries (the additional invariant rotations and reflections that characterize the specific crystal structure). The paper applies the method so several applications, in particular generative image modeling.

**Compliance With Llm Reviewing Policy:**

Affirmed.

**Final Justification:**

I have raised my score because the rebuttal has clearly explained the advantages of the proposed technique (in particular: more efficient than simple numerical basis design) and sketched a a convincing application area in material science with relevance to the venue. I would encourage the authors to incorporate some of this information into a revised paper in a suitable form (maybe the discussion of base-lines would fit into the appendix).

**Key Questions For Authors:**

What is the main benefit of the approach over a simple base-line (parametrize the function space by some generic basis functions and solve a linear system to get the (approximate) subspace that preserves symmetry (in case, add a suitable regularizer, e.g. when using a pixel-grid parametrization, or smooth basis functions for continuity/smoothness)?

Why is the paper of strong interest to some part of the ICML community? (this is not a "negatively-biased" question - positioning the paper well could help imo, so I am seriously interested in a good answer).

Also, if I am misunderstanding the problem, its solution or the proposed approach, please correct me.

My opinion at this point is that I am a bit skeptical, though not negative, because I am not fully convinced of the importance of the proposal to the community at ICML, both in terms of technical advantage and relation to the problems people are facing; positioning the work well could probably address this concern (basically potentially improving "presentation" and "significance" in the rating above).

**Limitations:**

I do not see issues with specific social consequences, and the paper discusses this aspect accordingly.

In terms of technical limitations, I found little discussion on how the basis functions $h_i$ are computed and how the problem is discretized in the end. This could come with some obstacles (global support of a Fourier-type basis) and reintroduce issues (such as non-conforming parameters when computations are done at the pixel/voxel-level; these issues are probably minor, but it seems to me that circumventing them in the first place is a big appeal of the approach).

**Strengths And Weaknesses:**

The paper proposes a method for smooth and (through the Fourier-basis) global parametrization of symmetric functions on lattices; the construction might be a bit niche in terms of application area but it looks technically sound (I was not able to check all the details of the Fourier construction) and elegant if a global functional characterization is needed (no explicit cells, no placement of finite elements or radial basis functions or the like used in prior work). The results appear convincing; it is hard to formally evaluate a parametrization of a function space, as the main impact on results probably originates from the generative models employed rather than from the kind of symmetry restriction. This means, the I would consider the results convincing as they are - the method seems to work.

The main point that I found to some extend not satisfactory was the rather involved derivation, theory and method: Fundamentally, from what I know, the problem at hand is rather easy and basic: If we have a group $G$ acting on $\mathbb{R}^d$ and a function $\mathbb{R}^d \rightarrow \mathbb{R}$, the constraint $\forall g \in G: f(g(x)) = f(x)$ is a linear constraint on a the broader function space where $f$ is from. This means, if we have any finite-dimensional paramterization of $f$, we can just numerically compute a basis that spans the space of all symmetric (invariant) functions. A challenge might be to make the base discretization conform with $G$; I would assume that this is the main contribution of the paper, but to my understanding, it does not state this (very) explicitly. I am aware of previous approaches that have reached this goal with very simple means, such as just using Gaussian radial basis functions on a symmetrized blue-noise pattern [R1]. This works because Gaussians impose a band-limitation (and thus controllable smoothness beyond $C^\infty$) and they are invariant under rigid motions, i.e., not distorted by the symmetry transformations. Other constructions, such as approximately discretizing functions on a pixel or voxel grid and then using least-squares constraints for symmetry would be easy to implement, too.

The proposed method still has advantages over the simple baselines (no need to find canonical cells or sample basis functions, it seems to naturally handle infinite grids), but my impression is that the paper does not describe this very clearly at this point. Overall, this dampens my enthusiasm, but not into negative territory.

A separate concern, which is very subjective, would be whether the paper is well-suited for ICML; it seems to me that the application area is a bit niche and topic-wise maybe a better fit for a venue in computer graphics or the similar.

Finally, I should state that my background in symmetry and group theory is very much from an engineering background; so I cannot vouch for the formal derivations in detail, and I do not know the "more mathematical" literature on how function spaces are discretized under symmetry constraints (which, arguably, constitutes a lot of what theoretical physics is concerned with; so I would guess that there might be more to say about this, but I cannot).

[R1] Wu et al.: Real-Time Symmetry-Preserving Deformation. Pacific Graphics 2014.
Note: This is probably not the canonical reference to cite at this point, please see also references in the paper; it was just the piece that came to my mind as it solves a similar problem in parametrizing 3D vector (deformation) fields in a symmetry preserving way, and it is really simple. The earliest related reference I am aware of is
[R2] Lipman et al.: Symmetry Factored Embedding And Distance, ACM Siggraph/TOG 2010 (Here, the application is more different, and they also did not "invent" linear symmetry constraints; in case, see also their related work.)

---

> ### Author Rebuttal · Authors · 2026-03-31
>
> > **Link of supplementary materials**
>
> For visualization results of all additional experiments, kindly refer to: https://anonymous.4open.science/r/Sym2D-EB27/supp.pdf.
>
> > **C1. Rather involved theory of easy problem**
>
> Insightful comment! We clarify the difficulty of symmetrization of continuous representations.
>
> **Symmetry constraint.** While symmetry constraints are linear and could in principle be enforced via invariant equations, the reviewer's approach is not directly applicable. First, plane groups are infinite, leading to infinitely many constraints and making numerical solutions intractable, unlike prior works [R1, R2] on finite groups. Second, in continuous representations (e.g., NeRF), the mapping from parameters to function values may be nonlinear, so solving for symmetric parameters is generally infeasible. This motivates alternative formulations such as projection-based methods.
>
> **Baselines.** There are three naive approaches: (1) group averaging, (2) parametrizing a asymmetric unit and extending via group actions, and (3) projection onto invariant function spaces. The first approach is infeasible for infinite groups. The second may introduce boundary discontinuities under non-reflective transformations. The third is theoretically feasible but, as shown in **C2**, suffers from practical limitations.
>
> > **C2. Advantage over the simple baseline**
>
> The projection-based approach suffers from two limitations: (1) prohibitive time and memory complexity, and (2) overly smooth initialization that degrades image quality.
>
> **Complexity analysis.** Let the image resolution be $n×n$. A natural basis for plane groups is the trigonometric basis. By the Nyquist sampling theorem, the number of recoverable frequency components is approximately equal to the pixel number within a asymmetric unit. Considering an $α$ fraction of bases and $β$ asymmetric units, the number of bases is $m\approx αn^2/β$.
>
> Projection requires orthogonalizing m basis functions, each sampled as an $n×n$ vector, leading to a time complexity of $O(m^2n^2)=O(n^6)$ and a memory complexity of $O(mn^2)=O(n^4)$. Both runtime and memory become prohibitive at high resolutions. In contrast, only constant per-pixel cost is introduced from reflections and the evaluation of the basis $η$, resulting in an overall complexity of $O(n^2)$, and thus achieving efficiency in time and memory.
>
> **Generation experiments.** For locally supported representations, pixel values in different unit cells are initialized independently. After projection, these values are aggregated, reducing variance and producing overly smooth initialization. Since SDS optimization is sensitive to initialization, and approximately Gaussian-like distributions are more favorable for generation, this smoothness leads to degraded image quality. A pattern generation comparison with the projection-based method shows better generation quality (CLIP-A) across resolutions.
>
> ||n=64|n=128|n=256
> -|-:|-:|-:
> $α$=0.25|3.8|3.1|3.5
> $α$=0.5|4.0|3.4|3.6
> $α$=1.0|4.0|3.4|3.5
> Ours|**4.3**|**4.2**|**4.0**
>
> > **C3. Position of the paper in the ICML community**
>
> Good suggestions! We clarify the positioning of our work.
>
> **Zero-shot controllable generation.**
> Since the emergence of diffusion models, controllable generation has been an important topic in the ICML community, with zero-shot methods based on generative priors being particularly attractive. Our work can be viewed as a case of controllable generation, focusing on embedding symmetry constraints into diffusion-based optimization. The problem is inherently zero-shot, as real-world datasets rarely contain strictly symmetry data. Unlike prior approaches that rely on regularization terms, constraints are enforced through a parametric representation. Effectiveness is demonstrated in visual design (e.g., wallpaper, papercutting, and fence patterns), and further extended to material design, demonstrating broader applicability.
>
> **Mechanical metamaterial generation.**
> We apply the same framework to mechanical metamaterial design, where structures are represented as binary patterns and symmetry is closely related to physical properties. The goal is to generate symmetric structures with high mechanical performance. A base diffusion model is trained on topology-optimized unit cells to maximize the bulk modulus c. Symmetry is enforced through the parametric representation, while the SDS loss guides the generation toward high-quality designs. Results on 10 planar groups with 1000 samples show that the method can generate high-quality structures while satisfying the volume constraint (MAE < 1.5%) without symmetry data. Detailed results are provided in the supplementary (Figs. 1–3).
>
> > **C4. Computation of basis**
>
> We thank the reviewer for the suggestion. Due to the repetitive derivations, we present a general procedure in Sec. 4, while full dissusion and results are provided in the appendix. These cover all plane groups, so no additional computation is required.

---

> > ### Author Rebuttal · Reviewer_xBcx · 2026-04-03
> >
> > Dear authors, thanks for the detailed replies. This resolves my questions; I will raise my score accordingly.

---

> > > ### Author Response · Authors · 2026-04-07
> > >
> > > Thank you for your follow-up response. We are very glad that our rebuttal has addressed your concerns and clarified the questions you previously raised. We also sincerely appreciate your recognition of our efforts during the rebuttal process.

---

### Decision · Program_Chairs · 2026-04-30

**Decision:**

Accept (regular)

**Comment:**

The initial reviews were positive; the rebuttal and the discussion with the reviewers were fruitful. The reviewers' consensus is to accept the work. The final version should incorporate the corrections and clarifications provided in the rebuttal/discussions. The reviewers felt that the work progresses an important subarea in ML.